# Integrating multiplexed imaging and multiscale modeling identifies tumor phenotype conversion as a critical component of therapeutic T cell efficacy

## Graphical abstract

## Authors

John W. Hickey, Eran Agmon, Nina Horowitz, ..., John B. Sunwoo, Markus W. Covert, Garry P. Nolan

## Correspondence

mcovert@stanford.edu (M.W.C.), gnolan@stanford.edu (G.P.N.)

## In brief

Hickey et al. integrated multiplexed CODEX tissue imaging with multiscale modeling, showing that initial T cell phenotype affects tumor conversion to anti-proliferative state, critical for controlling growth rate. Structural reprogramming also maintains T cell phenotype, aiding in continual tumor killing. Findings stress the importance of back-and-forth modeling and spatial-omics, guiding new T cell treatment designs.

## Highlights

- T cells' early induction of non-proliferative tumor phenotype crucial for efficacy

- Multiscale modeling is built and integrated with multiscale, spatial-omics data

- Distinct spatial environments in tumors are key for T cell phenotype preservation

- Modeling adds multiscale spatial dynamics, prompting new spatial-omics experiments

Hickey et al., 2024, Cell Systems *15*, 322–338
April 17, 2024 © 2024 The Author(s). Published by Elsevier Inc.

**Cell Systems**

**_CellPress_**

## Article

# Integrating multiplexed imaging and multiscale modeling identifies tumor phenotype conversion as a critical component of therapeutic T cell efficacy

John W. Hickey,[1,2,3,9] Eran Agmon,[4,5,9] Nina Horowitz,[4] Tze-Kai Tan,[2,6] Matthew Lamore,[7] John B. Sunwoo,[6,8] Markus W. Covert,[4,*] and Garry P. Nolan[2,10,*]

[1]Department of Microbiology & Immunology, Stanford University School of Medicine, Stanford, CA 94305, USA
[2]Department of Pathology, Stanford University School of Medicine, Stanford, CA 94305, USA
[3]Department of Biomedical Engineering, Duke University, Durham, NC 27708, USA
[4]Department of Bioengineering, Stanford University, Stanford, CA 94305, USA
[5]Center for Cell Analysis and Modeling, University of Connecticut Health, Farmington, CT 06032, USA
[6]Institute for Stem Cell Biology and Regenerative Medicine, Stanford University School of Medicine, Stanford, CA 94305, USA
[7]Department of Biomedical Engineering, Johns Hopkins University, Baltimore, MD 21205, USA
[8]Department of Otolaryngology, Head and Neck Surgery, Stanford Cancer Institute Stanford University School of Medicine, Stanford, CA 94305, USA
[9]These authors contributed equally
[10]Lead contact
*Correspondence: mcovert@stanford.edu (M.W.C.), gnolan@stanford.edu (G.P.N.)

### SUMMARY

Cancer progression is a complex process involving interactions that unfold across molecular, cellular, and tissue scales. These multiscale interactions have been difficult to measure and to simulate. Here, we integrated CODEX multiplexed tissue imaging with multiscale modeling software to model key action points that influence the outcome of T cell therapies with cancer. The initial phenotype of therapeutic T cells influences the ability of T cells to convert tumor cells to an inflammatory, anti-proliferative phenotype. This T cell phenotype could be preserved by structural reprogramming to facilitate continual tumor phenotype conversion and killing. One takeaway is that controlling the rate of cancer phenotype conversion is critical for control of tumor growth. The results suggest new design criteria and patient selection metrics for T cell therapies, call for a rethinking of T cell therapeutic implementation, and provide a foundation for synergistically integrating multiplexed imaging data with multiscale modeling of the cancer-immune interface. A record of this paper's transparent peer review process is included in the supplemental information.

## INTRODUCTION

Cancer is a complex system of interactions that unfold across molecular, cellular, and tissue scales (Figure 1A). Adoptive T cell immunotherapy—in which patients are given T cells specific for cancer—causes a systems-level perturbation to cancer and has shown decisive clinical results in certain types of cancer but limited efficacy in solid tumors.[1–9] Indeed, much still needs to be learned about the manners by which infused cellular products cause effective therapeutic results. For instance, it is well understood that T cell phenotype matters, but how does cancer cell phenotype influence T cell therapy efficacy? Can infused T cells transform cancer phenotype, and is this related to T cell phenotype at the time of therapeutic delivery? Are there additional mechanisms for T cell phenotype maintenance related to their environment? How does the phenotype of the T cell product affect restructuring of the tumor tissue?

Such questions remain unanswered given the difficulty of interrogating the native cancer-immune state that provides sufficiently reflective biological measurements that would allow researchers to build models to simultaneously capture the multiple scales (molecular, cellular, and tissue) of cancer. Single-cell measurement technologies have allowed characterization of molecular changes in intracellular processes but do not reveal the spatial features of intercellular interactions in cancer.[10,11] On the other hand, traditional histologic approaches can capture spatial features of cancer but is limited to only a few molecular markers at once—restricting the ability to co-define cell types or cell phenotypes _in situ_.[12–14] Computational modeling has been largely restricted to a single biological scale for either description or source of input data, which limits the ability of modeling to accurately predict interactions across multiple scales. Consequently, methods are needed that can provide data that both deconstruct cancer's interaction networks at

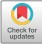

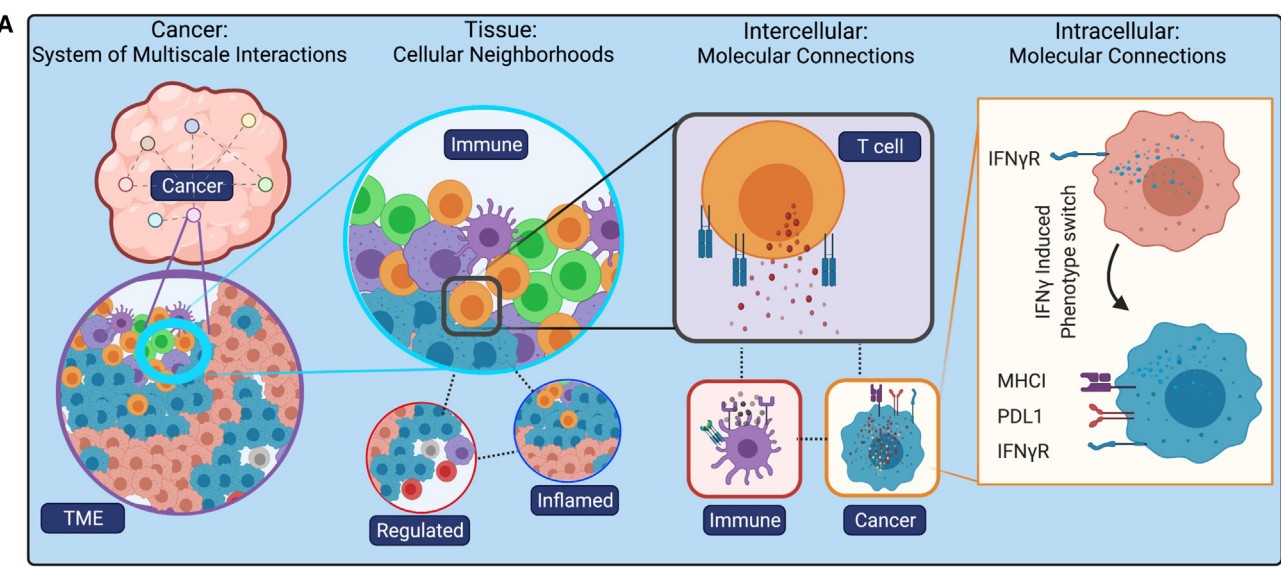

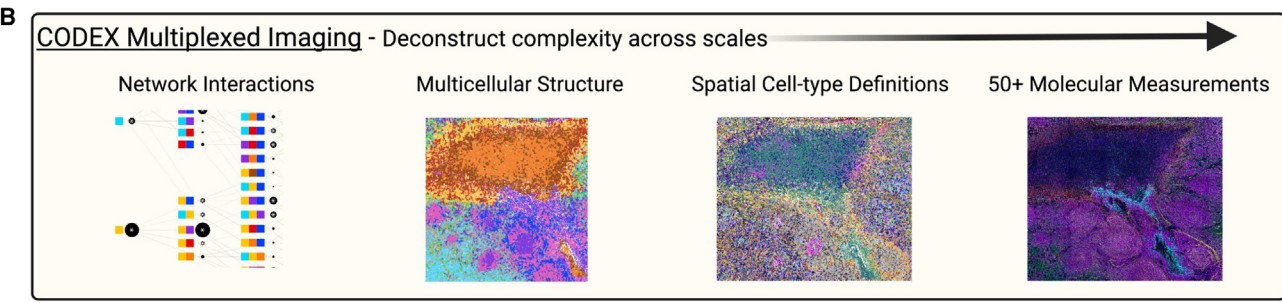

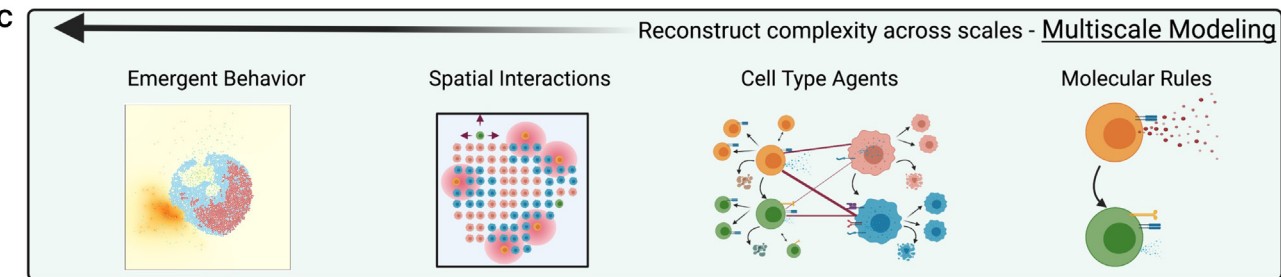

**Figure 1. Cancer is a system of network interactions, and its analysis requires methods that can deconstruct and reconstruct the complexity at multiple scales**

(A) At the tissue scale, multicellular neighborhoods form to make larger tissue structures and organs. At the cellular scale, cells engage with each other through intermolecular interactions, and intracellular interactions mediate cellular function.

(B) CODEX imaging enables multiplexed molecular measurements of 50 or more proteins that can be quantified at a single-cell level. These molecular profiles can be used to define both cell types and states. Using the spatial features of the data, multicellular structures can be identified based on conserved composition. Finally, network interactions across these scales can be interpreted to fully "deconstruct" the complexity of a tissue.

(C) Multiscale modeling enables "reconstruction" of complex biology across scales. Models are defined by molecular rules for cell agents that facilitate interactions. These interactions happen within a spatial microenvironment and result in emergent biological behavior. Models can guide hypothesis generation.

multiple scales and allow accurate modeling and reconstruction of such networks that in turn allow testing predictions or hypotheses to be made.

Multiplexed imaging is a recently developed technology that enables deconstructing the complexity of tissues from the top-down with spatial features preserved (Figure 1B).[15] The ability to probe more than 50 markers simultaneously using the CO-Detection by indEXing (CODEX) multiplexed imaging platform

makes it possible to identify molecules, cell states, cell types, and network interactions in space.[16,17] The concurrent development of computational systems biology approaches has facilitated quantifying and identifying key network interactions from this data such as multicellular neighborhoods.[18]

Multiscale modeling is a complementary approach for discovering critical interactions, by reconstructing the complexity of tissues from the bottom-up with computational simulations

(Figure 1C).[19–24] *Vivarium* is a recently introduced software tool that simplifies multiscale modeling, making it possible to connect modules of diverse mechanistic models into integrative simulations that cover multiple spatial and temporal scales.[25] This enables leveraging extensive prior knowledge about relevant biological mechanisms that were measured separately, as demonstrated previously with the construction of spatial bacteria colony models.[26]

We leveraged both CODEX multiplexed imaging and the *Vivarium* multiscale modeling software to understand the interactions of T cell therapies with cancer at multiple scales. To date, most studies have employed either top-down (deconstructing the data through analysis)[27,28] or bottom-up (reconstructing the data with mechanistic models)[29,30] approaches to the study of cancer. However, there is synergy in employing both approaches simultaneously to drive discovery of a more accurate tissue representation. As demonstrated here, multiscale modeling can be used to identify key points of the system for perturbation.

This marriage of multiscale modeling and multiplexed imaging share key data-driven features across scale, particularly the spatial positioning of distinct cells and molecules. Consequently, information from multiplexed imaging feeds multiscale agent-based models by providing more accurate parameter values, initial states (e.g., cell types and positions), and update rules. Multiplexed imaging data also represent a singular snapshot captured from valuable experimental or patient samples. Continuous monitoring at the individual cell level with similar detail is currently unfeasible. However, multiscale modeling presents an opportunity to augment our data, enabling the exploration of dynamic behaviors and the conduct of hypothetical experiments. For example, starting with a biopsy or tissue section, we can examine how different therapeutic approaches will play out.

By combining multiplexed imaging and multiscale modeling, we demonstrated that both tumor and T cell phenotype are key determinants of T cell therapeutic efficacy. T cell phenotype control has been a main focus to promote T cell longevity for killing cancer, with most approaches centering on intracellular molecular perturbation of T cells.[31,32] Much less attention has been given to tumor phenotype. Here, we observed that the conversion of tumor phenotype was a critical determinant in the control of cancer growth. Tumor phenotype conversion was dependent on a CD8+ T cell phenotype with ability to divide rapidly (memory-like) and secrete IFNγ (effector-like), suggesting this as a design criterion/goal for T cell therapies as well as a matching patient selection metric. The results suggest that integrating a multiscale modeling approach with multiplexed imaging data can provide a roadmap toward such a goal and establish it as a system for extending the dynamics of multiplexed imaging data.

## RESULTS

### Changing tumor phenotype to an inflammatory state enhances T cell recognition and killing

Cell therapies have emerged as a transformative therapeutic modality, with T cell therapies resulting in impressive clinical outcomes.[1–3,33] T cells achieve anti-tumor killing via direct recognition of tumor antigen presented in the context of major histocompatibility complex class I (MHC-I) through its cognate T cell receptor (TCR). Upon recognition, they secrete a number of effector molecules including cytotoxic granules that cause death in the target cell locally. Consequently, T cells offer an attractive approach to tumor therapy because of their antigen-specificity, proliferation, and long-term memory that enables durable responses.

However, the effectiveness of T cell therapies has primarily been observed in hematologic malignancies with genetically modified chimeric antigen-receptor (CAR) T cells,[4] which constitute a minor fraction of cancer-related mortality (only 5% of cancer deaths[34]). Furthermore, the broader implementation of T cell therapies has been hindered by systemic toxicities that limit their applicability.[35] Ongoing endeavors are focused on optimizing T cell functionality through the modulation of T cell phenotype,[27,36–38] designed to enhance capacity for self-renewal or killing.[31,32,39–42]

In parallel, there have been a number of immunotherapies developed to unleash endogenous, antigen-specific T cells within tumors.[43] For example, some checkpoint blockade therapies block T cell inhibition with antibodies targeting inhibitory signaling molecule programmed cell death protein 1 (PD-1) found on T cells. Alternatively, therapeutic antibodies have been made to block PD-1's cognate receptor of programmed death-ligand 1 (PD-L1), which tumor cells often express.[44] Consequently, we hypothesized that the tumor cell phenotype could be just as crucial in shaping T cell responses within a tumor as the phenotype of T cells themselves.

To understand the influence of tumor cell phenotype on T cell therapeutic efficacy, we evaluated how differences in tumor phenotype influenced killing by T cells *in vitro*. We incubated B16-F10 mouse melanoma cells with IFNγ overnight to induce intracellular signaling, which is known to be critical to the function and phenotype of cancer cells.[45–48] As shown by CyTOF analysis, there was a phenotype change in about half of the tumor cells—characterized by upregulation of both anti-inflammatory (PDL1) and inflammatory (H2Db) surface markers in the group treated with IFNγ (Figure 2A).

We hypothesized that a drastic change in tumor cell phenotype would be accompanied by a change in metabolism. Consequently, we compared the tumor cells by staining with a CyTOF panel focused on cellular metabolism (Table S2). Notably, we observed that for IFNγ-treated tumor, there is an increase in cells that were quiescent or in the G0 phase of the cell cycle (negative for IdU and pRb (S807 S811)) (Figure 2B). These reductions in cellular proliferation were enriched for cells that expressed high levels of MHC-I (H2Kb) (Figure S1A), and MHC-I (H2Kb)-positive cells that entered the cell cycle had a lower mitotic index than tumor cells that do not express high levels of MHC-I (Figure S1B). Furthermore, cells treated with IFNγ also exhibited lower levels of glucose-6-phosphate dehydrogenase (G6PD) enzyme (Figure S1C). This result suggests a reduction in the pentose phosphate pathway, indicating both an impaired cellular antioxidant, DNA synthesis, and cell division processes. Overall, these data suggest that the IFNγ not only causes inflammatory and anti-inflammatory markers like PD-L1 and MHC-I but also substantial metabolic changes that inhibit proliferation of the tumor cells.

The IFNγ-treated tumor cells express both anti-inflammatory and inflammatory molecules, but it was unclear whether this

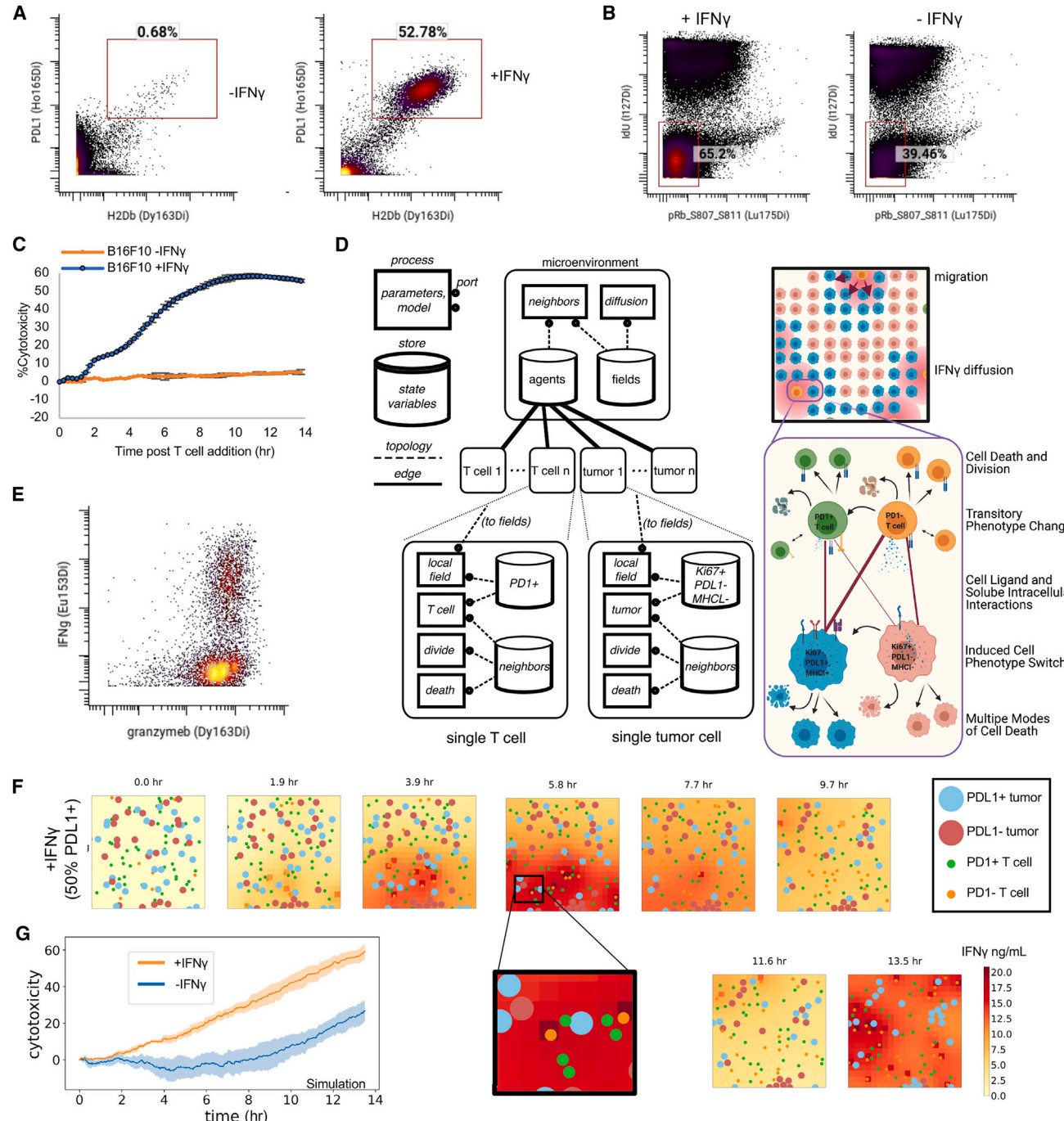

**Figure 2. Changing tumor phenotype to an inflammatory state enhances T cell recognition and killing**

(A) PD-L1 and H2Db per cell levels as measured by CyTOF of B16-F10 tumor cells after being incubated with IFNγ or no IFNγ for 18 h.

(B) CyTOF staining of B16-F10 tumor cells cultured either with or without the presence of IFNγ measuring the IdU and pRb S807–S811 and gate indicating double-negative populations in G0 phase of cell cycle.

(C) Percent killing of cognate tumor cells over time by expanded therapeutic T cells pre-incubated with IFNγ or not. Tumor and T cells were incubated at a 1:1 ratio (mean of $n$ = 3 replicates with error bars showing SEM).

(D) Multiscale agent-based model of the tumor microenvironment used to understand critical components governing efficacy of adoptive T cell therapies at multiple levels of scale.

(E) Evaluation of per cell levels of effector molecules, granzymeB and IFNγ, of restimulated therapeutic PMEL CD8$^+$ T cells after 10 days of activation.

(F) Snapshots of agent-based modeling results showing results from a simulation that was initialized to mirror *in vitro* killing by expanded therapeutic T cells pre-incubated with IFNγ or not.

(G) Cytotoxicity levels from multiscale agent-based modeling of initializing simulations with tumors that had similar phenotype to input tumor cells in (B), indicating being treated with or without IFNγ (mean of $n$ = 5 replicates with shading showing SEM).

phenotype would inhibit T cell killing or promote more efficient killing. We performed a dynamic *in vitro* killing assay with a 1:1 ratio of cognate, antigen-specific (PMEL), activated CD8$^+$ T cells and B16-F10 tumor cells or IFN$\gamma$-pretreated B16-F10 tumor cells. IFN$\gamma$-treated tumors were killed much more effectively (50% vs. 7% at 13 h) (Figures 2C and S1D).

To formulate a mechanistic explanation why this tumor phenotype conversion led to such enhanced killing, we created a multi-scale agent-based model using *Vivarium*. The ability to create agents at multiple scales (e.g., molecular and cellular) governed by defined biological rules within a spatial environment can be used to test multiple hypotheses and detect critical inflection points in network structures and dynamics. Figure 2D illustrates how multiple biological scales of cell-state interactions are modeled with *Vivarium* with a simplified wiring diagram (left panel).

We created the model to be focused on interactions between two subsets of therapeutic T cells and the two subphenotypes of cancer cells. We modeled fundamental immune-tumor interactions in T cell therapy, for instance (1) PD-1$^+$ T cells interaction with PD-L1 on the surface of PD-L1$^+$ MHC-I$^+$ tumor cells; (2) PD-1$^-$ T cells/PD1$^+$ T cells recognition of tumor cells through interactions of the TCR with MHC-I on tumor cells wherein PD-1$^-$ T cells can be converted to PD-1$^+$ T cells through repeated stimulation of their TCR; (3) CD8$^+$ T cells' release of IFN$\gamma$, which then converts tumor cells to PD-L1$^+$ MHC-I$^+$ tumor cells; and (4) CD8$^+$ T cells' release of cytotoxic granules that enable localized tumor killing (Figure 2D, right panel).

To create this model required encoding prior knowledge and lab-derived parameter values to create the rules governing individual cancer cells and T cells (see Table S1; Figures S2 and S3). These parameters include data sourced from both deep molecular and time-resolved dynamic interactions of T cells and tumors. For example, in our model, T cell migration reflects observed physiological changes, with distinct velocities based on biological input within the model, differing for PD-1$^-$ and PD-1$^+$ T cells and whether T cells are engaging with tumor cells. Specifically, T cell motility is decreased upon encountering MHC-I$^+$ antigen-presenting tumor cells. These modeled behaviors are informed by empirical *in vivo* imaging, utilizing techniques such as intravital and 2-photon microscopy to track T cell dynamics within tumors.[45,49] Such empirical grounding of the model parameters ensures its simulation accurately captures the biological activity of T cells in the tumor microenvironment. We additionally encoded intracellular and intercellular interactions in *Vivarium* and tuned the parameters by comparing process performance with expected behavior standards. *Vivarium* also enables environmental interaction such as migration and diffusion of molecules. We provide much more extensive documentation regarding model development, rationale, and biological background within supplemental information accompanying the manuscript. Also, Jupyter notebooks explaining the development and rationale behind the model are provided on the project's GitHub repository, where example notebooks ran different permutations of the model for testing. A documented code base also describes the rules and parameters of the model and can be found in the STAR Methods.

We then evaluated whether our *in silico* model would show the expected higher T-cell-killing efficacy of an IFN$\gamma$-induced tumor phenotype. To accurately initialize our model, we measured

*in vitro* levels of PD-1, effector molecules involving killing (granzyme B, perforin), and IFN$\gamma$ expression both on a single-cell level and quantitatively over time from restimulated PMEL CD8$^+$ T cells used for the *in vitro* killing assay (Figures 2E, S4A, and S4B). We used these values, in addition to *in-vitro*-measured PD-L1 and MHC-I expression by tumor cells (Figure 2A) and changes in cellular metabolism (Figure 2B), as inputs to initialize both T cells and tumor cells in the model and ran modeling simulations that mimicked our *in vitro* killing assay setup.

To observe cellular behavior from the *T cell-killing* simulation we plotted "snapshots" as output for the IFN$\gamma$-treated tumor (Figure 2F, time of each snapshot shown above starting with the initial state a 0–13.5 h). This figure thus contains rich information across spatial, time, agent, and molecular dimensions, where T cell migration, tumor proliferation, tumor phenotype change, secretion of IFN$\gamma$, and tumor killing can be observed over the course of 13.5 h (Figure 2F, larger circles indicate tumor cells, smaller circles indicate T cells, color represents the phenotype of a given cell type, and the red background color represents the local concentration of IFN$\gamma$). For instance, zooming in on the 5.8 h snapshot, there is a PD-1$^+$ T cell (green) interacting with a PD-L1$^+$ tumor cell (light blue) and several other tumor cells (upper left of zoomed figure) with no T cells next to them. In the next snapshot (7.7 h), this tumor cell has been killed, and other tumor cells have not moved or been killed, while the T cells have all migrated. This zoomed-in area (5.8 h) also has a concentration of soluble IFN$\gamma$ around 18 ng/mL with some variation in the grid squares. IFN$\gamma$ has increased with the number of T cell and tumor cell interactions from a starting concentration of 0 at snapshot 0 h and decreases by the next snapshot (7.7 h) due to some T cells beginning to downregulate TCR and tumor uptake of soluble IFN$\gamma$. This led to a majority of the tumor cells changing phenotype, as can be noticed from 11.6 h to 13.5 h window (salmon color to light blue color).

In this simulation, we quantified cytotoxicity to enable a comparison with our *in vitro* data. Cytotoxicity was quantified by evaluating the number of cell deaths and normalizing the results against a simulation lacking T cells. In addition to using lab-derived molecular parameters as input, we initialized the simulation with identical cellular parameters used within the *in vitro* experiment, such as the same effector-to-target ratio, percentage of T cell phenotypes, and duration in culture. Quantification of killing by cytotoxicity indicated an important role of tumor phenotype on T cell killing even at early time points (Figure 2G). Moreover, when tumor cells were pretreated with IFN-$\gamma$, we observed a remarkable increase in cytotoxicity, reaching nearly 60% by 12 h compared with untreated tumor cells incubated with an equivalent number and type of T cells. This finding mirrors the observed increase in our *in vitro* experiment's cytotoxicity (Figure 2C). Consequently, our simulation results not only replicated the kinetics and magnitude of our *in vitro* killing data but also reinforced the overall conclusion. This alignment validates both our model setup and molecular parameters with dynamic data.

Thus, starting with a population of tumor cells that have a PD-L1$^+$ MHC-I$^+$ phenotype has a large impact, potentially due to increased levels of interactions between tumor cells expressing higher levels of MHC-I and T cells. In summary, with identical cognate T cell inputs but different cancer cell phenotype ratio

inputs, as observed *in silico* and *in vitro* data demonstrated that conversion of tumor phenotype to an inflammatory state enhances the ability of T cells to kill tumor cells, resulting in fewer total cancer cells and an inflammatory microenvironment conducive to T cell killing.

### Initial phenotype of the input therapeutic T cells drives conversion of tumor cells to an inflammatory phenotype

Controlling T cell phenotype during *ex vivo* expansion prior to therapeutic transfer is expected to be critical, especially since cells are in foreign environments for extended periods of time.[37,50–52] Therapeutic T cell phenotype is known to cause dramatic differences in anti-tumor efficacy, especially from the perspective of T cell persistence and effector molecule expression.[31,32,39–42] Broadly, memory T cells are expected to persist longer and give rise to more daughter cells, whereas effector cells are expected to be shorter-lived and secrete effector molecules like perforin.[53,54] However, because T cells are usually isolated from subjects or dissociated from cancer tissues to be measured, the manners by which their phenotype relates to tumor phenotype, at the beginning and end of therapy, remain unknown. Indeed, clinical challenges and outstanding questions in targeting solid cancers are spatially related: e.g., T cell infiltration, local tumor antigen expression, and spatial co-enrichment with stimulating or inhibiting immune cells.[5–9,55–58] Thus, elucidating how these spatial relationships and multicellular interactions change based on therapeutic T cell features, particularly cytokine and effector molecule secretion remains understudied.

One approach shown to generate T cells with these distinct phenotypes, is via inhibition of acetyl-CoA production[27] by incubating CD8[+] T cells with 2-hydroxycitrate (2HC) during expansion (Figure 3A). Inhibition of acetyl-CoA formation pushes T cells toward memory stemness resulting in significantly better tumor control than conventionally activated T cells.[27] Indeed, cells incubated *in vitro* with 2HC had lower expression of PD-1[+] than the untreated cells (25% vs. 75% PD-1[+]) (Figure 3B). Additional characterization by CyTOF showed further subphenotypes that separate further beyond simple PD-1 staining,[59] but two main categories of memory and effector T cells were broadly separated by PD-1 status (Figure S4C), and we used these as inputs to our model.

We compared how these two phenotypes influence the ability of T cells to alter tumor cell phenotype under conditions found within the tumor microenvironment. For our previous *in vitro* experiments, the ratio of tumor to T cells was controlled; however, there are several barriers to entry into tumors *in vivo* (e.g., extracellular matrix, trafficking to non-tumor organ sites). Therefore, to create an accurate starting ratio of tumor to T cells in the tumor microenvironments, we transferred one million therapeutic CD8[+] T cells into each mouse bearing an established B16-F10 tumor that had been grown for 10 days. Harvesting tumors after adoptive T cell therapy showed that the CD8[+] T cell frequency in these tumors was approximately 1% of all cells (Figure 3C).

With biologically and therapeutically relevant initialization conditions for both phenotype and T cell ratio, we ran simulations with 1% CD8[+] T cells (12 T cells to 1,200 tumor cells) to compare our two differentially activated T cell phenotypes (with 2HC: 25% PD-1[+]; without 2HC: 75% PD-1[+]) as separate therapies over a period of 72 h. Snapshots from the simulations show distinct dy-

namics of IFNγ secretion, CD8[+] T cell proliferation, and tumor killing as well as different spatial phenomena relating to tumor phenotype (Figure 3D; Videos S1, S2, and S3). For example, tumor cells proliferated in all groups from 0 to 31.5 h. T cells also proliferated by 31.5 h, started to kill tumor cells, and converted tumor cells to PD-L1[+] phenotype in pockets (black arrows) for both 25% and 75% PD-1[+] T cell conditions. These T cell pockets have higher local concentration of IFNγ (red/brown) than other areas of the tumor, and highest levels of IFNγ in the 25% PD-1[+] T cell condition. By 73.3 h, the T cells killed enough tumor cells to coalesce into common pockets and escape the tumor bed (magnified area, bottom right of Figure 3D).

Quantification of the simulations showed that the starting condition of 25% PD-1[+] T cells inhibits tumor growth more effectively (~1,000 tumor cells at 60 h) than the starting condition of 75% PD-1[+] T cells (~3,000 tumor cells at 60 h) (Figure 3E). Under conditions without T cells, the tumor cells grew exponentially (~5,000 tumor cells at 60 h). These results were expected based on previous *in vivo* experiments.[27]

To explain the difference in tumor control *in silico*, we looked at the number of tumor cells killed in each of the conditions. Notably, control of tumor growth was not explained by the number of T-cell-induced killing events in the two conditions (25% PD-1[+] T cells, ~1,300 cell deaths at 60 h; 75% PD-1[+] T cells, ~1,200 deaths at 60 h) (Figures 3F and S5A). Since tumor phenotype made a large functional difference in our *in vitro* experiments and earlier simulations, we investigated the subphenotypes of tumor cells in these simulations. In the 25% PD-1[+] T cell condition, there was an earlier (at 18 h indicated by black arrow) and stronger conversion of tumors from a proliferative to non-proliferating inflammatory phenotype than in the 75% PD-1[+] T cell condition (Figure 3G).

The improved tumor control was linked to increased numbers of PD-1[−] T cells in the tumor, though in both conditions T cells became PD-1[+] over the course of the simulation (Figures 3H and S5B). This is not surprising, since it is known that repeated stimulation of antigen-specific T cells leads to exhaustion which includes phenotype changes resulting in lower cytokine secretion and cell division.[60–62] Similarly, since we mimic the design of our *in vivo* adoptive T cell therapy experiments, all CD8[+] T cells follow the same course *ex vivo*. Briefly, we use transgenic PMEL mice, wherein all CD8[+] T cells are specifically reactive to the melanoma-associated antigen gp100, expressed by the B16-F10 tumor cell line. We harvest the immune cells from PMEL mice and activate these cells for 10 days *in vitro* with cognate antigen and anti-CD3 activation, upon which the cells undergo considerable proliferation. The stimulatory regime for these transgenic, antigen-specific T cells, results in a relatively homogeneous activation state across the cell population, allowing for the initialization of treatment groups (e.g., with or without 2HC) with accurate phenotypic proportions that we previously characterized by CyTOF (Figures 3A–3C). Similarly, we inject B16-F10 tumor cells to recipient wild-type mice and allow them to grow for 9 days, lymphodeplete on day 9 with sublethal irradiation (mirroring clinical practice for adoptive therapy), and transfer T cells into recipient mice at day 10. Since all T cells are antigen-specific and all tumor cells express antigen, there is a high probability of interaction between antigen-specific T cells and gp100-expressing tumor cells within the tumor microenvironment. Consequently, we

**CellPress**

**Cell Systems**
Article

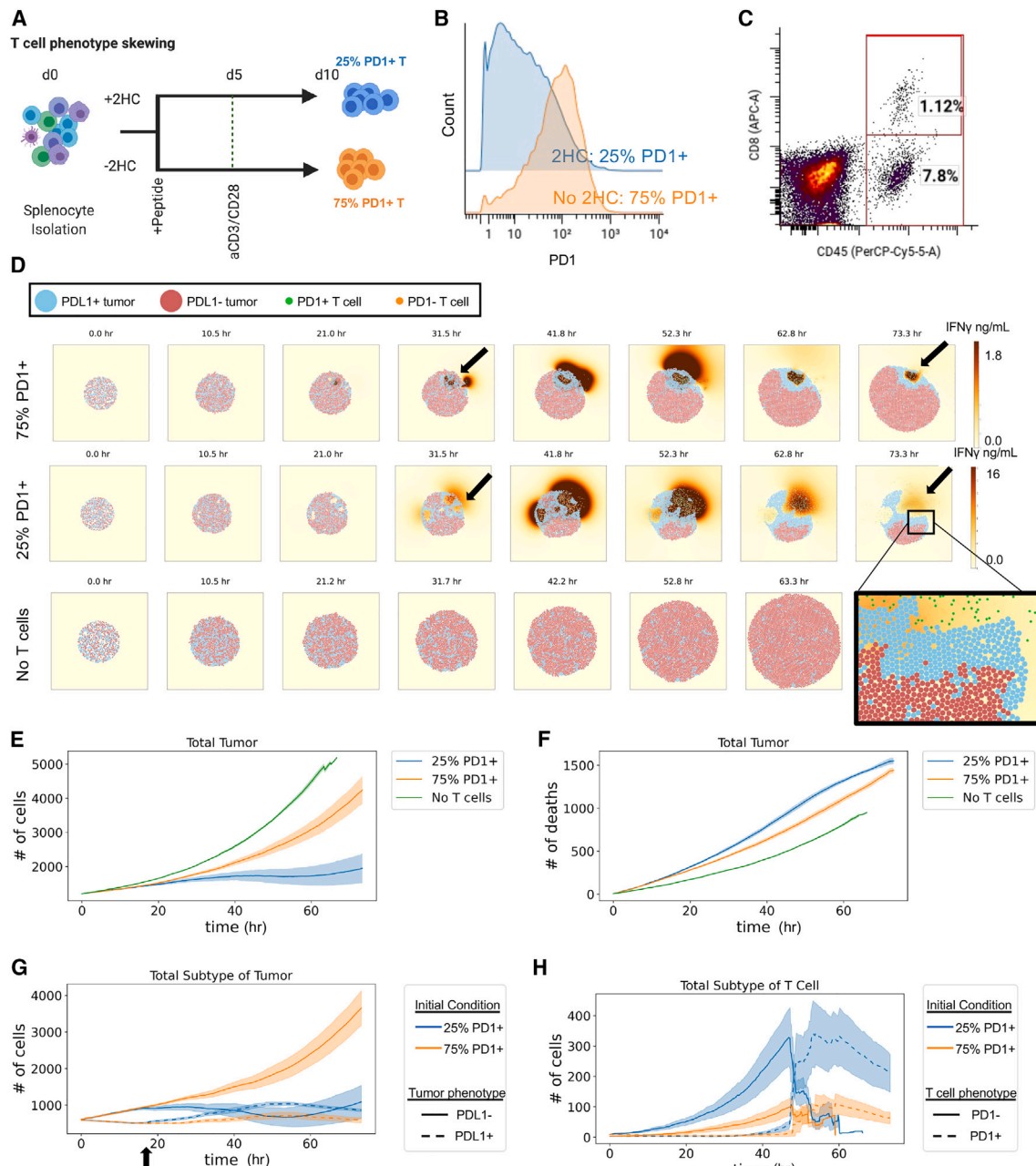

**Figure 3. The initial phenotype of transferred therapeutic T cells influences the ability of T cells to convert tumor cells to an inflammatory phenotype**

(A) Experimental layout for controlling T cell phenotype during *ex vivo* T cell expansion. Stimulating T cells in the presence of 2HC leads to a phenotypic shift, particularly in PD-1 where lower PD-1[+] cells are found in 2HC-treated condition as denoted by blue, where T cells stimulated in the absence of 2HC have higher levels of PD-1 and are denoted by orange.

(B) Histogram of per cell levels of PD-1 expression as measured by CyTOF of T cells treated with metabolic inhibitor 2HC or without the inhibitor.

(C) Percent of CD8[+] T cells within tumors post-treatment with therapeutically expanded T cells determined by flow cytometry that are positive for both CD45 and CD8. CD45[+] and CD8[−] cells (7.8%) represent non-CD8[+] immune cells within the tumor.

(D) Snapshots of simulation initialized with *in-vivo*-relevant cell numbers, ratios, and T cell phenotypes for 25% and 75% PD-1[+] T cell conditions compared with a simulation condition with no T cells. All simulations were initialized with a total of 1,200 tumor cells and 12 T cells with varying ratios of respective cell phenotypes.

(E) Total number of tumor cells over the course of the simulation that was 3 biological days under each condition.

(F) Number of tumor cell deaths over the time course of the simulation.

(G) Number of tumor cells separated by phenotype over the course of the simulation.

(H) Number of T cells separated by phenotype over the course of the simulation. For (E)–(H), mean of *n* = 4 replicates with shading showing SEM.

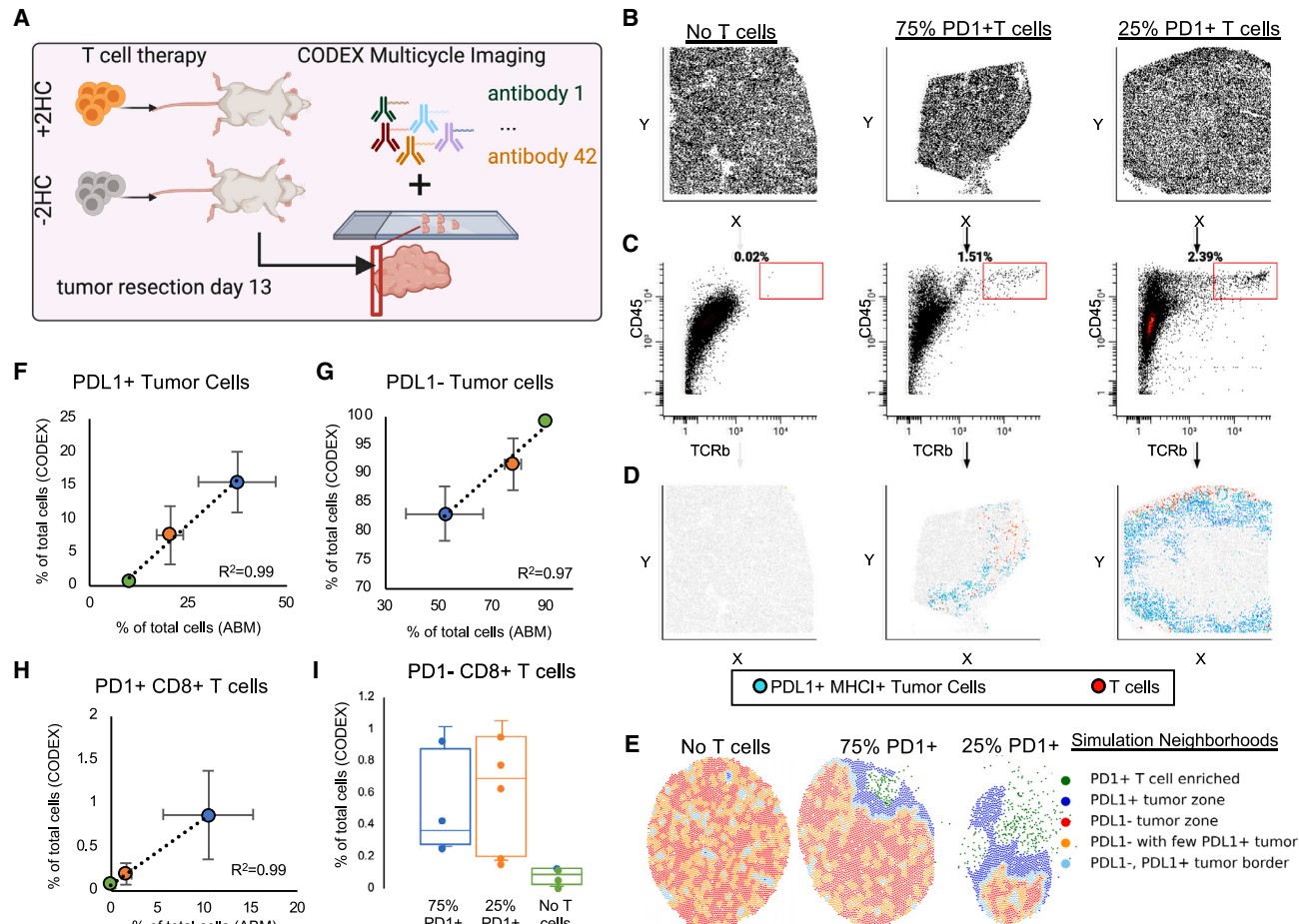

**Figure 4. T cells induce tumor cell phenotype conversion *in vivo***

(A) Experimental layout for *in vivo* adoptive T cell therapy and CODEX multiplexed imaging of tumors at day 3 post-treatment.

(B–D) CODEX multiplexed imaging results in single-cell data that are spatially resolved. Scatter plots for each treatment condition plotted for each cell in (B) x vs. y, (C) CD45 vs. TCRb, and (D) gated T cells (red) in the x axis vs. y axis together with PD-L1$^+$ MHC-I$^+$ tumor cells (blue)—to see spatial distribution of T cells in the tumor samples.

(E) Multicellular neighborhood analysis for each of the simulations at the day 3 endpoint reveals differential structures created by each of the responses, where responses are characterized into 5 overall neighborhoods.

(F–H) Correlation plots between percent of cells resulting after 3 days of T cell therapy for both CODEX multiplexed imaging of *in vivo* experiments and *in silico* simulations for (F) PD-L1$^+$ tumor cells, (G) PD-L1$^-$ tumor cells, and (H) PD-1$^-$ CD8$^+$ T cells (n = 4–5 per treatment group and per *in vivo* and *in silico* experiments; error bars represent SEM).

(I) PD-1$^-$ CD8$^+$ T cell percentages of total cells on day 3 from *in vivo* experiments as measured by CODEX multiplexed imaging (n = 4–5 per treatment group and per *in vivo* and *in silico* experiments; error bars represent SEM).

would expect a phenotype shift from the T cells that we transferred into the tumor. However, what was not expected was that the phenotype of the T cells would influence the ability to convert tumor phenotype and that this would play such a critical role in controlling the tumor growth rate. Thus, greater tumor control from phenotype-switched T cells was due to greater ability to inhibit tumor proliferation rather than differences in inhibition from T cell direct killing.

## T cells induce tumor cell phenotype conversion *in vivo*

Our attempts to reconstruct the complexity of the system in both *in vitro* and *in silico* models suggested the critical importance of a tumor phenotype transformation by T cells. To study this *in vivo*, we used CODEX multiplexed imaging to enable measurement of

tumor phenotype changes in a spatial context in a therapeutically relevant adoptive T cell model (Figure 4A).[27,63,64] Specifically, we transferred T cells that were treated or not with 2HC into B16F10 established tumors (day 10) and harvested them 3 days after treatment (day 13) for CODEX multiplexed imaging. Imaging was performed with a 42-antibody panel designed to detect cancer phenotypic markers (e.g., PD-L1, H2Kb, and Ki67); immune cell-type-defining markers (e.g., CD3, CD4, CD8, and F4/80); and functional markers (e.g., PD-1 and CD27).[59]

Because we observed spatial restriction of the T cells that was also associated with proximity to PD-L1$^+$ MHC-I$^+$ tumor cells within our simulations (Figure 3D), we predicted we would see the same proximal events in our *in vivo* data. To compare results

between *in vivo* and simulations, we first examined the positions of T cells within tumor sections, since CODEX generates single-cell data that enables cell-type identification.[65] Because of this, we can visualize each individual cell in x and y coordinates (Figure 4B). Each individual cell contains the quantification of each protein marker, so we used this to gate T cells and PD-L1[+] MHC-I[+] tumor cells (Figure 4C). An analysis of the locations of T cells (TCRb[+] CD45[+], red) and PD-L1[+] MHC-I[+] tumor cells (blue) in each tumor section revealed spatial restriction of the T cells and co-localization with inflamed tumor cells in both T cell conditions (Figure 4D), mirroring the *Vivarium* modeling prediction.

To further investigate this phenomenon, we compared the *in vivo* CODEX data and *in silico* modeling output by performing multicellular neighborhood analysis on our simulation data since both datasets preserve spatial features (Figure 4E).[17] In the *in silico* data, there are neighborhoods representing borders of immune attack on the tumor from both T-cell-treated groups, with larger borders in tumors treated with 25% PD-1[+] T cells than 75% PD-1[+] T cells and disorganized tumor neighborhoods in tumors without T cell treatment (Figure 4E, compare middle figure to rightmost). Here, the borders were enriched in both T cells and PD-L1[+] MHC-I[+] tumor cells. These features of the *in silico* models were consistent with major structural components in the *in vivo* data,[59] suggesting the important role of coordination of the neighborhood interactions, structure, and function. We verified that tumor cells that expressed PD-L1[+] and MHC-I[+] were also Ki67[−] and that tumor cells that were Ki67[+] were PD-L1[−] and MHC-I[−] (Figure S6A). Moreover, co-localization of transformed T cells and tumor cells supports the hypothesis that T cells are responsible for tumor phenotype conversion, which we observe both *in vitro* and *in vivo*.

We also compared the cell-type percentages in the CODEX data with *in silico* percentages at day 3 to understand whether relative phenotype conversion rates were similar. We found good correlations of ending percentages for PD-L1[+] tumor cells (Figure 4F, R = 0.99), PD-L1[−] tumor cells (Figure 4G, R = 0.97), and PD-1[+] T cells (Figure 4H, R = 0.99). Similarly, we also tested the relationship of the cell populations at day 1 from our simulations compared with *in vivo* data from our CODEX imaging and saw good correlation in cell-type frequencies (Figure S6B). However, this was not the case for PD-1[−] T cells where the number of PD-1[−] T cells were much lower in the *in silico* model, especially by day 3 post-treatment, than we observed in the CODEX multiplexed imaging data (Figures 3H and 4I). Because the death rate for PD-1[−] T cells was lower than for the PD-1[+] T cells in the *in silico* model, this suggests that death does not account for the ratio difference (Figures S6C and S6D). This result suggested that there was a mechanism for T cell phenotype maintenance missing from our model.

### Spatial location of T cells impacts the ability to maintain phenotype

We hypothesized that discrepancies between our model and *in vivo* data provide an avenue to uncover biological mechanisms of T cell phenotype preservation by combining CODEX data with our multiscale model. Part of the reason we chose an agent-based model design is because it complements the spatial and compositional structure of CODEX multiplexed imaging. Since

we quantify protein expression at the single-cell level with spatial coordinates that are linked to cell type, we can directly import our CODEX multiplexed imaging data as initial states of our multi-scale simulations (Figure 5A). This uniquely allows us to use the model to interpret complex CODEX data, extend the dynamics of static multiplexed imaging data, and establish more accurate initial conditions.

We were particularly interested to see if initializing our simulations with spatial information obtained from *in vivo* data would reveal the reason that the ratio of PD-1[+] T cells to PD-1[−] T cells observed in the *in silico* model was higher than in the CODEX multiplexed imaging data (Figure 4I). We took a region of ~2,000 cells from CODEX images to initialize our model and then simulated the changes in the tumor microenvironment over 3 days. Tumor cell growth rates from each condition matched expectations and previous simulations (Figures 5B and S7A–S7C). Similarly, we observed an exhaustion of the T cells in both T cell treatment conditions after about 50 h; however, only in the 75% PD-1[+] T-cell-treated condition did all the T cells become exhausted, whereas in the 25% PD-1[+] T-cell-treated condition a proportion of PD-1[−] T cells remained at 72 h (~250 PD-1[−] T cells) (Figures 5C and S7D). This contrasts with the results of our previous simulation (Figure 3H), where both T cell treatments led to complete phenotype conversion to PD-1[+] by the end of 3 days (Figure 5D).

Since T cells become exhausted through chronic TCR stimulation, we hypothesized that a spatial relationship might be responsible for phenotype preservation. The snapshots from the simulations initialized with CODEX data revealed that in the *in silico* 25% PD-1[+] T cell condition there is a front of attacking T cells on the periphery of the tumor and that the tumor cells on the border with these T cells are inflamed (Figure 5E). This can be seen by the increased IFNγ concentration at the edge of the periphery of the tumor (brown) and by zooming in on the interface of T cells and tumor cells at 30.8 h (Figure 5E, blue square). In contrast, the T cells in the *in silico* 75% PD-1[+] T cell condition initially attacked from the periphery but were soon surrounded by proliferating cancer cells seen at 30.8 h and remained so until 72 h (Figure 5E, orange square). This suggests that the spatial location of T cells on the periphery of tumors may be critical for T cell phenotype maintenance.

### Conversion of tumor cell phenotype is more critical for tumor control than T cell phenotype preservation

We hypothesized that since T cells were on the periphery of the tumor, they could escape chronic stimulation and thus delay exhaustion (Figure 6A). To test this, we initialized our simulations with 25% PD-1[+] T cells conditions (as in Figure 3), except that T cells were located outside the tumor bed rather than inside. Similar to our CODEX-initialized experiment (Figure 5), we observed tumor phenotype changes on the periphery of the tumor where tumor cells contacted T cells (Figure 6B, increase in light blue PD-L1[+] tumor cells). When the T cells were initialized outside the tumor, we saw a dramatic increase in total numbers of T cells compared with the numbers when T cells were initialized on the inside of the tumor (Figure 6C). This increase resulted from a delay in T cell exhaustion (Figures S8A and S8B).

Interestingly, despite the much higher numbers of T cells located outside the tumor, in our simulations, the tumors with

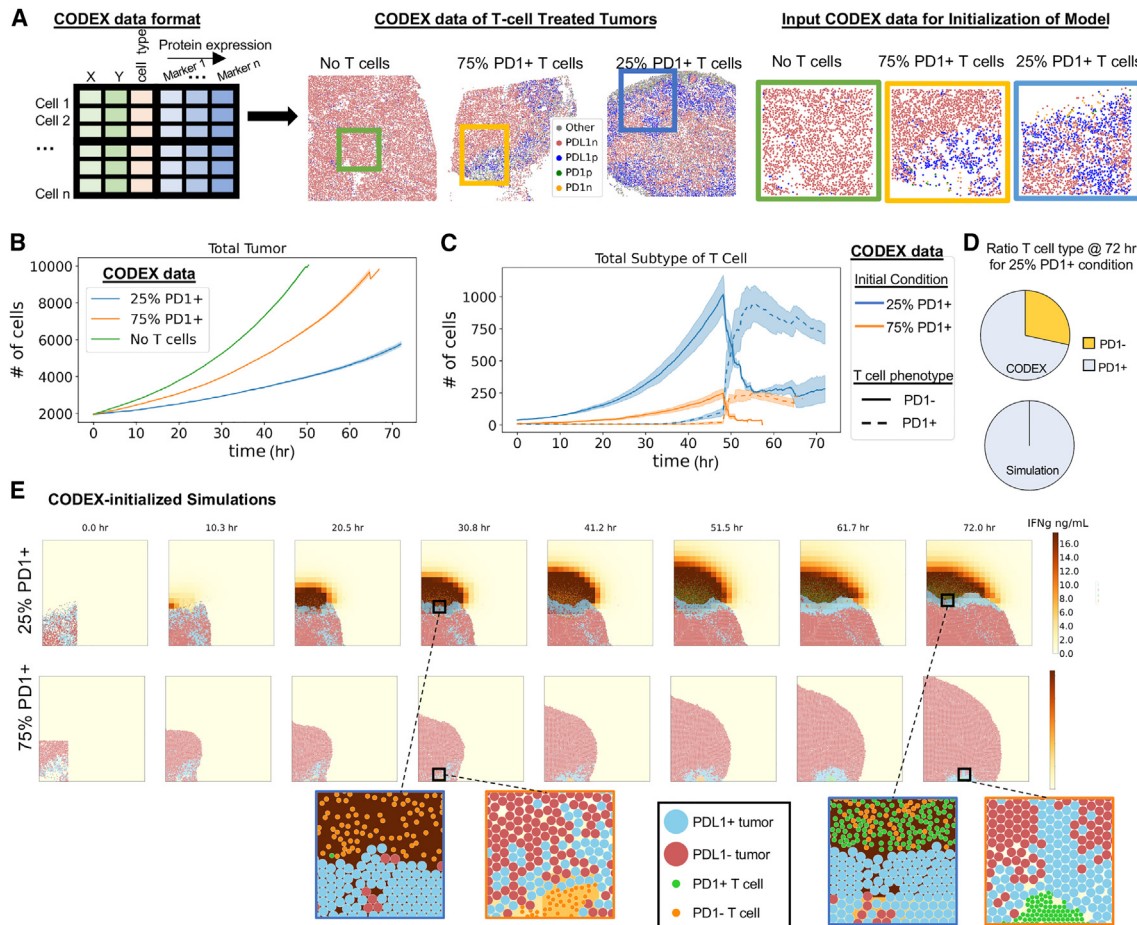

**Figure 5. Spatial location of T cells impacts the ability to maintain phenotype**

(A) Left: CODEX multiplexed data are amenable to initialize multiscale-agent-based models because it has single-cell information of cell type, x and y positions, and molecular protein expression. Middle: cell-type maps of CODEX images of tumor sections. Rectangles indicate subsets of 2,000 cells used to initialize the model. Right: high-magnification images of the areas indicated by rectangles in the middle panels.

(B) Number of tumor cells in T-cell-treated and control groups as a function of simulation time (mean of $n$ = 4 replicates with shading showing SEM).

(C) Number of PD-1$^+$ and PD-1$^-$ T cells in each T-cell-treated group as a function of simulation time (mean of $n$ = 4 replicates with shading showing SEM).

(D) Percent of PD-1$^+$ and PD-1$^-$ T cells at the end of the 72-h simulation started either with initial conditions of 25% PD-1$^+$ T cells (used for Figure 3H) or the conditions based on CODEX data (used in C) (average of $n$ = 4 simulations for both conditions).

(E) Snapshots of the tumor from agent-based modeling condition 25% PD-1$^+$ T cell and 75% PD-1$^+$ T cells from simulations initialized with CODEX data that illustrate spatial restrictions of T cells and zoomed-in regions to indicate phenotype status of T cells over time.

T cells initialized outside the tumor grew more quickly than tumors with T cells initialized inside (~4,500 cells vs. ~2,500 cells at 75 h) (Figure 6D). Consequently, delay of T cell exhaustion came at the cost of enhanced early tumor growth rates (Figure 6E), which even larger numbers of T cells were not able to overcome in the long run. Thus, trading less tumor engagement over time for an increased number of T cells in the future essentially gives the tumor a head-start in proliferation enabling exponential growth rates that extra T cells are not able to counteract through slightly enhanced killing later (Figures 6F and S8C). We confirmed this result by setting the 75% PD-1$^+$ T cell condition from Figure 5 to start in the center, preventing T cells from becoming trapped (Figure S8D). Here, we also observe an increase in the tumor growth rate for the center condition that the T cells are able to escape the tumor microenvironment and preserve phenotype but limit the earlier tumor conversion

(Figures S8E and S8F). Since resting preserves phenotype, but does not enhance tumor outcome, a mechanism for enhanced tumor control in metabolically treated T cells is missing from our model.

## Separate microenvironments for T cell proliferation are key for T cell phenotype preservation

We speculated therefore that T cells could come from outside the tumor enabling founder T cells to reside in supportive microenvironments such as the lymph node (LN), while daughter cells migrate into the tumor. We therefore extended our model by adding another cell type of the dendritic cell to our model. Dendritic cells are critical antigen-presenting cells that take up antigen and stimulate antigen-specific T cells in the LN. T cells then divide and leave the LN for effector function in the target tissue. We built our model system after these principles, leveraging the

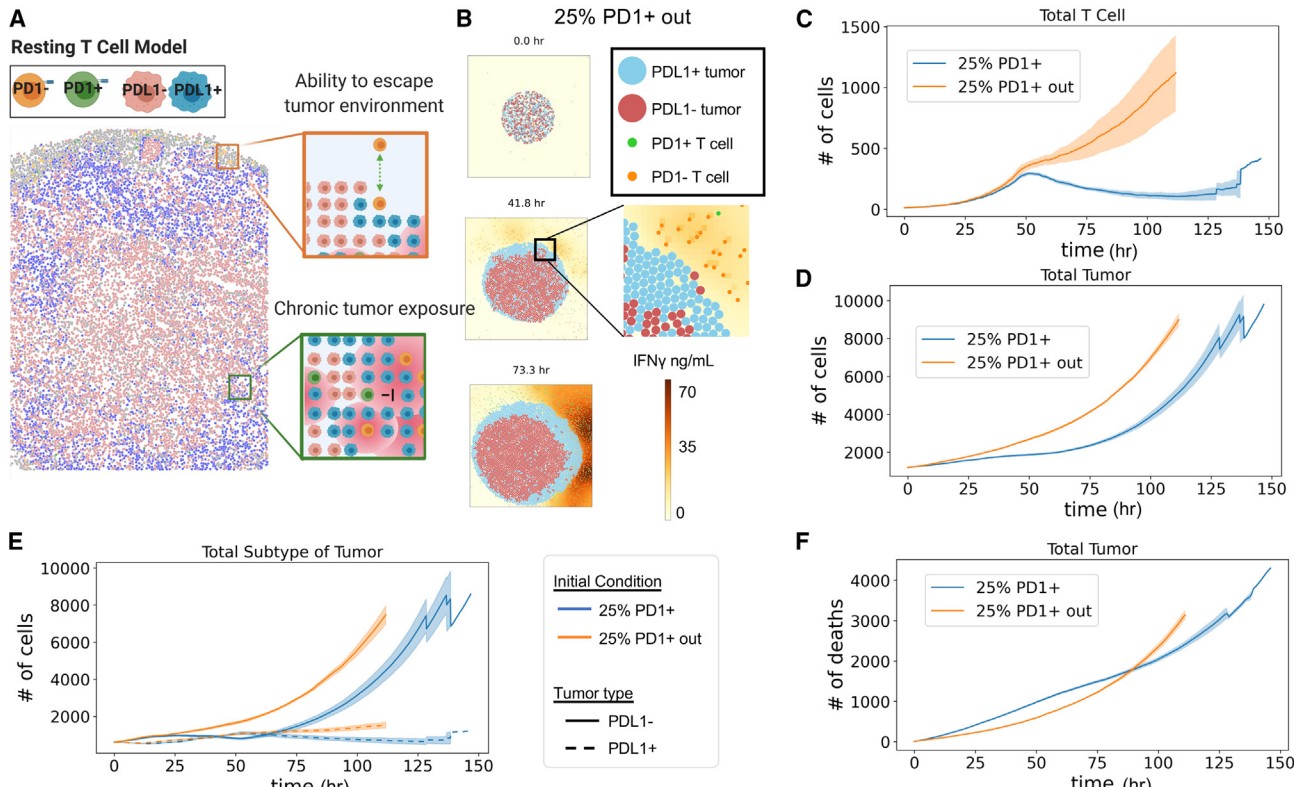

**Figure 6. Conversion of tumor cell phenotype is more critical for tumor control than T cell phenotype preservation**

(A) Theoretical sketch of how T cells are able to escape the tumor microenvironment to promote long-term survival and ability to control the tumor.

(B) Snapshots of the tumor from agent-based modeling of a tumor treated with 25% PD-1$^+$ T cells that were initialized outside the tumor such that they can escape chronic exposure to tumor and limit exhaustion.

(C) Total number of T cells over time of simulation when T cells are initialized outside the tumor bed or inside the tumor.

(D) Total number of tumor cells over time of simulation when T cells are initialized outside the tumor bed or inside the tumor.

(E) Number of tumor cells separated by phenotype over the course of the simulation.

(F) Number of tumor cell deaths over the time course of the simulation.

For (C)–(F), mean of *n* = 4 replicates with shading showing SEM.

same approach we took of literature and lab-derived parameters (Figure 7A). In particular, we encode the ability for dendritic cells to uptake tumor antigen from dying tumor cells within the LN, become activated by apoptotic debris, and migrate to the LN. We also add in our model that some of the transferred antigen-specific T cells move to and stay in the tumor-draining LN. There, they encounter dendritic cells and are activated by MHC-I$^+$ with tumor antigen expressed by these dendritic cells. After this activation, they proliferate before leaving the LN to the tumor microenvironment.

We compared simulations with 25% PD-1$^+$ T cell treatment conditions initialized with and without the extra LN process that incorporated dendritic cells. There were increased and sustained number of T cells over 3 days of simulation time in the conditions where dendritic cells were able to activate T cells in LN (Figure 7B). This came from added numbers of PD-1$^-$ T cells to the tumor from the LN (Figures S9A and S9B). This result more closely matches our CODEX multiplexed imaging results of tumors (Figure 4), where a subset of the T cells within the tumor are found to be PD-1$^-$ even several days after adoptive transfer. Particularly, this reduces the complete switch in pheno-

type from PD-1$^-$ to PD-1$^+$ we observed in previous simulations such as Figure 3H, further suggesting the importance of both preserving T cell phenotype and also multiple waves of non-exhausted T cells in control of tumor growth.

The sustained levels of T cells and percentages PD-1$^-$ T cells within the tumor over long periods of time had a drastic impact on the total number of tumor cells past 36 h of simulated time (Figure 7C). Part of the reason is due to increased T cell killing of tumor cells from conditions with dendritic cells (~1,000 vs. 700 tumor T cell-associated deaths) (Figures S9C and S9D). However, this only accounts for a total difference in 300 tumor cell deaths, whereas we observe a difference in ~900 total tumor cells by 72 h of simulated time (Figure 7C). Most of this difference instead was due to the ability to sustain the conversion of a greater proportion of tumors to the inflamed, non-proliferative PD-L1$^+$ phenotype (Figure 7D). Thus, emphasizing the importance of tumor phenotype conversion as a method of tumor cell containment within the anti-tumor immune response.

While there is a sustained supply of T cells in the tumor microenvironment following incorporation of the LN process (Figure 7B), there is a rapid phenotypic conversion for many T cells

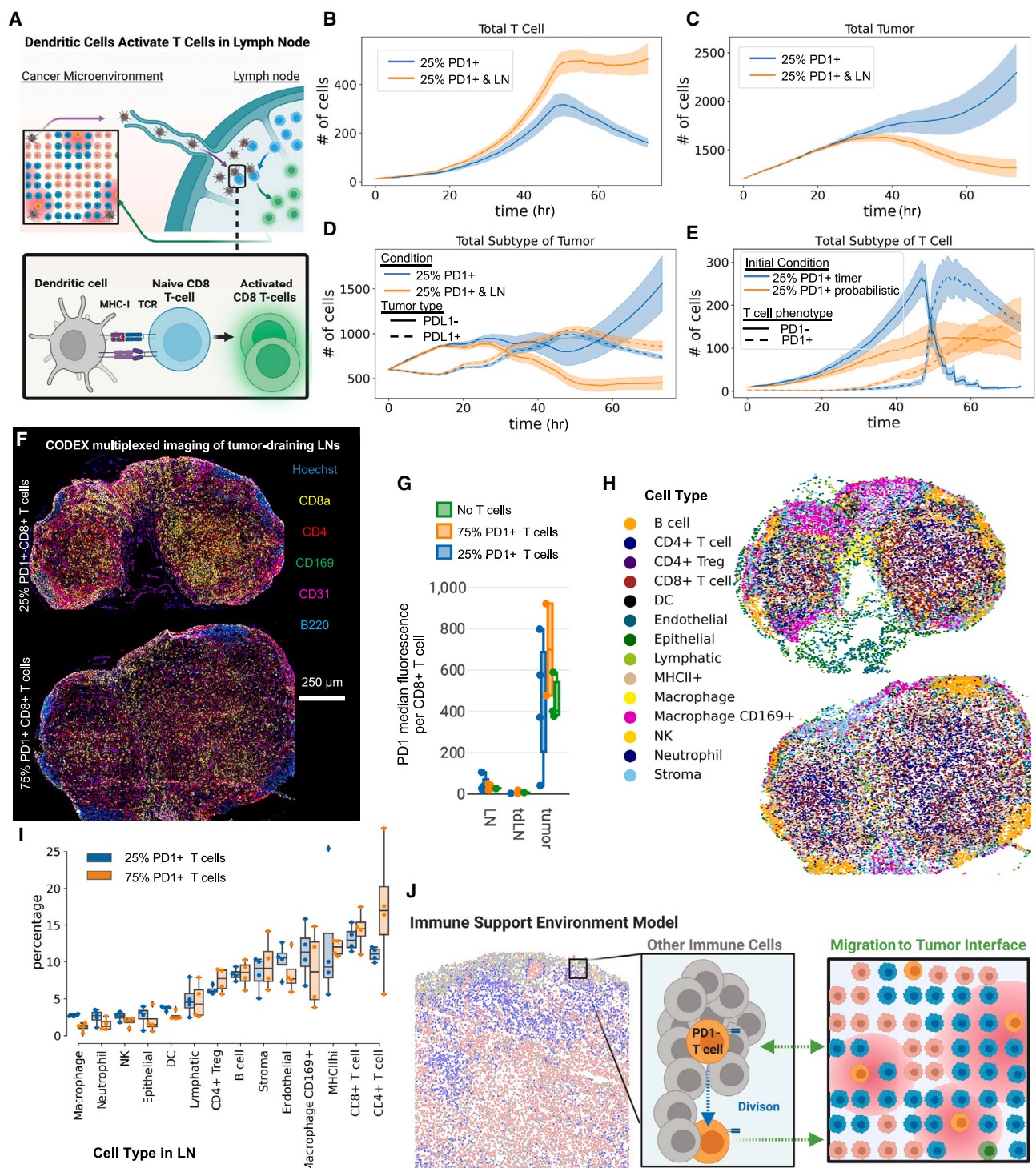

**Figure 7. Lymph nodes are sustained sources of T cells in the tumor microenvironment based on dendritic cell antigen presentation**

(A) Schematic of added components to the model. We add dendritic cells that start in the tumor microenvironment and take up tumor antigen from dying tumor cells and are activated by tumor debris. Lymph nodes (LNs) migrate to the tumor-draining lLN (tdLN) where they can encounter tumor-specific T cells. Upon engagement between MHC-I and TCR of the T cells, T cells proliferate within the LN and leave the LN to the tumor microenvironment.

(B–D) Comparing simulations with added dendritic cells and LN processes vs. simulations without LN and dendritic cells: (B) total number of T cells, (C) total number of tumors, and (D) number of tumor cells separated by phenotype over the time course of the simulation. For (B)–(D), mean of *n* = 8 replicates with shading showing SEM.

*(legend continued on next page)*

around 48 h post-initialization (Figure S9A). Despite uniform treatment and exposure to an identical number of stimulations across all cells, the stark nature of this phenotypic shift appeared to be influenced by the deterministic mechanisms of T cell activation and the fixed refractory periods in the model that we initially established. Upon modifying these timers to stochastic events, we noted a more gradual shift in T cell phenotype within the 25% PD-1$^+$ T-cell-treated condition (Figure 7D). This observation suggests a potential delay in the transition to PD-1$^+$ T cells within the tumor microenvironment. Nonetheless, a decline in PD-1– T cell numbers commenced around the 50-h mark, due to established biological mechanisms where repeated T cell stimulations lead to exhaustion.[60–62] Also, consistent with our previous results, the 25% PD-1$^+$ T cells exhibited sustained efficacy in suppressing tumor growth (Figure S9E), due to their enhanced capacity for early conversion of the tumor phenotype (Figure S9F).

Based on our LN simulations, we hypothesized that we would have greater levels of T cells or greater percentages of PD-1- T cells within tumor-draining LNs of mice treated with 25% PD-1$^+$ T cells based on these results. To test this hypothesis, we harvested LNs from mice with tumors treated with 75% PD-1$^+$ T cells, 25% PD-1$^+$ T cells, or no T cells treated 3 days before. We then created a tissue array of all these LNs and imaged them simultaneously with CODEX multiplexed imaging (Figure 7F). Comparing the median marker expression of PD-1 for CD8$^+$ T cells found in the LN to the CD8$^+$ T cells in the tumor showed increased levels of PD-1 in the tumor (Figure 7G). This agrees with our hypothesis that founder T cells reside in protected LN environments sending daughter cells to tumors. However, we did not observe drastic differences of PD-1 expression between treatment conditions and did not explain the discrepancy between treatments.

To see if the percentage of CD8$^+$ T cells were different between the two conditions, we segmented and clustered cell types in the CODEX LN datasets (Figure 7H). This analysis showed that there were no differences in cell-type percentages within the lymph nodes of both treatments (Figure 7I). Since we did not observe differences between conditions within the LN, perhaps this concept of supportive microenvironments for T cells extends to the tumor (Figure 7J). We propose that T cells actively build microenvironments within the tumor to support T cell phenotype and function. This led us to perform extensive studies evaluating the tumor microenvironment composition surrounding T cells and indicates immune cell supportive microenvironments provide critical support for productive T cell killing.[59]

## DISCUSSION

We developed a scalable agent-based model of T cell therapy of tumors to complement, leverage, and probe CODEX multiplexed imaging datasets of T-cell-treated tumors. Despite the richness of multiplexed imaging data, current analytical techniques are not sufficient to interpret the multidimensional data that are represented by multiple scales (molecule, cell, and tissue) spatially, leaving untapped biology within existing data. By connecting to *Vivarium*, we use biological data to initialize the model, making it possible to connect modules of diverse mechanistic models into integrative simulations that cover multiple spatial and temporal scales.

Adding dynamics to project behavior from static multiplexed imaging datasets, captured at a single time point, is critical. While we were able to compare our model through collecting multiple time points in a mouse model, most often this will not be the case because the majority of collected multiplexed imaging data involve invaluable human biopsies, surgical resections, or donor tissue obtained at a single time point from both healthy and diseased tissues.[15,66] While multiplexed imaging offers insights into cellular interactions across space, it lacks the ability to reveal the temporal evolution of these interactions. This limitation hinders our capacity to make predictions or conduct hypothetical experiments with authentic patient data, which is inherently heterogeneous. Establishing a framework for integrating data and multiscale modeling here serves as a foundational step. It provides a starting point to construct models that deep multiplexed imaging-based biological data, thus extending our understanding of cellular dynamics of tissues. This approach will enable us to comprehend how interactions unfold over time and will offer valuable insights into identifying effective therapies for specific patients.

Using this system, we specifically explored the importance of tumor phenotype on T cell therapy efficacy by incorporating molecular switches of cellular phenotype and function that led to tissue-level phenomena in our data-informed multiscale agent-based model. We then used findings from the simulations to guide design of an *in vivo* experiment and choice of antibodies for use in CODEX imaging to catalog cell types and transitions we saw within our models. This synergistic back-and-forth of model and data across biological scales revealed critical design components of effective T cell therapies.

The results indicated that tumor phenotype considerably influences the ability of T cells to control tumor cell growth through inhibiting proliferation and increasing killing. The importance

(E) Comparison simulations of tumors treated with 25% PD-1$^+$ T cells simulated with either a timer or a probabilistic T cell activation and refractory timing mechanism and showing total number of T cells separated by phenotype over simulation time ($n$ = 5–8 replicates with shading showing SEM). This was done with code on an experimental branch (probabilistic-refractory) of tumor T cell repository.

(F) Representative CODEX images of tdLNs from B16-F10 tumors from mice treated on day 10 with activated PMEL therapeutic T cells (of either 25% PD-1$^+$ T cells, 75% PD-1$^+$ T cells) and harvested 3 days following treatment. Scale bar, 250 mm.

(G) PD1 median fluorescent signal across all CD8$^+$ T cells within tdLNs or LNs or the tumors from mice treated with 25% PD-1$^+$ T cells, 75% PD-1$^+$ T cells, or no T cells ($n$ = 2¬3 replicates with error bars showing SEM).

(H) Representative images of cell types mapped to LN tissues that correspond to figure G images.

(I) Cell type percentages from CODEX multiplexed imaging data of the tdLN of mice treated with either 25% PD-1$^+$ T cells or 75% PD-1$^+$ T cells 3 days post therapy ($n$ = 4 replicates with error bars showing SEM).

(J) Model of T cells supported by immune cells in a microenvironment within the tumor for preservation of phenotype, proliferation, killing, and tumor inhibition locally.

and magnitude of a tumor phenotype change was only clear after analysis of the multiscale model and was not intuitive from the multiplexed imaging data alone. Most recent work has focused on controlling T cell phenotype for both the secretion of killing molecules and self-preservation.[31,32] We found that T cell phenotype also impacts its ability to change tumor phenotype, and a focus on converting tumor phenotype results in greater control than minimizing T cell exhaustion. Based on our findings, T cell therapies should be designed with a profile capable of concurrently inhibiting tumor cell proliferation, enhancing the inflammatory state of tumor phenotype, initiating tumor killing, and sustaining T cell longevity and efficacy. Various methods have previously been employed to modify T cells to achieve each of these objectives.[27,52,67,68] However, what has been lacking is the ability to comprehensively investigate how alterations in T cell phenotype impact all these parameters simultaneously. With a new goal, there will arise a plethora of new strategies to modify T cells that arise to accomplish this objective, but an immediate example next step could be to engineer T cells to secrete cytokines and observe whether they can produce antiproliferative and inflammatory effects while also preserving T cell longevity.

Comparing *in silico* results to CODEX multiplexed imaging reinforced the importance of T cell phenotype and influence on tumor phenotype. Our simulations indicated agreement both in terms of percentages of different cell types and phenotypes and also in the organization of these cells with respect to each other. For example, inflamed tumor cells were found proximal to T cells within both simulation and CODEX imaging results. CODEX data also indicated that T cells were able to change tumor phenotype and minimize T cell exhaustion. Most methods of analysis of T cell phenotype preservation are focused on molecular mechanisms of control since most assays require dissociation of tumors or *ex vivo* manipulation of T cells.[10,11] In contrast, both multiscale modeling and multiplexed imaging preserve spatial features of the data, and our spatial analysis of CODEX and *in silico* experiments demonstrated that the spatial positioning of T cells influenced T cell phenotype. In the comparison with CODEX data, this suggested that our model was missing a mechanism for T cell phenotype preservation. This led us to add dendritic cells and a connection to lymph nodes within our model system, which suggested a sustained source for non-exhausted T cells in the tumor microenvironment. This result indicates that while it is important to design T cells that traffic to the tumor, it may be just as important to also have a subset that will traffic to tumor-draining lymph nodes to be a constant supply of T cells over time or additive effects of continual additions of T cells over time. However, we did not detect differences by CODEX multiplexed imaging in either the phenotype or number of T cells from within the tumor-draining lymph nodes of mice treated with T cell therapies with different phenotypes.

This incongruence motivated parallel research on T cell therapies, where we observed that therapeutic T cell phenotype changes the structure and cellular composition of the tumor microenvironment.[59] We found in this other work that T cells create distinct multicellular neighborhoods based on their phenotype and molecular expression profiles. For example, 2HC-T-cell-treated tumors result in more productive T cell and tumor neighborhoods, whereas T cells not treated with 2HC

secrete more anti-inflammatory cytokines and have T cell and tumor areas also enriched with regulatory neighborhoods. Thus, T cells should be engineered to be agents of structural change of the tumor microenvironment in addition to transforming tumor cell phenotype.

The modularity of the model will enable our group and others to build from this starting point to investigate the effects of various T-cell-based therapies in solid tumors. Integration of models and measurements across biological scales with spatial features preserved will enable decoding of the rules that govern complex networks from biological and clinical samples. Overall, we and others can leverage this model as a template for integrating and using the growing number of spatial datasets, such as CODEX imaging datasets, in other disease settings.[69–71] Integration of multiscale modeling and imaging data will enable better interpretation through leave-one-out experiments and ensemble simulations while simultaneously increasing the complexity and accuracy of agent-based models. Finally, the ability to simultaneously evaluate interactions across scales will guide development of better therapies that interrupt problematic networks or create beneficial ones.[66]

### Limitations of the study

In future work, driven by molecular data acquired through multiplexed tissue imaging, the model should be expanded to include other cell types and anti-inflammatory molecules. Since the *Vivarium* framework for multiscale modeling is modular and compositional, it will be straightforward to add additional cell types with different representations of internal mechanisms and environmental interactions, such as how we added the dendritic cell and LN process. Not straightforward are the selection of molecular features and phenotypes that will increase the accuracy of the model under a broader range of conditions. Although we have some clues about missing components from our CODEX and RNA dataset,[59] additional data collection will be needed. For example, single-cell RNA sequencing of the cell types within the tumor will shed light on key molecular and cellular interactions. Building this complexity within *in silico* models will be necessary because as the number of intercellular connections are increased, it will become more difficult to recapitulate and deconvolute these networks within *in vitro* systems. Similarly, we focused on one *in vivo* tumor model system to understand relationship between tumors and antigen-specific T cells—in particular one that mimics tumor that lose or downregulate MHC-I expression.[72,73] This one model does not recapitulate the diversity of human tumors, and consequently, additional tumor models and human tumor samples will need to be studied and integrated to build future branches of the model.

### STAR★METHODS

Detailed methods are provided in the online version of this paper and include the following:

- KEY RESOURCES TABLE
- RESOURCE AVAILABILITY
  - Lead contact
  - Materials availability
  - Data and code availability

- EXPERIMENTAL MODEL AND SUBJECT DETAILS
  - Mice
- METHOD DETAILS
  - Model Development
  - T Cell Culture and Stimulation
  - IFNγ *ELISA*
  - *In Vitro* T-cell Killing Assay
  - CyTOF Phenotyping
  - Flow Cytometry for Intratumoral T Cell Measurement
  - CODEX multiplexed imaging of tumor-draining lymph nodes
- QUANTIFICATION AND STATISTICAL ANALYSIS
  - CODEX multiplexed imaging data analysis
  - Neighborhood Analysis of Simulation Output

**SUPPLEMENTAL INFORMATION**

**ACKNOWLEDGMENTS**

This work was supported by the U.S. National Institutes of Health (2U19AI057229-16, 5P01HL10879707, 5R01GM10983604, 5R33CA18365403, 5U01AI101984-07, 5UH2AR06767604, 5R01CA19665703, 5U54CA20997103, 5F99CA212231-02, 1F32CA233203-01, 5U01AI140498-02, 1U54HG010426-01, 5U19AI100627-07, 1R01HL120724-01A1, R33CA183692, R01HL128173-04, 5P01AI131374-02, 5UG3DK114937-02, 1U19AI135976-01, IDIQ17X149, 1U2CCA233238-01, and 1U2CCA233195-01); the U.S. Department of Defense (W81XWH-14-1-0180 and W81XWH-12-1-0591); the U.S. Food and Drug Administration (HHSF223201610018C and DSTL/AGR/00980/01); Cancer Research UK (C27165/A29073); the Bill and Melinda Gates Foundation (OPP1113682); the Cancer Research Institute; Hope Realized Medical Foundation (209477); the Silicon Valley Community Foundation (2017-175329 and 2017-177799-5022); and the Rachford & Carlotta A. Harris Endowed Chair to G.P.N. J.W.H. was supported by an NIH T32 Fellowship (T32CA196585) and an American Cancer Society - Roaring Fork Valley Postdoctoral Fellowship (PF-20-032-01-CSM). E.A. was supported by an NIH F32 Fellowship from NIGMS (F32GM137464) and by the Paul G. Allen Frontiers Group via the Allen Discovery Center at Stanford. The authors thank Ryan Spangler for helping set up the simulation development environment. Some figures were created with BioRender.com.

**AUTHOR CONTRIBUTIONS**

J.W.H. and E.A. conceived and developed the agent-based model, analyzed data, and wrote the manuscript. J.W.H. and N.H. completed *in vitro* and *in vivo* based experiments with T cells. M.L. researched parameters for the model system. T.-K.T. ran metabolic CyTOF tumor studies. J.B.S., M.W.C., and G.P.N. supervised the project, provided resources and feedback, and helped in writing the manuscript.

**DECLARATION OF INTERESTS**

G.P.N. has equity in and is a scientific advisory board member of Akoya Biosciences, Inc.

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

# Cell Systems
## Article

**CellPress**

## STAR★METHODS

### KEY RESOURCES TABLE

| REAGENT or RESOURCE | SOURCE | IDENTIFIER |
|---|---|---|
| **Antibodies** | | |
| Custom-conjugated CyTOF antibodies | N/A | All information on clones, companies, |
| Custom-conjugated murine CODEX antibodies | | RRIDs, etc. are included in Table S2. |
| anti-CD3 (clone 145-2C11) | Bioxcell | Catalog #BE0001-1; RRID: AB_1817016 |
| anti-CD28 (clone 37.51) | Bioxcell | Catalog #BE0015-1; RRID: AB_1817016 |
| TruStain FcX™ (anti-mouse CD16/32) Antibody | Biolegend | Catalog #101319; RRID: AB_2783137 |
| **Chemicals, peptides, and recombinant proteins** | | |
| Potassium hydroxycitrate tribasic monohydrate | Sigma-Aldrich | 59847-1G |
| gp100 - KVPRNQDWL | AnaSpec | AS-62589 |
| Recombinant Human IL-2 (carrier-free) | Biolegend | 589106 |
| Protein Transport Inhibitor (Containing Monensin) | BD Biosciences | 554724 |
| Protein Transport Inhibitor (Containing Brefeldin A) | BD Biosciences | 555029 |
| Recombinant Mouse IFN-$\gamma$ (carrier-free) | Biolegend | 575302 |
| cis-Diammineplatinum(II) dichloride (cisplatin) | Sigma-Aldrich | P4394-25MG |
| EMS 16% Paraformaldehyde aqueous | Fisher | 50-980-487 |
| **Critical commercial assays** | | |
| xCELLigence Real-Time Cell Analysis | Agilent | Single Plate |
| **Experimental models: Cell lines** | | |
| B16-F10 tumor cell line | ATCC | RRID:CVCL_0159 |
| **Experimental models: Organisms/strains** | | |
| B6 - C57BL/6J | Jackson Laboratories | RRID:IMSR_JAX:000664 |
| PMEL - B6.Cg-Thy1a/Cy Tg(TcraTcrb)8Rest/J | Jackson Laboratories | RRID:IMSR_JAX:005023 |
| **Software and algorithms** | | |
| tumor-tcell | 10.5281/zenodo.10779282 | https://github.com/vivarium-collective/tumor-tcell |
| CODEX Processor | https://github.com/nolanlab/CODEX | N/A |
| Segmenter | https://michaellee1.github.io/CellSegSite/index.html | N/A |
| Neighborhood analysis | https://github.com/nolanlab/NeighborhoodCoordination | N/A |
| ImageJ | https://imagej.net/software/fiji/ | N/A |
| Scanpy | https://scanpy.readthedocs.io/en/stable/ | N/A |
| **Other** | | |
| Fisherbrand™ Superfrost™ Plus Microscope Slides | Thermo Fisher Scientific | 12-550-15 |

## RESOURCE AVAILABILITY

### Lead contact
Further information and requests for resources and reagents should be directed to and will be fulfilled by the lead contact, Garry Nolan (gnolan@stanford.edu).

### Materials availability
This study did not generate new materials.

**CellPress**

**Cell Systems**
Article

### Data and code availability

- **Data**: Data from the CODEX experiments and initializations can also be found within the repository folder *data*. All other data reported in this paper will be shared by the lead contact upon request.
- **Code**: All original code has been deposited at https://github.com/vivarium-collective/tumor-tcell and is publicly available as of the date of publication. The code was released with pypi with version number 1.0.0 for the version associated with this paper https://pypi.org/project/tumor-tcell/1.0.0/. This version of the github code is https://zenodo.org/records/10779283 and is listed in the key resources table. The README file documents how this can be used as a Python library or cloning the repository locally.
- Any additional information required to reanalyze the data reported in this paper is available from the lead contact upon request.

## EXPERIMENTAL MODEL AND SUBJECT DETAILS

### Mice

B6 and PMEL transgenic mice were maintained per guidelines approved by Stanford University's Institutional Review Board. C57BL/6J and PMEL mice were purchased from Jackson
   Laboratories.

## METHOD DETAILS

### Model Development
#### *The* Vivarium *framework*

*Vivarium* is an open-source software tool for multi-scale modeling. The aim was to make it easier for scientists to define any imaginable mechanistic model, combine it with existing models, and execute them together as an integrated simulation. It provides an interface that makes individual simulation tools into modules that can be wired together, parallelized across multiple CPUs, and simulated across many spatial and temporal scales.[25]

   *Vivarium's* basic elements are processes and stores (Figure 2C). A *Vivarium process* is an object that contains parameters and the update function, which describes the inter-dependencies between the variables and how they map from one time to the next. A *store* is an object that holds the system's state variables and applies the processes' updates. Processes include *ports*, which allow users to wire processes together through variables in shared stores with connections called a *topology*. Multiple processes can be wired together as integrated models called *composites*. These models are implemented in a nested hierarchy, which has stores within stores to allow an environmental model to run at the top of the hierarchy, with individual agents running in parallel within the model.

#### *Cell Processes*

The model is composed of two major cell types (T cells and tumor cells), each with two separate phenotypes. Each cell type has an associated *Vivarium* process that represents the mechanisms that make a cell switch between phenotypes. These processes define fundamental rules that govern cellular interactions with the other cell types and with the inputs it receives from the environment. The tumor process is focused on two phenotypic states: proliferative with low levels of immune molecules (MHC-I and PD-L1) and quiescent with high levels of immune molecules (MHC-I and PD-L1). Its transition from the proliferative state is dependent on the level of IFN$\gamma$ secreted by T cells. Both tumor types can be killed by receiving cytotoxic packets from the T cells. The T cell process is focused on two phenotypic states. The PD-1$^-$ T cells secrete larger amounts of immune molecules (IFN$\gamma$ and cytotoxic packets) than PD-1$^+$ T cells. These immune molecules impact the state and death of tumor cells. The transition from the PD-1$^-$ state to the PD-1$^+$ state is dependent on the length of time the T cell is engaged with tumor cells. Each process was tested individually to meet expected outcomes based on literature or lab data. Testing the processes individually reveals whether underlying parameters derived from literature values or primary data accurately represent behavior expected based on such research.

#### *Cell Composites*

The T cell and tumor processes are combined with additional processes to create T cell and tumor composite agents. These include a division process, which waits for division to be triggered and then carries out division; a death process, which waits for death to be triggered and then removes the agent; and a local field process, which interfaces the external environment to support uptake and secretion for each agent. Testing individual composite cells adds additional complexity and is another accuracy check of the model.

#### *Tumor Microenvironment*

The Tumor Microenvironment is a composite model that simulates a 2D environment with agents that can move around in space, exchange molecules with their neighbors, and exchange molecules with a molecular field. A *neighbors* process models individual agents as circular rigid bodies that can move, grow, and collide. This process tracks the locations of individual agents and handles the exchanges between neighboring cells. A *diffusion* process operates on the molecular fields of IFN$\gamma$, and handles the cells uptake and secretion from the environment.

#### *Connecting Cell Composites in the Tumor Microenvironment*

After validating all individual processes, we connected processes and composites and endowed individual elements with additional behaviors like migration. In our model, the T cells can interact with tumor cells through 1) the TCR on T cells and MHC-I molecules on tumor cells to activate T cells, induce IFN$\gamma$ and cytotoxic packet secretion, and inhibit T-cell migration, 2) PD-1 receptor on T cells and

PD-L1 receptor on tumor cells that can inhibit T-cell activation and induce apoptosis, and 3) indirectly through secretion of IFN$\gamma$ by T cells, which is taken up by tumor cells to cause a state switch to upregulate MHC-I and PD-L1 and decrease proliferation.

### Initialization of Experiments

The number of T cells and tumor cells as well as the proportions of phenotypes for each were based on experimental data. Briefly, most of the experiments started off with a total of 1200 tumor cells and 12 CD8+ T cells. For simulations in Figures 2 and 3, T cells were initialized randomly in the tumor bed. For simulations in Figure 5, T cells were initialized based on the locations of CODEX multiplexed imaging data. For simulations in Figure 6, T cells were located randomly inside or outside the tumor bed as specified.

For additional information on model development see our documented code base and README at https://github.com/vivarium-collective/tumor-tcell.

### T Cell Culture and Stimulation

#### Immune cell isolation

Murine cells were obtained from adult mouse lymph nodes and spleens. Obtained cells were treated with ACK lysis buffer to lyse red blood cells, and lysates were filtered through cell strainers to isolate splenocytes.

#### T cell media

Supplemented media was made with RPMI 1640 media with glutamine, 1x non-essential amino acids, 1 mM sodium pyruvate, 0.4x vitamin solution, 92 $\mu$M 2-mercaptoethanol, 10 $\mu$M ciprofloxacin, and 10% fetal bovine serum.

#### CD3-coated plate preparation

To each well of a 96-well, U-bottomed plate was added 50 $\mu$L of a solution of 5 $\mu$g/mL anti-CD3 (Bioxcell, clone 145-2C11) in PBS. After incubation at 4 °C overnight, liquid was decanted.

#### T-cell stimulation

Isolated murine immune cells were stimulated by incubation with 1 $\mu$M cognate peptide GP100 (KVPRNQDWL) and 50 IU/mL IL-2. For 2-hyroxycitrate (2HC) conditions, 2HC was added to culture media at a concentration of 5 mM. Cells were seeded at a density of 2-5×10$^6$ cells/mL. Cells were fed with additional IL-2 in T-cell media every other day. On day 5, cells were added to CD3-coated plates in culture media containing 2 $\mu$g/mL anti-CD28 (Bioxcell, clone 37.51). On day 8 cells were removed from plates and plated on uncoated plates and fed with IL-2-containing media until day 10.

### IFN$\gamma$ ELISA

T cells were stimulated in the presence of 2HC per the above T-cell stimulation protocol. On day 10 the T cells were removed from culture plates, spun down and refreshed with new T-cell media. T cells were then placed on 96-well, U-bottomed CD3-coated plates in culture media containing 2 $\mu$g/mL anti-CD28 (Bioxcell, clone 37.51) for 24 hrs. A total of 36 wells, each with 100 $\mu$L of media was sampled by taking 5 $\mu$L of media per well and centrifuged to remove any cellular debris. 150 $\mu$L of this media was then used within a LEGEND MAX Mouse IFN$\gamma$ ELISA Kit as specified per manufacturer's instructions (Biolegend, #430807).

### In Vitro T-cell Killing Assay

T-cell killing and tumor cell growth rates were determined using the xCELLigence Real-Time Cell Analysis platform. Wells of xCELLigence E-plates were coated with gold nanoparticles and electrical potential was passed across the plate every 15 min. Monitoring of changes in the electrical impedance enabled quantification of adherent cells over time. For these assays, T cells were expanded. B16-F10 melanoma cells were split and then left as control cells or treated with 10 ng/mL IFN$\gamma$ for 24 hours prior to plating. After the pretreatment, 10,000 B16-F10 cells were plated in each well of the xCELLigence E-plate and allowed to adhere for 12 h. Next, T cells were added at 1:1 effector to target ratio. The growth of tumor cells and killing by T cells was monitored for up to 24 h. Killing was calculated by normalizing the cell index of each well to the time point just before addition of the T cells and then quantifying the differences between T cell and control wells without T cells over time.

### CyTOF Phenotyping

#### Heavy metal conjugation of Antibodies

Primary antibody transition metal-conjugates were prepared in-house using 100-$\mu$g antibody lots and the MaxPAR antibody conjugation kit (DVS Sciences) according to the manufacturer's recommended protocol. Following conjugation, antibodies were diluted in Candor PBS Antibody Stabilization solution and stored at 4 °C.

#### Tumor Metabolic CyTOF Staining

Following incubation with IFN$\gamma$, 2×10$^6$ B16-F10 tumor cells were pulsed with IdU for DNA labeling by incubating with 100uM of IdU in Ham's F-12 medium (LifeTechnologies), 1% ITSX, 10 mM HEPES, 1% P/S/G, 0.1% polyvinyl alcohol (PVA) at 37°C with 5% CO$_2$ for 30 mins. After 30 mins, IdU was quenched and diluted by adding PBS. Cells were spun down at 300 x g, RT, 5 mins. Cells were fixed with 1.6% PFA (Electron Microscopy Sciences) at RT for 10 mins. After fixation, fixative was quenched by adding CSM (CSM; PBS with 0.5% bovine serum albumin and 0.02% sodium azide) and cells were spun down at 300 x g, RT, 5 mins. Cells were blocked with FcBlock (0.25 $\mu$g/1×10$^6$ cells) for 15 min at RT. Cell surface antibody master mix in CSM was filtered through a pre-wetted 0.1-$\mu$m spin-column (Millipore) to remove antibody aggregates and added to the samples. After incubation for 30 min at RT, cells were washed once with CSM. To enable intracellular staining, cells were permeabilized by incubating with ice-cold MeOH for 10 min on ice and washed two times with CSM to remove any residual MeOH. Intracellular antibody master mix in CSM was added to

the samples and incubated for 1 hour at RT. Cells were washed once with CSM and resuspended in intercalation solution (1.6% PFA in PBS and 0.5 μM rhodium intercalator (Fluidigm)) for 20 min at RT or overnight at 4°C. Before acquisition, samples were washed once in CSM and twice in ddH2O and filtered through a cell strainer (Falcon). Cells were then resuspended at $1 \times 10^6$ cells per ml in ddH2O supplemented with 1× EQ Four Element Calibration Beads (Fluidigm) and acquired on a CyTOF2 mass cytometer (Fluidigm).

### Tumor and T cell Phenotype CyTOF Staining

Following T-cell stimulation on day 10, $2 \times 10^6$ cells were stained with cisplatin at 25 μM in PBS in 1 mL for 1 min at 4°C, quenched with 1 mL of fetal bovine serum, and washed with cell staining medium (CSM; PBS with 0.5% bovine serum albumin and 0.02% sodium azide). Cells were blocked with FcBlock (0.25 μg/1×10⁶ cells) for 15 min at room temperature, then the surface antibody cocktail was added and incubated 1 h at room temperature on a shaker at 100 rpm. Cells were washed with CSM and then with PBS. Cells were fixed and stained with intercalators overnight at 4°C in a solution of 1.6% PFA in PBS. The next day, the cells were washed once with CSM and twice with doubly distilled water, resuspended in doubly distilled water, and analyzed using CyTOF.

### Flow Cytometry for Intratumoral T Cell Measurement

#### In vivo tumor model

On day 0, B6 mice were injected with $2 \times 10^5$ B16-F10 melanoma tumor cells. On day 0, immune cells were isolated from a PMEL mouse and stimulated as described above for 10 days to produce stimulated T cells for adoptive transfer. On day 9, mice were given a central dose of 500 cGy, which induces transient lymphopenia.[74] On day 10, T cells cultured *ex vivo* were harvested and adoptively transferred intravenously in volumes of 100 μL with $1 \times 10^6$ stimulated T cells per mouse. Tumors were harvested 3 days after treatment with stimulated T cells.

#### Flow cytometry staining

Harvested tumors were dissociated by maceration over a sterile 70-μm cell strainer with frequent washes of PBS. Cells were then stained with a mixture of a 1:100 PBS solution of APC-conjugated rat anti-mouse CD8a, clone 53–6.7 (BD Pharmingen), PerCP-conjugated rat anti-mouse CD45, clone 30-F11 (Biolegend), and 1:1000 of LIVE/DEAD Fixable Violet Dead Cell Stain (ThermoFisher) for 15 min at 4°C. Cells were then washed with FACS wash buffer and analyzed using a BD FACSCalibur flow cytometer with CellEngine gating for live cells.

### CODEX multiplexed imaging of tumor-draining lymph nodes

#### Tumor model

On day 0, B6 mice were injected with $2 \times 10^5$ B16-F10 melanoma tumor cells. On day 0, immune cells were isolated from a PMEL mouse and stimulated as described above with or without 2HC for 10 days to produce stimulated T cells for adoptive transfer. On day 9, mice were given a central dose of 500 cGy, which induces transient lymphopenia.[67] On day 10, T cells cultured *ex vivo* were harvested and adoptively transferred intravenously in volumes of 100 μL with $1 \times 10^6$ of conventionally stimulated or 2HC-activated T cells per mouse. Tumor draining lymph nodes were harvested 3 days after adoptive transfer.

#### Array creation

Imaging data was collected from multiple mice from multiple experiments. We included all tdLNs into two arrays, which were subsequently cut onto the same coverslip. Arrays were constructed on the cryostat and sectioned at a width of 7 μm.

#### CODEX antibody conjugation and panel creation

CODEX multiplexed imaging was executed according to the CODEX staining and imaging protocol.[29] CODEX imaging involves iteratively annealing and stripping of fluorophore-labeled oligonucleotide barcodes complimentary to the barcodes attached to 40+ antibodies used to stain the tissue. Antibody panels were chosen to include targets that identify subtypes of tumor, stromal, innate, and adaptive immune cells. Detailed panel information can be found in Table S2. Each antibody was conjugated to a unique oligonucleotide barcode, after which the tissues were stained with the antibody-oligonucleotide conjugates. We validated that staining patterns matched patterns observed by immunohistochemical analysis within positive control tissues of tumor or mouse spleen. Antibody-oligonucleotide conjugates were first tested and titrated in low-plex fluorescence assays, and signal-to-noise ratio was evaluated, then antibody-oligonucleotide conjugates were tested together in a single CODEX multicycle. Signal-to-noise ratio was again evaluated, and the optimal dilution, exposure time, and appropriate imagine cycle was determined for each conjugate.

#### CODEX multiplexed imaging

The tissue arrays were stained with the validated panels of CODEX antibodies and imaged.[29] Briefly, this entailed cyclic stripping, annealing, and imaging of fluorescently labeled oligonucleotides complementary to the oligonucleotide conjugated to the antibody.

## QUANTIFICATION AND STATISTICAL ANALYSIS

### CODEX multiplexed imaging data analysis

#### CODEX data processing

Raw imaging data were processed using the CODEX Uploader for image stitching, drift compensation, deconvolution, and cycle concatenation. CODEX enables single-cell resolution protein quantification that can be used for evaluating cell type definition, state, and location. To obtain quantitative single cell information, we processed the multiplexed imaging data, segmented individual cells, and extracted single-cell protein expression. Processed data were then segmented using the CellVisionSegmenter, a neural network

R-CNN-based single-cell segmentation algorithm.[68] Both the CODEX Uploader and Segmenter software can be downloaded from our GitHub site (https://github.com/nolanlab/CODEX), and the CellVisionSegmenter software can be downloaded at https://github.com/michaellee1/CellSeg. After the upload, images were evaluated for specific signal: Any markers that produced an untenable pattern or a low signal-to-noise ratio were excluded from the ensuing analysis. Uploaded images were visualized in ImageJ (https://imagej.nih.gov/ij/).

### Cell-type analysis

Single cells were identified and classified into cell types and states based on marker expression for murine studies. Cell type identification were done following the strategies we have developed.[29,53] Briefly, nucleated cells were selected by gating DRAQ5, Hoechst double-positive cells, followed by z-normalization of protein markers used for clustering (some phenotypic markers were not used in the unsupervised clustering). The data were overclustered with Leiden-based clustering with the scanpy Python package. Clusters were assigned a cell type based on average cluster protein expression and location within image. Impure clusters were split or reclustered following mapping back to original fluorescent images.

### Neighborhood Analysis of Simulation Output

Neighborhood analysis was performed as described previously[17] on simulation output for day 3 after treatment of tumors with T cells of different phenotype compositions. Briefly a window size of 10 nearest neighbors was taken across the tissue cell type maps clustered into 5 neighborhoods. These clusters were mapped back to the tissue and evaluated for cell type enrichments to determine overall structure.

