## [Document S2. Transparent peer review records for Hickey et al · Cell Systems]

Integrating Multiplexed Imaging and Multiscale Modeling Identifies Tumor Phenotype Conversion as a Critical Component of Therapeutic T Cell Efficacy

John W. Hickey, Eran Agmon, Nina Horowitz, Tze Kai Tan, Matthew Lamore, John B. Sunwoo, Markus Covert, and Garry P. Nolan

Summary

Initial Submission: Received Dec 14, 2022
Preprint: doi.org/10.1101/2023.12.06.570168
Scientific editor: Suzanne de Bruijn, Ph.D & Bernadett Gaal, DPhil

First round of review: Number of reviewers: 3
3 confidential, 0 signed
Revision invited March 7, 2023
Major changes anticipated
Revision received Nov 1, 2023

Second round of review: Number of reviewers: 3
3 original, 0 new
3 confidential, 0 signed
Accepted March 19, 2024

This Transparent Peer Review Record is not systematically proofread, type-set, or edited. Special characters, formatting, and equations may fail to render properly. Standard procedural text within the editor's letters has been deleted for the sake of brevity, but all official correspondence specific to the manuscript has been preserved.

Editorial decision letter with reviewers' comments, first round of review

Dear Dr. Nolan,

I'm enclosing the comments that reviewers made on your paper, which I hope you will find useful and constructive. As you'll see, they express interest in the study, but they also have a number of criticisms and suggestions. Based on these comments, it seems premature to proceed with the paper in its current form; however, if it's possible to address the concerns raised with additional experiments and/or analysis, we'd be interested in considering a revised version of the manuscript.

As a matter of principle, I usually only invite a revision when I'm reasonably certain that the authors' work will align with the reviewers' concerns and produce a publishable manuscript. In the case of this manuscript, the reviewers and I have 3 make-or-break concerns.

1. Currently, it is not clear whether the conclusions are supported by the data and the analysis. This is in large part due to lack of clarity and detail about the model and the assumptions, and lack of clear discussion of support for some of the key assumptions. This is essential to fix for the Reviewers and readers to be able to fully evaluate the work and we ask that you pay special attention to this aspect of the revisions.
2. Relatedly, Reviewer 1 requests that you validate the model prior to applying it to derive new insight, which we feel would increase confidence in the conclusions.
3. Reviewer 1 (points 2 & 3) and Reviewer 2 (points 2 & 5) suggest ways in which the model could (and perhaps should) be improved. Please consider these points carefully, extend the model if the extensions improve it with respect to the goals of the work presented in this manuscript, and if you find that it does not, please provide a convincing explanation.

I would also request you to include a copy of the related Hickey et al. manuscript in the resubmission as a supplemental 'related manuscript' item, as there are extensive references to the data in this manuscript (the manuscript described as 'cosubmitted').

In addition to the concerns I've detailed above, I've highlighted portions of the reviews that strike me as particularly critical. I'd also like to be explicitly clear about an almost philosophical stance that we take at Cell Systems...

We believe that understanding how approaches fail is fundamentally interesting: it provides critical insight into understanding how they work. We also believe that all approaches do fail and that it's unreasonable, even misleading, to expect otherwise. Accordingly, when papers are transparent and forthright about the limitations and crucial contingencies of their approaches, we consider that to be a great strength, not a weakness. Please keep this in mind when addressing the Reviewers' comments about how the model may be extended and improved, and their requests for further validation of the model and the new insights derived from it.

We believe that the figures are the scientific backbone of the paper. Currently, it's not possible to understand the manuscript's conceptual advance from figures presented. Similarly, it's not possible to understand where your approach gets its analytical power. These things need to be demonstrated with data and analysis, in the form of figures with their legends or mathematical argumentation, and then supported with explanatory text. Simply stating them as facts is not sufficient. Please keep this

in mind when addressing the concern of reviewer 2 and 3 that the modelling is not described clearly, and the suggestion of reviewer 2 to present the how the model is derived, including all details of the input data, in a flowchart.

As you address these concerns, it's important that you and I stay on the same page. I'm always happy to talk, either over email or by phone, if you'd like feedback about whether your efforts are moving the manuscript in a productive direction. Do note that we generally consider papers through only one major round of revision, so the revised manuscript would be either accepted or rejected based on the next round of comments we receive from the reviewers. If you have any questions or concerns, please let me know. More technical information and advice about resubmission can be found below my signature. Please read it carefully, as it can save substantial time and effort later.

STAR PROTOCOLS

Complement your primary research article by publishing a step-by-step procedure with STAR Protocols, an open-access peer-reviewed journal from Cell Press. STAR Protocols aims to make the daily work of the scientific researcher easier by providing complete, authoritative, and consistent instructions on how to conduct experiments. The primary criteria for publication in STAR Protocols is usability and reproducibility. You can check out their most recent protocols here. If you have any questions, please email starprotocols@cell.com.

I look forward to seeing your revised manuscript.

All the best,

Suzanne
Suzanne de Bruijn, Ph.D.
Scientific Editor, Cell Systems

Reviewers' comments:

Reviewer #1: In this manuscript, the authors integrated CODEX multiplexed tissue imaging with multiscale modeling software, to model key factors that influence T cell therapy efficacy and further understand the interactions of T cell therapies with cancer at multiple scales. Utilizing the Vivarium, an integrated model that covered multiple spatial and temporal scales, the authors first demonstrated that both the phenotype conversion of T cells and tumor cells enhanced T cell recognition and killing.

An appreciated innovation of this work is to use the CODEX multiplexed imaging derived spatial context of tumors as initial states of multiscale simulations to study the spatial localization and function of T cells. They demonstrated the synergy of employing both the top-down and the bottom-up approaches, in other words, the deconstruction and reconstruction of the interaction networks. By doing so, they concluded that the conversion of tumor cell phenotype is more critical for tumor control than T cell phenotype preservation, which suggested potential design criteria and patient selection metrics for T cell therapies. In practice, they provide both a conceptual and a practical paradigm,

which as exemplified in this manuscript, is fueled by several rounds of experiment-derived theory, theory-guided simulation, and simulation-inspired experiment.

However, I have major concerns about the methodology applied and some of the conclusions it led to. Briefly, the multiscale simulations in this study are insufficient to reflect both the tumor microenvironment and the process changes during T-cell therapies. In addition, given that some key cellular components such as CAF, which have been recognized to influence the infiltration and function of T cells, have been ignored in this work, some biological conclusions in this manuscript should be reconsidered.

Below are listed major concerns:

1. It is necessary to prove the reliability of the multiscale agent-based model before using it to explain specific biological processes rather than using it to formulate mechanistic explanations directly. Biological processes that are proven can be used as standards to evaluate the model. It is commendable that the authors strive to provide literature-supported and lab-derived parameters to create the model. Additionally, it is necessary to include information about specific contexts and conditions in which these values were chosen, such as the specific antigen, or chronic or acute infection. In addition, some of the sources should be rechecked. I randomly check the sources of two parameters, "PD1n_IFNg_production" (1.62e4 molecules/cell/s) and "PD1p_IFNg_production" (1.62e3 molecules/cell/s). One denoted source, "Bouchnita et al., 2017" provides "the secretion rate of type I IFN by single activated APC (plasmacytoid dendritic cell): 1.6×10^4 molec/hr" in its appendix without mentioning IFNg, which belongs to type II IFN. In the other source, "Zelinskyy et al., 2005" only measured perforin, granzyme B, and b-actin of activated CD8+ T cells.
2. It was too simplified to only include T cells and tumor cells in the model as the tumor micro-environment was very complicated. In the section T cells induce tumor cell phenotype conversion in vivo, cell-type percentages were based on only T cells and tumor cells in in silico model but were based on all cells in CODEX multiplexed imaging data. Moreover, there were only three points in figure 4F, figure 4G, and figure 4H; more repetitions are needed, like selecting CODEX data of different regions and different time points except for three days.
3. It is not rational to use a proportion of PD1+ T cells as a substitution for effector and memory T cells. There are many other different characteristics between effector T cells and memory T cells, like proliferating capacity. More parameters should be built into the model.
4. This model was more suitable to raise hypotheses and provide ideas for experimental validations. More experiments are needed to confirm the results.

Below are listed minor concerns:

1. Probability, the supportive micro-environments for T cells extending to the tumor in Figure 6I can be correlated with tertiary lymphoid structures.
2. It was not clear about the spatial distribution of T cells and tumor cells in the first and second parts of the results. Whether cells were randomly distributed or distributed according to certain rules was not clear.

Reviewer #2: This manuscript presents the results of studies in which data from in vitro and in vivo studies were used in an iterative process to generate an in silico model for T cell/tumor cell interactions. There were, however, some points that need to be further addressed.

1. It was difficult to follow the way in which the model was derived, and rather than including Fig. 1 which was somewhat inscrutable it would be useful to present a flowchart of detailing precisely what data was fed into the model from the in vivo and in vitro experiments.
2. The in silico model appears to be derived from a single in vivo time point along with 1 in vitro study of the time course of tumor cell cytotoxicity. It would seem that additional in vitro and in vivo data collected over an extended time course would result in the generation of a better model. If not, please explain the reasoning behind the decision to base the model on a more limited set of experimental data. In addition, it is not clear how IFN-g expression data was integrated into the model without any experimental measurements of expression of this cytokine.
3. Expression of class I on B16 is nearly undetectable in the absence of IFN-g but additional studies have demonstrated that extended culture will lead to recognition by PMEL class I-restricted T cells can be boot-strapped and thus the time course in Fig. 2B should have been extended beyond 14 hours. This also emphasizes what is somewhat unique about the B16 model, as the majority of human tumors, with the exception of B2M knock-out tumors, do not demonstrate such a severe deficiency of MHC expression. As such, it would be useful to model these interactions with a murine tumors that are not deficient for class I expression.
4. The results presented in Fig. 5 appear to indicate some control of tumor cell growth using 2-HC-treated T cells whereas the results presented in Fig. 6 do not demonstrate tumor control when T cells are initiated outside of the tumor. This may reflect the need for additional depots of T cells in local lymph nodes that can be drawn upon to mediate effective control, but it is still unclear why the model should perform worse when T cells are initiated outside of the tumor if the discrepancies between the data and the model are proposed to be resolved by assuming that an external depot provides the source for less terminally-differentiated T cells capable of controlling tumor growth.
5. It would also appear that the fact that the blood can provide an additional input represents a crucial input that is not taken into consideration in this model.

Reviewer #3: The paper by Hickey et al. develops a model of tumor-immune interactions via a spatial agent-based approach, using a previously published modeling framework (Vivarium) developed by some of the authors on this paper. This model is partly calibrated from some experimental results, as well as multiplex tissue imaging.

The approach, integration of modeling and multiplex data, and the results are interesting; however, the paper suffers from a lack of detail when it comes to the modeling methods in particular, to the extent that it is difficult to put any results from the study in context of the modeling assumptions that are made. There is no presentation of any of the equations used in the study, despite the numerous interactions that exist in the model. There are transitions with cell states that depend on the amount of environmental molecules; are these sharp transitions, probabilistic, smooth functions? Other transitions depend on length of time; linearly? Or saturation? Many processes in biology have classic forms such as Michaelis-Menten, Hill functions, exponential, logistic, etc. When building a model, the use of these different terms changes the outcome of the model and impacts the interpretation of the results.

The authors have provided a link to GitHub where the code is stored. However, this should not be substitute for a full description of each aspect of the model in a supplementary methods section within

the paper. A reader interested in the modeling should not need to be fluent in python, etc., to investigate the mechanistic aspects of the model. I.e., the mathematical model is distinct from the implementation of it in a programming language.

Therefore, the paper should be revised by including a detailed description of each biological process in the model along with associated mathematical forms used to model such. Even when using a framework such as Vivarium or similar, the model should be able to be described outside of the functionality of such a framework or the specific implementation within a code language.

Regarding the results, much of the modeling outcomes rest on the premise that tumor cells switch between a proliferative state that has low PD-L1 and MHC-I expression, and a quiescent state that has high expression of the molecules. The authors do not explain the reason for correlating proliferation with these elements. At the very least, the interaction between PD-L1 expression and cell cycle/proliferation is complicated. Some papers suggest the opposite of the assumptions used in this paper (see PMID 30728908 for example, among others). The authors should investigate the effect of this assumption. What happens if this relationship is decoupled, or reversed? Basically, if the results imply that tumor phenotype switching is important for t-cell therapy, then the chosen phenotypes must be robust in the literature.

Some other points:

- * The paper is written with the expectation that the reader knows tumor immunology. The biological background of the mechanisms being modeled is not well described or referenced. Molecules need to be defined and their function briefly explained in context of the study (TCR etc.). Furthermore, common nomenclature includes dashes: PD-L1, PD-1, MHC-I.
- * Supplemental Fig 2, B, D, F, & H are not particularly clear or useful without better descriptions in the caption. What are the colors, the axes, etc.
- * In Fig 2E, why is there a reduction in INF γ at t=9.7 hr when the numbers and positions of PD-1+ T cells and tumor cells are (more or less) the same across t = 5.5hr, 7.7hr and 9.7hr? Also, what caused the increase in INF γ at 13.5 hr?
- * In Fig 3A: What do the different colored cells mean at d0 and d10? Provide a legend.
- * In Fig 3C, what do the gated 7.8% cells indicate? Additionally, is the 1.12% cells that indicate T cells always the case, or is this a function of the 'amount' of T cells transferred into the tumors?
- * In Fig 3D; until t = 31.5hrs, are T cells present or not shown?
- * Figures 3E-H, what units of time? Also other subpanels need better axes labels

Minor points and typos:

- * T cell / T-cell: hyphenate when T-cell is an adjective
- o E.g.: The T cells enter the system, and T-cell proliferation begins

Authors' response to the reviewers' first round comments

Attached.

Editorial decision letter with reviewers' comments, second round of review

Dear Garry,

I hope this email finds you well. The reviews are back on your revised manuscript and I've appended them below. You will see that while the Reviewers appreciate your revisions, they continue to raise some important concerns that we will need to see addressed. We are not worried about Reviewer 1's comments about lack of clarity about the immediate clinical translatability of your findings (major comment 1), and we would like you to focus on ensuring the validity of the claims made in the paper.

To help guide that revision, I've made a few notes directly on the reviews and highlighted points that seem to warrant special attention. If you have any questions or concerns about the revision, I'd be happy to talk about them, either over email or over Zoom. More technical information and advice about resubmission can be found below my signature. Please read it carefully, as it can save substantial time and effort later.

- While this work does strike me as a good use of modelling, in light of Reviewer 1's major comment 2, I'd ask that you see if you can better articulate why this is so for the benefit of the broader readership.
- Reviewer 1 Minor Point 3: Please clarify how the DC interactions with T cells are modelled and justify your modelling choices and its granularity. Please consider whether adding detail to your models along what the reviewer suggests would be productive and let us know what conclusion you come to and why. In general we would not ask to add detail to models unless these are justified given the problems being addressed, but we would need to see these choices and lack of detail convincingly justified.
- While Reviewer 3 is clearly very positive about the revised manuscript, we consider addressing their major comment crucial to address in full if we are to move forward towards publication - we need to be confident that we are learning about the biology of interest, not just about the model. (It is for similar reasons that we ask that you also address Reviewer 1's remaining concerns about the parameters.)

I look forward to seeing your revised manuscript.

All the best,

Bernadett

Bernadett Gaal, DPhil
Editor-in-Chief, *Cell Systems*

Reviewer comments:

Reviewer #1: The authors have addressed the concerns raised in the previous review and made the necessary modifications accordingly. However, there are still several remaining questions and suggestions.

Two general questions:

1. While the tumor microenvironment is more complex and consists of various other cell types, it is acceptable to just focus on T cells and tumor cells as the primary cell types in the model to investigate their direct interactions in T cell therapy. However, the article does not provide a definitive conclusion on how to enhance the effectiveness of T cell therapy according to the results. One significant perspective highlighted in this article is the significance of the tumor cell phenotype, but how to utilize it in T cell therapy remains unclear.

2. It appears that the many findings of this article could also be investigated directly through in vitro or in vivo experiments, complemented by spatial technologies like CODEX. Additionally, conclusions derived from computational models typically require further experimental validation. Therefore, it is crucial to clearly articulate the distinct advantages and usage of this model compared to conventional experiments to demonstrate its value. It is somewhat confusing in this article as the results of conventional experiments were utilized both as input for the model and for its validation.

Additional minor points:

1. In this article, the authors emphasized the importance of spatial positioning for T cells and tumor cells. Apart from the initial position setting, other parameters, such as migration parameters, also play a vital role in this process. It is crucial to provide a rationale for these parameters under varying conditions. In the second part of the Results, the authors kept the ratio of tumor to T cells consistent with the in vivo mouse model but did not make any adjustments to other parameters.

2. The authors incorporated a substantial amount of experimental data, such as the spatial information from CODEX, to initialize the model and enhance its realism. However, the model still lacks dynamic experimental data to validate its accuracy, particularly regarding the selection of parameters.

3. It is commendable to incorporate additional cell types into this model. However, there is a lack of detailed description regarding the inclusion of DCs and their interactions with T cells. It is crucial to consider specific molecular interactions such as MHC-I or MHC-II with TCRs, as well as the involvement of costimulatory or coinhibitory molecules with their respective receptors when exploring the interactions between DCs and T cells.

4. In the final section of the Results, the authors introduced the concept of the lymph node (LN) as a supportive microenvironment for T cells outside the tumor. While the focus was primarily on the tumor regions in silico, the mouse model emphasized the tumor-draining LNs. It remains unclear how the characteristics of T cells in the tumor region differ between the mouse model with 25% PD1+ T cells and 75% PD1+ T cells, and whether these differences align with the in-silico results.

Reviewer #2: The reviewers' comments were adequately addressed in the manuscript. There was one statement, however, that was confusing:

"Notably, we observed that for IFN γ -treated tumor there is a reduction in cells that were quiescent or in the G0 phase of the cell cycle (negative for IdU and pRb (S807 S811))"
Should this statement be that there was an increase, not a reduction of quiescent cells following treatment?

Reviewer #3: The revision by Hickey et al. has greatly improved the presentation, content, and interpretability of the paper, and the results have become more interesting as a result. I commend the authors in their thorough efforts to address the reviews and in the process creating an excellent paper. I have a few remaining questions regarding the results, now that the methods and interpretations have become clearer.

In general, the model is highly complex, and it can be hard to distinguish intuitively what salient

features are emergent behavior due to spatiotemporal dynamics, and what features are the result of potential artifacts of model parameterization and initialization. It would be useful then, to examine a couple of aspects of the results where the distinction is not clear.

First, Figures 3H, 5C, and 7C show the T-cell subtypes. In all three cases, representing outputs of the model with different situations, the 25% and 75% curves show the sharp transient of phenotypic switching at what appears to be the same time. It is unclear why this should be the case. There is some slight variation, but invariably around 45-48hrs, everything happens. In a stochastic simulation I'd expect more variation in when such transitions occur, unless there is some "timer effect" based on T-cell function that starts counting from the start of the simulation? If there is a "timer" effect, then the question is, would such a timer naturally be in place if the simulation were started "a few days earlier"? The point here is that these tumor and immune cells are in positions and states that have a history. How much of this transient behavior then is due to initialization?

Similarly, Figure 5E feels a little odd in terms of the initialization. In panel A, on the right, the 75% and 25% sections chosen appear reasonably similarly chosen, in that it captures a section of the tumor and also a little bit of the edge. However, in panel E, why are these initialization pieces set in a larger domain, and more to the point, in the lower left corner? In particular, I wonder how the fact that in 25%, the edge is exposed to the open domain above, whereas in 75%, the edge of the sample is trapped down in the corner. Does this matter? It would be interesting to see that same simulation, but starting with the sample cells in the center of the open domain, allowing growth in all directions. Equally interesting would be starting these sample initial conditions in a narrow domain where the edge of the sample had open domain to grow into, but the three other sides of the sample were constrained by the edge of the domain (because in the larger original sample, they were inside the tumor and therefore constrained).

Overall, this is truly an excellent integrated paper and I cannot thank the authors enough for fully embracing and expanding upon my previous review (and that of the other reviewers). If possible I'd love to see some exploration of the above points to ensure that the behaviors shown are not partially due to artifacts of the initialization.

Minor points and typos:

- * "achieve anti-tumor killing direct recognition", missing "via" or similar
- * The issue with T-cell hyphenation is still inconsistent: T-cell therapy, T-cell functionality, T-cell inhibition, T-cell persistence, etc.
- * Similarly, PD-1 and PD-L1 are often lacking the proper hyphens
- * Minor aesthetic detail, but in Fig 3D, the little inset window at the end of second row isn't positioned over the tumor in the way the large inset view would indicate.

Authors' response to the reviewers' second round comments

Attached.

Editorial decision letter with reviewers' comments, third round of review

Dear Dr. Nolan,

I'm very pleased to let you know that the reviews of your revised manuscript are back, the peer-review process is complete, and only a few minor, editorially-guided changes are needed to move forward towards publication.

In addition to the final comments from the reviewers, I've made some suggestions about your manuscript within the "Editorial Notes" section, below. Please consider my editorial suggestions carefully, ask any questions of me that you need, make all warranted changes, and then upload your final files into Editorial Manager. **We hope to receive your files within 5 business days. Please email me directly if this timing is a problem or you're facing extenuating circumstances.**

I'm looking forward to going through these last steps with you. Although we ask that our editorially-guided changes be your primary focus for the moment, you may wish to consult our FAQ (final formatting checks tab) to make the final steps to publication go more smoothly. More technical information can be found below my signature, and please let me know if you have any questions.

All the best,

Bernadett

Bernadett Gaal, DPhil
Editor-in-Chief, Cell Systems

Editorial Notes

Transparent Peer Review: Thank you for electing to make your manuscript's peer review process transparent. As part of our approach to Transparent Peer Review, we ask that you add the following sentence to the end of your abstract: "A record of this paper's Transparent Peer Review process is included in the Supplemental Information." Note that this **doesn't** count towards your 150 word total!

Also, if you've deposited your work on a preprint server, that's great! Please drop me a quick email with your preprint's DOI and I'll make sure it's properly credited within your Transparent Peer Review record.

Manuscript Text:

- House style disallows editorializing within the text (e.g. strikingly, surprisingly, importantly, etc.), especially the Results section. These terms are a distraction and they aren't needed—your excellent observations are certainly impactful enough to stand on their own. Please remove these words and others like them. "Notably" is suitably neutral to use once or twice if absolutely necessary.

- Please double-check that you use the word "significantly" in the statistical sense only.

Figures and Legends:

Also, please look over your figures keeping the following in mind:

- Bar graphs are not acceptable because they obscure important information about the distributions of the underlying data. Please display individual points within your graphs unless their large number obscures the graph's interpretation. In that case, box-and-whisker plots are a good alternative.
- Please ensure that every time you have used a graph, you have defined "n's" specifically and listed statistical tests within your figure legend.
- When figures include micrographs, please ensure that scale bars are included and defined within the legend, montages are made obvious, and any digital adjustments (e.g. brightness) have been applied equally across the entire image in a manner that does not obscure characteristics of the original image (e.g. no "blown out" contrast). **Note that all accepted papers are screened for image irregularities, and if this advice is not followed, your paper will be flagged.**
- Please ensure that all figures included in your point-by-point response to the reviewers' comments are present within the final version of the paper, either within the main text or within the Supplemental Information.

STAR Methods: Note that Cell Press has recently changed the way it approaches "availability" statements for the sake of ease and clarity. Please revise the first section of your STAR Methods as follows, noting that the particular examples used might not pertain to your study. Please consult the STAR Methods guidelines for additional information.

RESOURCE AVAILABILITY

Lead Contact: Further information and requests for resources and reagents should be directed to and will be fulfilled by the Lead Contact, Jane Doe (janedoe@qwerty.com).

Materials Availability: This study did not generate new materials. *-OR-* Plasmids generated in this study have been deposited at [Addgene, name and catalog number]. *-OR-* etc.

Data and Code Availability:

- **Source data statement** (described below)
- **Code statement** (described below)
- Any additional information required to reanalyze the data reported in this paper is available from the lead contact upon request.

Data and Code Availability statements **have three parts and each part must be present. Each part should be listed as a bullet point, as indicated above.**

Instructions for section 1: Data. The statements below may be used in any number or combination, but at least one must be present. They can be edited to suit your circumstance. **Please ensure that**

all datatypes reported in your paper are represented in section 1. For more information, please consult this list of standardized datatypes and repositories recommended by Cell Press.

- [Standardized datatype] data have been deposited at [datatype-specific repository] and are publicly available as of the date of publication. Accession numbers are listed in the key resources table.
- [Adjective] data have been deposited at [general-purpose repository] and are publicly available as of the date of publication. DOIs are listed in the key resources table.
- [De-identified human/patient standardized datatype] data have been deposited at [datatype-specific repository]. They are publicly available as of the date of publication until [date or delete “until”]. Accession numbers are listed in the key resources table.
- [De-identified human/patient standardized datatype] data have been deposited at [datatype-specific repository], and accession numbers are listed in the key resources table. They are available upon request until [date or delete “until”] if access is granted. To request access, contact [insert name of governing body and instructions for requesting access]. [Insert the following when applicable] In addition, [summary statistics describing these data/processed datasets derived from these data] have been deposited at [datatype-specific repository] and are publicly available as of the date of publication. These accession numbers are also listed in the key resources table.
- Raw [standardized datatype] data derived from human samples have been deposited at [datatype-specific repository], and accession numbers are listed in the key resources table. Local law prohibits depositing raw [standardized datatype] datasets derived from human samples outside of the country of origin. Prior to publication, the authors officially requested that the raw [adjective] datasets reported in this paper be made publicly accessible. To request access, contact [insert name of governing body and instructions for requesting access]. [Insert the following when applicable] In addition, [summary statistics describing these data/processed datasets derived from these data] have been deposited at [datatype-specific repository] and are publicly available as of the date of publication. These accession numbers are also listed in the key resources table.
- The [adjective] data reported in this study cannot be deposited in a public repository because [reason]. To request access, contact [insert name of governing body and instructions for requesting access]. [Insert the following when applicable] In addition, [summary statistics describing these data/processed datasets derived from these data] have been deposited at [datatype-specific or general-purpose repository] and are publicly available as of the date of publication. [Accession numbers or DOIs] are listed in the key resources table.
- This paper analyzes existing, publicly available data. These accession numbers for the datasets are listed in the key resources table.
- [Adjective or all] data reported in this paper will be shared by the lead contact upon request.

Instructions for section 2: Code. The statements below may be used in any number or combination, but at least one must be present. They can be edited to suit your circumstance. *If you are using GitHub, please follow the instructions here to archive a “version of record” of your GitHub repo at Zenodo, then report the resulting DOI. Additionally, please note that the Cell Systems strongly recommends that you also include an explicit reference to any scripts you may have used throughout your analysis or to generate your figures within section 2.*

- All original code has been deposited at [repository] and is publicly available as of the date of publication. DOIs are listed in the key resources table.
- All original code is available in this paper’s supplemental information.
- This paper does not report original code.

Instructions for section 3. Section 3 consists of the following statement: Any additional information required to reanalyze the data reported in this paper is available from the lead contact upon request.

In addition,

STAR Methods follows a standardized structure. Please reorganize your experimental procedures to include these specific headings in the following order: LEAD CONTACT AND MATERIALS AVAILABILITY (including the three statements detailed above); EXPERIMENTAL MODEL AND SUBJECT DETAILS (when appropriate); METHOD DETAILS (required); QUANTIFICATION AND STATISTICAL ANALYSIS (when appropriate); ADDITIONAL RESOURCES (when appropriate). We’re happy to be flexible about how each section is organized and encourage useful subheadings, but the required sections need to be there, with their headings. They should also be in the order listed. Please see the STAR Methods guide for more information or contact me for help.

Please ensure that original code has been archived in a general purpose repository recommended by Cell Press and that its DOI is provided in the Software and Algorithms section of the Key Resources Table. If you’ve chosen to use GitHub, please follow the instructions here to archive a “version of record” of your GitHub repo at Zenodo, complete with a DOI. Thank you!

Currently, you don’t have a **Key Resources Table** (KRT). Note that the key resources table is required for manuscripts with an experimental component, and if a purely computational manuscript links to any external datasets (previously published or new), code-containing websites (e.g. a GitHub repo, noting that DOIs are strongly preferred), or uses non-standard software, it needs to include a key resources table that details these aspects of the paper. Purely computational or theoretical papers that don’t contain any external links and use standard software don’t require a key resources table, although you’re welcome to include one if you like. For details, please refer to the Table Template or feel free to ask me for help.

Thank you!

Reviewer comments:

Reviewer #1: The latest manuscript has effectively addressed the concerns raised by the reviewers. This article introduces a method to improve our understanding of the interactions between T cells and tumor cells in terms of spatiotemporal dynamics. Several intriguing ideas have been proposed. For example, it is suggested that the conversion of tumor phenotype may have a more significant impact than direct killing, and that external supplementation of T cells is crucial and necessary for maintaining the T cell phenotype in the tumor microenvironment. The results have also prompted further questions that require additional exploration. It was observed that PD-L1+ tumor cells inhibit T cells through the interaction between PD-1 and PD-L1, but at the expense of their own expansion ability. No differences were observed within the lymph nodes of different conditions, which may be due to the expansion preference of T cells with different phenotypes in lymph nodes or other functional structures in the tumor microenvironment, such as tertiary lymphoid structures.

Only some minor questions and suggestions:

- 1) In the Figure 2D model, there were multiple modes of cell death, indicating that the number of cell deaths did not correspond to T cell killing. This requires correction for deaths due to other reasons, such as apoptosis.
- 2) In the third paragraph of the last section, should the statement be "The sustained levels of T cells and percentages of PD-1- T cells within the tumor over long periods of time had a drastic impact on the total number of tumor cells past 36 hours of simulated time (Fig. 7C)"?
- 3) This article solely focused on the function of IFN- γ on tumor cells. However, IFN- γ secreted by T cells can also affect T cells themselves. The authors should take this into consideration if they further develop the model later. IFN- γ has been demonstrated to play a critical role in the regulation of CD8 T cell expansion and contraction during immune responses.

Reviewer #3: I'd like to thanks the authors for doing such an extensive and complete response on this round of revision, well done!

Authors' response to the reviewers' third round comments

Attached.

Response to Reviewers for Hickey et. al. in Cell Systems-D-22-00549

Response to Referees

Integrating Multiplexed Imaging and Multiscale Modeling Identifies Tumor Phenotype Transformation as a Critical Component of Therapeutic T Cell Efficacy

Overall Response

We thank the referees for their valuable feedback and positive evaluation of our manuscript "Integrating Multiplexed Imaging and Multiscale Modeling Identifies Tumor Phenotype Transformation as a Critical Component of Therapeutic T Cell Efficacy" (Cell Systems-D-22-00549). We carefully considered every comment raised by the referees and revised the paper to address the concerns raised by the referees. Our revised manuscript includes new analyses and has substantial changes that expand and clarify our original findings, including additional validations and explanations of our model. We believe that the revised manuscript strengthens the confidence in our findings that T cells modulate tumor cell phenotype, this is critical for overall T cell efficacy, preserving T cell phenotype in the tumor microenvironment is a critical step while also balancing tumor engagement, and now responses from distal immune organs.

In summary, in response to the referees' feedback we have:

- Substantially expanded our model documentation both within the text, Methods, and a new Supplemental Text that includes model flow charts, describes model development, mathematical foundations, data and parameter sources, and implementation.
- Performed additional in vitro (CyTOF metabolic profiling of tumor cells) and in vivo experiments (14 additional mice tumors treated with T cells imaged with CODEX multiplexed imaging panel of 42 markers) to increase our insight into biological processes modeled and further validate our model. This generated 4 new main figures and 8 new Supplemental figures in the revised manuscript.
- Added an additional cell type of dendritic cell and added another compartment of the lymph node where cells could go, get activated, and come to the tumor microenvironment to more accurately model in vivo systems. We performed additional in silico simulations that generated 6 new main figures and 3 new Supplemental figures in the revised manuscript.

Our response is structured as follows:

- Response to Referee 1 (page 2)
- Response to Referee 2 (page 14)
- Response to Referee 3 (page 21)

Responses to reviewers and changes in the manuscript are denoted in blue color.

Reviewers' comments

Reviewer #1: In this manuscript, the authors integrated CODEX multiplexed tissue imaging with multiscale modeling software, to model key factors that influence T cell therapy efficacy and further understand the interactions of T cell therapies with cancer at multiple scales. Utilizing the Vivarium, an integrated model that covered multiple spatial and temporal scales, the authors first demonstrated that both the phenotype conversion of T cells and tumor cells enhanced T cell recognition and killing.

An appreciated innovation of this work is to use the CODEX multiplexed imaging derived spatial context of tumors as initial states of multiscale simulations to study the spatial localization and function of T cells. They demonstrated the synergy of employing both the top-down and the bottom-up approaches, in other words, the deconstruction and reconstruction of the interaction networks. By doing so, they concluded that the conversion of tumor cell phenotype is more critical for tumor control than T cell phenotype preservation, which suggested potential design criteria and patient selection metrics for T cell therapies. In practice, they provide both a conceptual and a practical paradigm, which as exemplified in this manuscript, is fueled by several rounds of experiment-derived theory, theory-guided simulation, and simulation-inspired experiment.

However, I have major concerns about the methodology applied and some of the conclusions it led to. Briefly, the multiscale simulations in this study are insufficient to reflect both the tumor microenvironment and the process changes during T-cell therapies. In addition, given that some key cellular components such as CAF, which have been recognized to influence the infiltration and function of T cells, have been ignored in this work, some biological conclusions in this manuscript should be reconsidered.

Below are listed major concerns:

1. **It is necessary to prove the reliability of the multiscale agent-based model before using it to explain specific biological processes rather than using it to formulate mechanistic explanations directly.** Biological processes that are proven can be used as standards to evaluate the model. It is commendable that the authors strive to provide literature-supported and lab-derived parameters to create the model. Additionally, it is necessary to include information about specific contexts and conditions in which these values were chosen, such as the specific antigen, or chronic or acute infection. In addition, some of the sources should be rechecked. I randomly check the sources of two parameters, "PD1n_IFNg_production" (1.62e4 molecules/cell/s) and "PD1p_IFNg_production" (1.62e3 molecules/cell/s). One denoted source, "Bouchnita et al., 2017" provides "the secretion rate of type I IFN

by single activated APC (plasmacytoid dendritic cell): 1.6×10^4 molec/hr" in its appendix without mentioning IFN γ , which belongs to type II IFN. In the other source, "Zelinskyy et al., 2005" only measured perforin, granzyme B, and b-actin of activated CD8+ T cells.

We agree with the reviewer that it is important to validate the model before using it extensively to understand biological processes. To this end we worked to build a model that met with expectations. One experiment we performed to calibrate model parameters was an in vitro killing assay where we were able to control exactly how many T cells and tumor cells we co-incubated together and measured the amount of tumor death over time. We performed this killing assay with PMEL T cells with either B16-F10 tumor cells that were incubated with IFN γ or B16-F10 cells that were not incubated with IFN γ , shown as **Fig. 2C**.

Our simulation that we had adjusted parameters both from our own data and the literature matched the data from our in vitro experiment **Fig. 2G**, where we see increased killing of cancer cells that have been incubated with IFN γ and nearly 60% cytotoxicity by 12 hours in both the simulation and in vitro data.

Consequently, we have added the following text to the manuscript:

"In this simulation, we quantified cytotoxicity to enable a comparison with our in vitro data. Cytotoxicity was quantified by evaluating the number of cell

deaths and normalizing the results against a simulation lacking T cells. In addition to using lab-derived molecular parameters as input, we initialized the simulation with identical cellular parameters used within the in vitro experiment, such as the same effector-to-target ratio, percentage of T cell phenotypes, and duration in culture. Quantification of killing by cytotoxicity indicated an important role of tumor phenotype on T-cell killing even at early time points (Fig. 2G). Moreover, when tumor cells were pre-treated with IFN- γ , we observed a remarkable increase in cytotoxicity, reaching nearly 60% by 12 hours compared to untreated tumor cells incubated with an equivalent number and type of T cells. This finding mirrors the observed increase in our in vitro experiment's cytotoxicity (Fig. 2C). Consequently, our simulation results not only replicated the kinetics and magnitude of our in vitro killing data but also reinforced the overall conclusion. This alignment validates both our model setup and molecular parameters.”

We also thank the reviewer for acknowledging our effort to provide literature-supported or lab derived-values for our model. We agree that the IFN γ secretion rate is an important parameter. We measured this at the single cell resolution using CyTOF and observed there is a variation of IFN γ expression per T cell at a given snapshot of time, shown as **Fig. 2D**.

We would expect this based on T cell activation kinetics that depend on activation signals and observe variable IFN γ expression in our simulation. Indeed we see this if we measure the IFN γ expression at the single-cell level (**Supplemental Fig. 3B**).

We have now also done experiments to measure quantitatively the IFNg secretion rates from T cells that have been activated to mimic secretion levels within the tumor microenvironment of CD8+ T cells. This essentially captures averages of the overall expression of the population over time so that we can compare our value for IFNg expression programmed for activated T cells. We did this by harvesting splenocytes, activating them for 10 days (per Methods) *in vitro* to mimic antigen encounter within the tumor. Then we measured IFNg in the media of the T cells with an ELISA assay. We found that the rate $\sim 1e4$ IFNg mol/s/T cell was on the same order of magnitude to our previous value we obtained from literature of $1.62e4$ IFNg mol/s/T cell. We have added this as **Supplemental Fig. 4A**. We believe that this adds additional rigor and validity to our model system with this additional parameter validated.

In addition to measuring the IFNg secretion rate directly, we also have gone to greater lengths in collecting additional replicates of spatial multiplexed imaging data of tumors taken from mice that were treated with T cells over multiple replicates and time points (**see response to your comment #2**). Consequently, with support of additional *in vivo* multiplexed imaging data we are confident our model is reproducing the biological processes we are studying, specifically conversion of Tumor cell phenotype by CD8+ T cells and spatial responses of CD8+ T cell attack in the tumor microenvironment.

2. It was too simplified to only include T cells and tumor cells in the model as the tumor micro-environment was very complicated. In the section T cells induce tumor cell phenotype conversion in vivo, cell-type percentages were based on only T cells and tumor cells in in silico model but were based on all cells in CODEX multiplexed imaging data. Moreover, there were only three points in figure 4F, figure 4G, and figure 4H; more repetitions are needed, like selecting CODEX data of different regions and different time points except for three days.

We thank the reviewer for the in depth analysis of our model and acknowledgement that the tumor microenvironment is indeed incredibly complex. Our initial model was to understand better the interactions of T cells with tumor cells and particularly the relationship of T cells with tumor phenotype conversion. Nevertheless, we have also added another cell type, dendritic cells, to the model to improve physiological relevance. In particular, we intended to improve the model by adding processes of antigen presentation and activation of T cells within the lymph node and migration of T cells from the lymph node into the tumor. Adding dendritic cells and a lymph node compartment to the simulation, corrected the differences between our model and the in vivo data from CODEX multiplexed imaging, particularly the maintenance of PD1- CD8+ T cells over time. Adding an additional cell type and compartment led to 5 additional main Figures Fig. 7A-E and 3 additional Supplementary Fig. 9A-C.

Fig. 7

Supplementary Fig. 9

We added the following text surrounding the model developments and results:

"We speculated therefore that T cells could come from outside the tumor enabling founder T cells to reside in supportive microenvironments like the lymph node (LN), while daughter cells migrate into the tumor. We therefore extended our model by adding another cell type of the dendritic cell to our model. Dendritic cells are critical antigen-presenting cells that take up antigen and stimulate antigen-specific T cells in the LN. T cells then divide and leave the LN for effector function in the target tissue. We built our model system after these principles leveraging the same approach we took of literature and lab-derived parameters, which we used in building the previous model system (Fig. 7A).

We compared simulations with 25% PD1+ T cell treatment conditions initialized with and without the extra LN process that incorporated dendritic cells. There were increased and sustained number of T cells over three days of simulation time in the conditions where dendritic cells were able to activate T cells in LN (Fig. 7B). This came from added numbers of PD1- T cells to the tumor from the LN (Fig. 7C, Supplemental Fig. 9A). This result more closely matches our CODEX multiplexed imaging results of tumors (Fig. 4), where a subset of the T cells within the tumor are found to be PD1- even several days after adoptive transfer.

The sustained levels of T cells and PD1- T cells within the tumor over long periods of time had a drastic impact on the total number of tumor cells past 36 hours of simulated time (Fig. 7D). Part of the reason is due to increased T cell killing of tumor cells from conditions with dendritic cells (~1000 vs. 700 tumor T cell associated deaths) (Supplemental Fig. 9B, C). However, this

only accounts for a total difference in 300 tumor cell deaths, whereas we observe a difference in ~900 total tumor cells by 72 hours of simulated time (Fig. 7D). Most of this difference instead was due to the ability to sustain the conversion of a greater proportion of tumors to the inflamed, non-proliferative PDL1+ phenotype (Fig. 7E). Thus, emphasizing the importance of tumor phenotype conversion as a method of tumor cell containment within the anti-tumor immune response.”

Despite adding the additional cell type and lymph node compartment to our model that added additional clarity to processes and more closely matched in vivo data, we recognize that this still does not fully recapitulate the complex tumor microenvironment fully. Consequently, we created a new section called **Limitations of the Study** to add limitations of our model with the following text added:

“In future work, driven by molecular data acquired through multiplexed tissue imaging, the model should be expanded to include other cell types and anti-inflammatory molecules. Since the Vivarium framework for multiscale modeling is modular and compositional, it will be straightforward to add additional cell types with different representations of internal mechanisms and environmental interactions, such as how we added the dendritic cell and lymph node process. Not straightforward are the selection of molecular features and phenotypes that will increase the accuracy of the model under a broader range of conditions. Although we have some clues about missing components from our CODEX and RNA dataset (Hickey et. al., cosubmitted), additional data collection will be needed. For example, single-cell RNA sequencing of the cell types within the tumor will shed light on key molecular and cellular interactions. Building this complexity within in silico models will be necessary because as the number of intercellular connections are increased, it will become more difficult to recapitulate and deconvolute these networks within in vitro systems. Similarly, we focused on one in vivo tumor model system to understand relationship between tumors and antigen-specific T cells—in particular one that mimics tumor that lose or downregulate MHCI expression (Dersh et al., 2021; Dhatchinamoorthy et al., 2021). This one model does not recapitulate the diversity of human tumors, and consequently, additional tumor models and human tumor samples will need to be studied and integrated to build future branches of the model.”

In addition to adding another cell type, we also have gone to greater lengths in collecting additional replicates and time points of spatial multiplexed imaging data of tumors taken from mice that were treated with T cells.

Consequently, we completed additional in vivo experiments from a total of 14 additional mice. Briefly, we harvested T cells from transgenic mice and activated the T cells for 10 days in vitro with or without 2HC (metabolic inhibitor). At the same time we injected tumors into mice. Ten days following tumor injection, we adoptively transferred T cells to tumor-bearing mice. We then either waited 1 or 3 days before harvesting the tumors. We then imaged the tumors with our CODEX multiplexed imaging antibody panel of 42 markers, processed images, segmented out cells, and performed unsupervised cell type clustering to identify cell types and locations within the tumors.

In tandem we extracted information from simulations that ran for the same amount of biological time. We co-analyzed these new experimental results with simulations. We replaced original Figures 4F-H with new **Figures 4F-H**, which now represent extra replicates across the conditions *No T cells*, *25% PD1+ T cells*, and *75% PD1+ T cells* from tumors taken 3 days post-transfer of adoptive T cells (n=4-5 per group). We also include two additional figures: **Figures 4I**, which highlights the percentages of PD1- CD8+ T cells in tumors taken 3 days post-transfer of adoptive T cells (n=4-5 per group) and **Supplemental Fig. 6B**, which shows the correspondence of the different cell types taken from tumors after 1 day post-transfer of adoptive T cells (n=4).

Taken together, these data demonstrate that simulation cell type percentages follow overall trends within tumors taken from *in vivo* experiments. Consequently, with support of additional *in vivo* multiplexed imaging data paired with replicates of our simulations we are confident our model is reproducing the biological processes we are studying.

3. It is not rational to use a proportion of PD1+ T cells as a substitution for effector and memory T cells. There are many other different characteristics between effector T cells and memory T cells, like proliferating capacity. More parameters should be built into the model.

While PD1 has been suggested as a marker of effector T cells [Sharpe & Pauken, 2018], partially due to the fact that all recently activated CD8+ T cells will express PD1, we agree with the reviewer that it is not as simple as using one marker such as PD1+ as a substitution for separating out the memory and effector cells. Consequently, we further stained our two populations of cells 25% PD1+ T cells and 75% PD1+ T cells with two additional markers used in phenotyping CD8+ T cells (CD62L and CD44) to further differentiate effector versus memory populations in our T cells. While not every PD1+ T cells is effector, largely the separation of memory versus effector is conserved (25% PD1+ T cells: 68% memory and 75% PD1+ T cells: 65% effector). We have added this as **Supplementary Fig. 4C**.

We have also adapted our language in the text to be careful of the separation within the text and include these nuances when broadly referencing CD8+ T cell phenotypes:

*"Additional characterization by CyTOF showed further subphenotypes that separate further beyond simple PD1 staining (Hickey et al cosubmitted), but two main categories of memory and effector T cells were broadly separated by PD1 status (**Supplemental Fig. 4C**), and we used these as inputs to our model."*

REF 1 - Sharpe, A., Pauken, K. The diverse functions of the PD1 inhibitory pathway. **Nat Rev Immunol** 18, 153–167 (2018). <https://doi.org/10.1038/nri.2017.108>

4. This model was more suitable to raise hypotheses and provide ideas for experimental validations. More experiments are needed to confirm the results.

We thank the reviewer for this comment and agree that this model has been helpful to raise hypotheses and also provide ideas for experimental validations. We highlight that this is a focus of the model within both the **Introduction:**

"However, there is synergy in employing both approaches simultaneously to drive discovery of a more accurate tissue representation. As demonstrated here, multiscale modeling can be used to identify key points of the system for perturbation. Furthermore, information from multiplexed imaging feeds multiscale agent-based models by providing more accurate parameter values, initial states, and update rules."

and **Discussion:**

"We developed a scalable agent-based model of T cell therapy of tumors to complement, leverage, and probe CODEX multiplexed imaging datasets of T cell-treated tumors. We specifically explored the importance of tumor phenotype on T cell therapy efficacy by incorporating molecular switches of cellular phenotype and function that led to tissue-level phenomena in our data-informed multiscale agent-based model. We then used findings from the simulations to guide design of an in vivo experiment and choice of antibodies for use in CODEX imaging to catalog cell types and transitions we saw within our models. This synergistic back-and-forth of model and data across biological scales revealed critical design components of effective T cell therapies (Fig. 7)."

We think that this is a strength of this modeling approach where iterative modeling and experimentation can build off each other and help to answer biological questions. While no model will perfectly represent the true system, one that helps to understand key relationships within the system in a dynamic manner is useful. We will continue to iterate and improve the model as discussed within the **Limitations of the Study** section as mentioned in **response to your comment #2**. Already, we have added another cell type (dendritic cells) and another compartment of the lymph node, where these cells can activate T cells to divide and leave for the tumor as discussed in detail in the **response to your comment #2**.

While philosophically we believe that iterative model development mixed with experimentation is useful, we also performed extensive model testing and benchmarking including experimental derivation of parameters and validating model performance against *in vitro* (**Fig. 1**) and *in vivo* experiments (**Fig. 3**). This is described in detail within the **response to your comments #1 & #2**. Consequently, we have confidence that while our model is far from perfect, it is recapitulating the processes we are interested in studying in the temporospatial T cell response within the tumor.

Below are listed minor concerns:

1. Probability, the supportive micro-environments for T cells extending to the tumor in Figure 6I can be correlated with tertiary lymphoid structures.

We thank the reviewer for this suggestion. We have done careful analysis of the immune microenvironments within these tumors and in this case we do not see mature tertiary lymphoid structures (since there is an absence of B cells in the CODEX multiplexed imaging data), though we see dense areas of

immune infiltration (see accompanying co-submitted manuscript by Hickey et. al.).

2. It was not clear about the spatial distribution of T cells and tumor cells in the first and second parts of the results. Whether cells were randomly distributed or distributed according to certain rules was not clear.

We have now clarified within the **STAR Methods** how spatial distribution of T cells were initialized in the various experiments:

Initialization of Experiments: *The number of T cells and tumor cells as well as the proportions of phenotypes for each were based on experimental data. Briefly, most of the experiments started off with a total of 1200 tumor cells and 12 CD8+ T cells. For simulations in Figures 2 and 3, T cells were initialized randomly in the tumor bed. For simulations in Figure 5, T cells were initialized based on the locations of CODEX multiplexed imaging data. For simulations in Figure 6, T cells were located randomly inside or outside the tumor bed as specified.*

Reviewer #2: This manuscript presents the results of studies in which data from in vitro and in vivo studies were used in an iterative process to generate an in silico model for T cell/tumor cell interactions. There were, however, some points that need to be further addressed.

1. It was difficult to follow the way in which the model was derived, and rather than including Fig. 1 which was somewhat inscrutable **it would be useful to present a flowchart of detailing precisely what data was fed into the model from the in vivo and in vitro experiments.**

We thank the reviewer for the suggestion to improve the manuscript. A new **Supplemental Text document** describes overall model development, design, and implementation in great detail (>10 pages) to illustrate how data and models are connected to each other within our simulation. To this end we created multiple flow charts that outline the various process modules and wiring diagrams for how they connect through shared states. We include more information about Vivarium, the integrative simulation engine that orchestrates these various processes. Processes are individually described with their modeling assumptions, mechanisms, and equations. We also include an overall figure that illustrates which data comes from in vitro and in vivo experiments, and parameter tables that specify whether data or parameters were input to the model based on either literature values or from in vitro or in vivo derived data. One example of a flow chart that identifies such data input that is only one of several flow charts included is included below with yellow outline indicating data from experiments getting input into the simulation architecture:

See Supplemental Text document for additional details and examples.

2. The in silico model appears to be derived from a single in vivo time point along with 1 in vitro study of the time course of tumor cell cytotoxicity. It would seem that additional in vitro and in vivo data collected over an extended time course would result in the generation of a better model. If not, please explain the reasoning behind the decision to base the model on a more limited set of experimental data. In addition, it is not clear how IFN-g expression data was integrated into the model without any experimental measurements of expression of this cytokine.

We thank the reviewer for the suggestion of additional *in vivo* and *in vitro* data collection. We also have gone to greater lengths in collecting additional replicates and time points of spatial multiplexed imaging data of tumors taken from mice that were treated with T cells. Consequently, we completed additional in vivo experiments from a total of 14 additional mice. See Response #2 to Reviewer #1 for a more detailed description. Briefly, we co-analyzed these new experimental results with simulations and replaced original Figures 4F-H with new Figures 4F-H.

We also include two additional figures: Figures 4I, which highlights the percentages of PD1- CD8+ T cells in tumors taken 3 days post-transfer of adoptive T cells (n=4-5 per group) and Supplemental Fig. 6B, which shows the correspondence of the different cell types taken from tumors after 1 day post-transfer of adoptive T cells (n=4).

Taken together, these data demonstrate that simulation cell type percentages follow overall trends within tumors taken from *in vivo* experiments. Consequently, with support of additional *in vivo* multiplexed imaging data paired with replicates of our simulations we are confident our model is reproducing the biological processes we are studying.

Also, we attempted to provide literature-supported or lab derived-values for our model parameters such as IFN-g expression from a number of papers including [Bouchnita et al., 2017; Zelinsky et al., 2005]. We agree that the IFNg secretion rate is an important parameter and we have now also done experiments to measure quantitatively the IFNg secretion rates from T cells that have been activated to mimic secretion levels within the tumor microenvironment of PD1- CD8+ T cells. This essentially captures averages of the overall expression of the population over time so that we can compare our value for IFNg expression programmed for activated T cells. We did this by harvesting splenocytes, activating them for 10 days (per Methods) *in vitro* to mimic antigen encounter within the tumor. Then we measured IFNg in the media of the T cells with an ELISA assay. We found that the rate $\sim 1e4$ IFNg mol/s/T cell was on the same order of magnitude to our previous value we obtained from literature of $1.62e4$ IFNg mol/s/T cell. We have added this as **Supplemental Fig. 4A**. We believe that this adds additional rigor and validity to our model system with this additional parameter validated.

3. Expression of class I on B16 is nearly undetectable in the absence of IFN-g but additional studies have demonstrated that extended culture will lead to recognition by PMEL class I-restricted T cells can be boot-strapped and thus the time course in Fig. 2B should have been extended beyond 14 hours. This also emphasizes what is somewhat unique about the B16 model, as the majority of human tumors, with the exception of B2M knock-out tumors, do not demonstrate such a severe deficiency of MHC expression. As such, it would be useful to model these interactions with a murine tumors that are not deficient for class I expression.

We thank the reviewer for their careful consideration of our data. We now provide an extended killing beyond 14 hours of its original and include it as new **Supplemental Fig. 1D**:

Largely beyond about 12 hours the cytotoxicity decreases for CD8+ T cells with B16F10 tumors that have been incubated with IFN-g. Cytotoxicity slightly increases for CD8+ T cells beyond 12 hours, peaking around 30 hours at about 10% cytotoxicity, and never reaches the peak cytotoxicity of CD8+ T cells with B16F10 tumors that have been incubated with IFN-g. Likely the decrease of cytotoxicity by both conditions represent decrease in viability of T cells within these long-term cultures without media change or addition of cytokines.

For MHC1 expression, indeed MHC1 expression is lower at baseline without IFN γ for B16-F10 as we showed in **Figure 2A**.

Additionally, from our CODEX multiplexed imaging stains of the tumor (*Hickey et. al co-submitted*), MHC1 is expressed at much higher levels on these tumor cells than all cell types (including immune subsets as antigen-presenting cells), suggesting that these levels of MHC1 are higher than even base-line levels across normal cells types. This may represent an increased dynamic range that we observe in the CyTOF, with low-level expression being simplified in the paper as negative. We have clarified this in the **Results** section with describing as upregulation of surface markers:

"As shown by CyTOF analysis, there was a phenotype change in about half of the tumor cells—characterized by upregulation of both anti-inflammatory (PDL1) and inflammatory (H2Db) surface markers in the group treated with IFN (Fig. 2A)."

Also, while we realize this may not be representative of all human tumors, it is representative of many tumors that have lost or downregulated MHC1 expression [Dersh et. al. 2021; Dhatchinamoorthy 2021]. We have added this to a new section that we have added to **Discussion** section called **Limitations of the Study**:

"Similarly, we focused on one in vivo tumor model system to understand relationship between tumors and antigen-specific T cells—in particular one that mimics tumor that lose or downregulate MHC1 expression [Dersh et. al. 2021; Dhatchinamoorthy 2021]. This one model does not recapitulate the diversity of human tumors, and consequently, additional tumor models and

human tumor samples will need to be studied and integrated to build future branches of the model."

REF 1] Dersh, D., Hollý, J. & Yewdell, J.W. A few good peptides: MHC class I-based cancer immunosurveillance and immunoevasion. Nat Rev Immunol 21, 116–128 (2021).

REF 2] Dhatchinamoorthy, Karthik, Jeff D. Colbert, and Kenneth L. Rock. "Cancer immune evasion through loss of MHC class I antigen presentation." Frontiers in immunology 12 (2021): 636568.

4. The results presented in Fig. 5 appear to indicate some control of tumor cell growth using 2-HC-treated T cells whereas the results presented in Fig. 6 do not demonstrate tumor control when T cells are initiated outside of the tumor. This may reflect the need for additional depots of T cells in local lymph nodes that can be drawn upon to mediate effective control, **but it is still unclear why the model should perform worse when T cells are initiated outside of the tumor if the discrepancies between the data and the model are proposed to be resolved by assuming that an external depot provides the source for less terminally-differentiated T cells capable of controlling tumor growth.**

We thank the reviewer for this comment that now has led to additional model parameters that we believe have improved the model and led to further insights into the T cell interaction and response in the tumor microenvironment. First, the model performs worse when T cells are initiated outside the tumor because despite the increased amounts of T cells that are able to get generated (from a less exhausted set of cells engaging with the tumor cells) they are unable to keep up with the exponential growth of cancer if they do not engage with the tumor in prolonged efforts. We have now clarified this with new text within the **Results** section:

*"Interestingly, despite the much higher numbers of T cells located outside the tumor, in our simulations, the tumors with T cells initialized outside the tumor grew more quickly than tumors with T cells initialized inside (~4500 cells vs. ~2500 cells at 75 h) (**Fig. 6D, Supplemental Fig. 8C-E**). Consequently, delay of T cell exhaustion came at the cost of enhanced early tumor growth rates, which even larger numbers of T cells were not able to overcome in the long run. Thus, trading less tumor engagement over time for an increased number of T cells in the future essentially gives the tumor a head-start in proliferation enabling exponential growth rates that extra T cells are not able to counteract. Since resting preserves phenotype, but does*

not enhance tumor outcome, a mechanism for enhanced tumor control in metabolically treated T cells is missing from our model.”

Second, to address the comment about additional depots of T cells in local lymph nodes, we have added another component of dendritic cells and a separate lymph node compartment that cells are able to be activated and then recruited to the tumor. In particular, we intended to improve the model by adding processes of antigen presentation and activation of T cells within the lymph node and migration of T cells from the lymph node into the tumor. Adding dendritic cells and a lymph node compartment to the simulation, corrected the differences between our model and the in vivo data from CODEX multiplexed imaging, particularly the maintenance of PD1- CD8+ T cells over time as the reviewer suggested. Adding an additional cell type and compartment led to 5 additional main Figures Fig. 7A-E and 3 additional Supplementary Fig. 9A-C.

Fig. 7

Supplementary Fig. 9

Response to Reviewers for Hickey et. al. in Cell Systems-D-22-00549

See response to Reviewer #1 Comment #2 for the additional text surrounding the model developments and results.

5. It would also appear that the fact that the blood can provide an additional input represents a crucial input that is not taken into consideration in this model.

We agree with the reviewer that the blood and the lymph nodes are important sources of T cells that could be continuously added to the tumor microenvironment. As a consequence, we have added both a dendritic cell type and also a separate lymph node compartment to the model system (see response to your **#4 comment**).

Reviewer #3: The paper by Hickey et al. develops a model of tumor-immune interactions via a spatial agent-based approach, using a previously published modeling framework (Vivarium) developed by some of the authors on this paper. This model is partly calibrated from some experimental results, as well as multiplex tissue imaging. The approach, integration of modeling and multiplex data, and the results are interesting; however, the paper suffers from a lack of detail when it comes to the modeling methods in particular, to the extent that it is difficult to put any results from the study in context of the modeling assumptions that are made. There is no presentation of any of the equations used in the study, despite the numerous interactions that exist in the model. There are transitions with cell states that depend on the amount of environmental molecules; are these sharp transitions, probabilistic, smooth functions? Other transitions depend on length of time; linearly? Or saturation? Many processes in biology have classic forms such as Michaelis-Menten, Hill functions, exponential, logistic, etc. When building a model, the use of these different terms changes the outcome of the model and impacts the interpretation of the results.

The authors have provided a link to GitHub where the code is stored. However, this should not be substitute for a full description of each aspect of the model in a supplementary methods section within the paper. A reader interested in the modeling should not need to be fluent in python, etc., to investigate the mechanistic aspects of the model. I.e., the mathematical model is distinct from the implementation of it in a programming language.

Therefore, the paper should be revised by including a detailed description of each biological process in the model along with associated mathematical forms used to model such. Even when using a framework such as Vivarium or similar, the model should be able to be described outside of the functionality of such a framework or the specific implementation within a code language.

We thank the reviewer for the suggestion. We have gone to much greater lengths to describe overall model development, design, and implementation in the new **Supplemental Text document**. This document goes through in great detail (>10 pages) how the model is set up including a detailed description of each biological process's parameters, assumptions, interfaces, and mathematical equations. Since much of what is important about this model is focused on interactions between processes, we also provide detailed flow charts and explanations how processes are connected to each other within the simulation, and how Vivarium orchestrates their co-simulation. While our codebase includes extensive documentation and testing (15 unit tests for different processes and compositions) for reproducibility and interpretability, the new Supplemental text and documentation enables

greater interpretation of what is modeled and how we model it from both a biological and mathematical perspective that is tied to the manuscript.

Regarding the results, much of the modeling outcomes rest on the premise that tumor cells switch between a proliferative state that has low PD-L1 and MHC-I expression, and a quiescent state that has high expression of the molecules. **The authors do not explain the reason for correlating proliferation with these elements.** At the very least, the interaction between PD-L1 expression and cell cycle/proliferation is complicated. Some papers suggest the opposite of the assumptions used in this paper (see PMID 30728908 for example, among others). **The authors should investigate the effect of this assumption.** What happens if this relationship is decoupled, or reversed? Basically, if the results imply that tumor phenotype switching is important for t-cell therapy, **then the chosen phenotypes must be robust in the literature.**

We thank the reviewer for this suggestion. Previously we had data that suggested this phenotype separation where there was a negative correlation between the percentage of Ki67+ Tumor cells and PD-L1+ MHC-I+ Tumor cells within our multiplexed imaging data regardless of treatment (*Hickey et. al. co-submitted*). We have confirmed this at the single-cell level by analyzing all the tumor cell subsets and quantifying the expression by their antibody staining in the multiplexed imaging dataset. We show now that the PDL1+ MHC1+ Tumor cells did not express Ki67 and Ki67+ Tumor-classified cells do not express PDL1 and MHC1. We have now added this as **Supplemental Fig. 6A.**

In addition, per the suggestion, we have performed an experiment now where we cultured a subset of B16-F10 tumor cells with IFNg and measured their metabolic activity with CyTOF panel targeting metabolic enzymes involved in various metabolic pathways. This was very interesting as it also suggested anti-proliferative effect on the tumor cells treated with IFNg. We

have added a new **Figure 2B** and **Supplementary Figures 1A-C** as a result.

Figure 2B

Supplementary Fig. 1A-C

We have added the following text to accompany these new results:

*"We also hypothesized that a drastic change in tumor cell phenotype would be accompanied by a change in metabolism. Consequently, we compared the tumor cells by staining with a CyTOF panel focused on cellular metabolism (**Supplemental Table 1**). Notably, we observed that IFN γ -treated tumor there is a reduction in cells that were quiescent or in the G0 phase of the cell cycle (negative for IDU and pRB-S807 S811) (**Fig. 2B**). These reductions in cellular proliferation were enriched for cells that expressed high levels of MHC-I (**Supplemental Fig. 1A**), and MHC-I positive cells that had entered cell cycle, had a lower mitotic index than MHC-I low counterparts (**Supplemental Fig. 1B**). Cells treated with IFN γ also had lower levels of the pentose phosphate pathway (**Supplemental Fig. 1C**), indicating both an impaired cellular antioxidant, DNA synthesis, and cell division processes.*

Overall, these data suggest that the IFNg not only causes inflammatory and anti-inflammatory markers like PDL1 and MHCI, but also causes substantial metabolic changes that inhibit proliferation of the tumor cells.”

This helps confirm that interaction between cell cycling and this inflamed phenotype are indeed conserved when exposed with IFNg. Likely there are several nuanced sub cell states that a tumor cell could express PDL1 (e.g., PMID 30728908) that may not necessarily correlate with cell cycling; however, for the purposes of our study and this model we observe that this expression is correlated both *in vitro* and *in vivo*.

Some other points:

* The paper is written with the expectation that the reader knows tumor immunology. **The biological background of the mechanisms being modeled is not well described or referenced.** Molecules need to be defined and their function briefly explained in context of the study (TCR etc.). Furthermore, common nomenclature includes dashes: PD-L1, PD-1, MHC-I.

We thank the reviewer for this suggestion and agree that more introduction is needed. We have provided more tumor immunology background within the beginning of the Results section of the text in the first two sections of the manuscript with the following text:

“Cell therapies have emerged as a transformative therapeutic modality, with T cell therapies resulting in impressive clinical outcomes^{1–3,33}. T cells achieve anti-tumor killing direct recognition of tumor antigen presented in the context of Major Histocompatibility Complex Class I (MHCI) through its cognate T cell receptor (TCR). Upon recognition they secrete a number of effector molecules including cytotoxic granules that cause death in the target cell locally. Consequently, T cells offer an attractive approach to tumor therapy because of their antigen-specificity, proliferation, and long-term memory that enables durable responses.

However, the effectiveness of T cell therapies has primarily been observed in hematologic malignancies with genetically modified chimeric antigen-receptor (CAR) T cells⁴, which constitute a minor fraction of cancer-related mortality (only 5% of cancer deaths³⁴). Furthermore, the broader implementation of T cell therapies has been hindered by systemic toxicities that limit their applicability³⁵. Ongoing endeavors are focused on optimizing T cell functionality through the modulation of T cell phenotype^{27,36–38}, designed to enhance capacity for self-renewal or killing^{31,32,39–42}.

In parallel there have been a number of immunotherapies developed to unleash endogenous, antigen-specific T cells within tumors⁴³. For example, some checkpoint blockade therapies block T cell inhibition with antibodies targeting inhibitory signaling molecule Programmed Cell Death Protein 1 (PD-1) found on T cells. Alternatively, therapeutic antibodies have been made to block PD-1's cognate receptor of Programmed Death-Ligand 1 (PD-L1), which tumor cells often express⁴⁴. Consequently, we hypothesized that the tumor cell phenotype could be just as crucial in shaping T cell responses within a tumor as the phenotype of T cells themselves."

Also, we added introduction in the second section of the Results:

"Controlling T cell T-cell phenotype during ex vivo expansion prior to therapeutic transfer is expected to be critical, especially since cells are in foreign environments for extended periods of time^{37,49–51}. Therapeutic T cell phenotype is known to cause dramatic differences in anti-tumor efficacy, especially from the perspective of T cell persistence and effector molecule expression^{31,32,39–42}. Broadly, memory T cells are expected to persist longer and give rise to more daughter cells, whereas effector cells are expected to be shorter-lived and secrete effector molecules like perforin^{52,53}. However, because T cells are usually isolated from subjects or dissociated from cancer tissues to be measured, the manners by which their phenotype relates to tumor phenotype, at the beginning and end of therapy, remain unknown. Indeed, clinical challenges and outstanding questions in targeting solid cancers are spatially related: e.g., T cell infiltration, local tumor antigen expression, and spatial co-enrichment with stimulating or inhibiting immune cells^{5–9,54–57}. Thus, elucidating how these spatial relationships and multicellular interactions change based on therapeutic T cell features, particularly cytokine and effector molecule secretion remains understudied."

* Supplemental Fig 2, B, D, F, & H are not particularly clear or useful without better descriptions in the caption. What are the colors, the axes, etc.

We have updated the descriptions within the captions for Supplemental Fig. 2:B,D,F, & H (that are now **Supplemental Fig. 3:B,D,F, & H**):

"Supplemental Figure 3: Representation and development of individual components of the multiscale agent-based model. A) T cell process is composed of two T cell states. PD1- CD8+ T cells become PD1+ T cells upon chronic stimulation and both PD1+ T cells and PD1- T cells can downregulate TCR. Both T cell types express TCR, IFN, and produce cytotoxic packets.

Molecular regulation is governed by activation and stimulation of tumor cells. B) Representative output of simulating only the T cell process. Each graph represents a specific molecule or molecular process that is being tracked within the simulation, and the first word describes the level in which that molecular process is connected to other pieces of the simulation. Starting with top-to-bottom and right-to-left, neighbors present PD1 measures the surface level ligand expression of the molecule PD1 on a T cell; internal total cytotoxic packets measures the total number of cytotoxic granules within a T cells over time; neighbors present TCR measures the surface level ligand expression of the T cell receptor on a T cell; internal refractory count measures the total number of times T cells have been activated and entered a refractory state post stimulation; neighbors transfer cytotoxic packets measures the total number of cytotoxic granules a T cells is outputting to tumor cells over time; internal TCR timer measures the total time that T cells have been activated for. C) Tumor cell process is composed of two tumor cell states. Ki67+ PDL1- MHC-I- tumor cells can become Ki67- PDL1, MHC-I+ tumor cells upon exposure to IFN. Both tumor cell types express IFNR. Molecular regulation is governed by interaction with T cells. D) Representative output of simulating only the tumor cell process. Starting with top-to-bottom and right-to-left, neighbors present PDL1 measures the surface level ligand expression of the molecule PDL1 on a tumor cell; internal IFNg measures the total number of IFN molecules a tumor cell has taken up over time; neighbors present MHC-I measures the surface level ligand expression of MHC-I on a tumor cell. E) The T cell compartment extends the T cell process by adding division and death processes that can be asymmetric. F) Representative output of simulating the T cell compartment. Starting with top-to-bottom, boundary PD1p divide count measures the number of times the PD1+ CD8+ T cells have divided in the course of the simulation; boundary PD1n divide count measures the number of times the PD1- CD8+ T cells have divided in the course of the simulation. G) The tumor cell compartment adds proliferation and death processes. H) Representative output of simulating the tumor cell compartment, such as boundary PDL1n divide count measures the number of times the PDL1- tumor cells have divided in the course of the simulation.”

* In Fig 2E, why is there a reduction in IFNg at t=9.7 hr when the numbers and positions of PD-1+ T cells and tumor cells are (more or less) the same across t = 5.5hr, 7.7hr and 9.7hr? Also, what caused the increase in IFNg at 13.5 hr?

In this in vitro based simulation, the T cells are all added to the tumor cells in the well at the same time. In our model T cells go through a refractory state where T cells downregulate their TCR and also IFNg expression and then

after a period of rest can be restimulated. This is also clarified within the model explanation document (see response to your comment #1).

* In Fig 3A: What do the different colored cells mean at d0 and d10? Provide a legend.

We have provided a legend for the differently colored cells at d0 and d10 in the figure now and clarified within the caption of Fig. 3A:

"A) Experimental layout for controlling T cell phenotype during ex vivo T cell expansion. Stimulating T cells in the presence of 2HC leads to a phenotypic shift, particularly in PD1 where lower PD1+ cells are found in 2HC-treated condition as denoted by blue, where T cells stimulated in the absence of 2HC have higher levels of PD1 and are denoted by orange."

* In Fig 3C, what do the gated 7.8% cells indicate? Additionally, is the 1.12% cells that indicate T cells always the case, or is this a function of the 'amount' of T cells transferred into the tumors?

The 7.8% cells indicate other CD45+ cells (immune cells) that are not CD8+. We have now added this to the figure legend:

"C) Percent of CD8+ T cells within tumors post-treatment with therapeutically expanded T cells determined by flow cytometry that are positive for both CD45 and CD8. CD45+ and CD8- cells (7.8%) represent non-CD8+ immune cells within the tumor."

The 1.12% is an average amount from the tumors; if we had transferred a higher amount of T cells likely we would observe more T cells based on prior experiments and literature reports. However, we have standardized all experiments with transferring 1 million CD8+ T cells at d10 into mice and now highlight this within the text:

"Therefore, to create an accurate starting ratio of tumor to T cells in the tumor microenvironments, we transferred one million therapeutic CD8+ T cells into each mouse bearing an established B16-F10 tumor that had been

grown for 10 days. Harvesting tumors after adoptive T cell therapy, showed that the CD8+ T cell frequency in these tumors was approximately 1% of all cells (Fig. 3C)."

* In Fig 3D; until t = 31.5hrs, are T cells present or not shown?

T cells are present, but because the simulation starts with ~1300 cells, the resolution of the images makes it difficult to "see" the T cells. This can be seen in the zoomed in feature of the **panel D of Figure 3**. These can also be better seen in the Supplementary Videos. We have clarified how T cells were initialized for experiments in **STAR Methods**:

"Initialization of Experiments: *The number of T cells and tumor cells as well as the proportions of phenotypes for each were based on experimental data. Briefly, most of the experiments started off with a total of 1200 tumor cells and 12 CD8+ T cells. For simulations in Figures 2 and 3, T cells were initialized randomly in the tumor bed. For simulations in Figure 5, T cells were initialized based on the locations of CODEX multiplexed imaging data. For simulations in Figure 6, T cells were located randomly inside or outside the tumor bed as specified."*

We have also clarified this in the Figure 3D caption:

"D) Snapshots of simulation initialized with in vivo-relevant cell numbers, ratios, and T-cell phenotypes for 25% and 75% PD1+ T cell conditions compared to a simulation condition with no T cells. All simulations were initialized with a total of 1200 tumor cells and 12 T cells with varying ratios of respective cell phenotypes."

* Figures 3E-H, what units of time? Also other subpanels need better axes labels

We thank the reviewer for catching this; it is in hrs. We have now added this to the axis labels and have specified other axis labels to make the data more understandable.

Minor points and typos:

* T cell / T-cell: hyphenate when T-cell is an adjectiveo E.g.: The T cells enter the system, and T-cell proliferation begins

We have now modified all T cell/T-cell references.

Manuscript Number: CELL-SYSTEMS-D-22-00549R1 - "Integrating Multiplexed Imaging and Multiscale Modeling Identifies Tumor Phenotype Transformation as a Critical Component of Therapeutic T Cell Efficacy"

Response to Referees

Overall Response

We thank the referees for their time and valuable feedback and positive evaluation of our manuscript "Integrating Multiplexed Imaging and Multiscale Modeling Identifies Tumor Phenotype Transformation as a Critical Component of Therapeutic T Cell Efficacy" (Cell Systems-D-22-00549). We carefully considered every comment raised by the referees and revised the paper to address the concerns raised by the referees. Our revised manuscript includes new analyses, simulations, and text that expand and clarify our original findings. We believe that the revised manuscript strengthens the confidence in our findings that T cells modulate tumor cell phenotype, this is critical for overall T cell efficacy, preserving T cell phenotype in the tumor microenvironment is a critical step while also balancing tumor engagement, and responses from distal immune organs.

In summary, in response to the referees' feedback we have:

- Provided further context within the Introduction and Discussion for the benefits, applications, and use-cases for both the type of modeling and specific model we were developing within this manuscript.
- Expanded our model parameter documentation and biological reasoning for parameter and model design both within the text and additions to the Supplemental table.
- Added additional simulations (6 new Figure panels) that help clarify and contextualize our original simulations and provide confidence to the overall biology we are studying while also improving the overall model design.

Our response is structured as follows:

- Response to Referee 1 (page 2)
- Response to Referee 2 (page 8)
- Response to Referee 3 (page 8)

Responses to reviewers and changes in the manuscript are denoted in blue color.

Reviewers' comments:

Reviewer #1: The authors have addressed the concerns raised in the previous review and made the necessary modifications accordingly. However, there are still several remaining questions and suggestions.

Two general questions:

1. While the tumor microenvironment is more complex and consists of various other cell types, it is acceptable to just focus on T cells and tumor cells as the primary cell types in the model to investigate their direct interactions in T cell therapy. However, the article does not provide a definitive conclusion on how to enhance the effectiveness of T cell therapy according to the results. One significant perspective highlighted in this article is the significance of the tumor cell phenotype, but how to utilize it in T cell therapy remains unclear.

We thank the reviewer for this helpful comment in making our findings more impactful. We have added additional detail and specific clarification surrounding how the significance of our findings could be used to enhance T cell efficacy. We have included some additional text to 2 paragraphs in the Discussion (*italicized=new*) below:

"The results indicated that tumor phenotype considerably influences the ability of T cells to control tumor cell growth through inhibiting proliferation and increasing killing. The importance and magnitude of a tumor phenotype change was only clear after analysis of the multiscale model and was not intuitive from the multiplexed imaging data alone. Most recent work has focused on controlling T-cell phenotype for both the secretion of killing molecules and self-preservation^{31,32}. We found that T-cell phenotype also impacts its ability to change tumor phenotype, and a focus on converting tumor phenotype results in greater control than minimizing T cell exhaustion. Based on our findings, T-cell therapies should be designed with a profile capable of concurrently inhibiting tumor cell proliferation, enhancing the inflammatory state of tumor phenotype, initiating tumor killing, and sustaining T cell longevity and efficacy. Various methods have previously been employed to modify T cells to achieve each of these objectives^{27,52,67,68}. However, what has been lacking is the ability to comprehensively investigate how alterations in T-cell phenotype impact all these parameters simultaneously. With a new goal, there will arise a plethora of new strategies to modify T cells that arise to accomplish this objective, but an immediate example next step could be to engineer T cells to secrete cytokines and observe whether they can produce anti-proliferative and inflammatory effects while also preserving T cell longevity."

"This incongruence motivated parallel research on T-cell therapies, where we observed that therapeutic T-cell phenotype changes the structure and cellular composition of the tumor microenvironment⁵⁹. We found in this other work that T cells create distinct multicellular neighborhoods based on their phenotype and molecular expression profiles. For example, 2HC T cell treated tumors result in more productive T cell and tumor neighborhoods, whereas T cells not treated with 2HC secrete more anti-inflammatory cytokines and have T cell and tumor areas also enriched with regulatory neighborhoods. Thus, T cells should be engineered to be agents of structural change of the tumor microenvironment in addition to transforming tumor cell phenotype."

2. It appears that the many findings of this article could also be investigated directly through in vitro or in vivo experiments, complemented by spatial technologies like CODEX. Additionally, conclusions derived from computational models typically require further experimental validation. Therefore, **it is crucial to clearly articulate the distinct advantages and usage of this model compared to conventional experiments to demonstrate its value.** It is somewhat confusing in this article as the results of conventional experiments were utilized both as input for the model and for its validation.

We agree that it is important to articulate the distinct advantages and usages of this model. With our new text to the Introduction and Discussion sections of the manuscript we highlight the integration of shared spatial features between the data and how critical the dynamic feature of the multiscale modeling allows us to either extend the time of a specific sample, or perform hypothetical perturbations of the samples. This is particularly important when we think about patient samples, where time-dependent validation of dynamic information will not be possible at the resolution and depth of information acquired by multiplexed imaging. Furthermore, it was only after we ran multiscale modeling simulations that enabled quantification of the dynamics of tumor growth rates did the magnitude of the effect of a tumor phenotype change become clear, and not from the multiplexed imaging data. We have added additional statements in the Discussion surrounding this as well.

Introduction

"This marriage of multiscale modeling and multiplexed imaging share key data-driven features across scale, particularly the spatial positioning of distinct cells and molecules. Consequently, information from multiplexed imaging feeds multiscale agent-based models by providing more accurate parameter values, initial states (e.g., cell types and positions), and update rules. Multiplexed imaging data also represents a singular snapshot captured from valuable experimental or patient samples. Continuous monitoring at the individual cell level with similar detail is currently unfeasible. However, multiscale modeling presents an opportunity to augment our data, enabling the exploration of dynamic behaviors and the conduct of hypothetical experiments. For example, starting with a biopsy or tissue section, we can examine how different therapeutic approaches will play out."

*"By combining multiplexed imaging and multiscale modeling, we demonstrated that both tumor and T-cell phenotype are key determinants of T cell therapeutic efficacy. T-cell phenotype control has been a main focus to promote T cell longevity for killing cancer, with most approaches centering on intracellular molecular perturbation of T cells^{31,32}. Much less attention has been given to tumor phenotype. Here we observed that the conversion of tumor phenotype was a critical determinant in the control of cancer growth. Tumor phenotype conversion was dependent on a CD8+ T-cell phenotype with ability to divide rapidly (memory-like) and secrete IFN (effector-like), suggesting this as a design criterion/goal for T cell therapies as well as a matching patient selection metric. *The results suggest that integrating a multiscale modeling approach with multiplexed imaging data can provide a roadmap towards such a goal and establish it as a system for extending the dynamics of multiplexed imaging data.*"*

Discussion

"Adding dynamics to project behavior from static multiplexed imaging datasets, captured at a single time point, is critical. While we were able to compare our model through collecting multiple time points in a mouse model, most often this will not be the case because the majority of collected multiplexed imaging data involves invaluable human biopsies, surgical resections, or donor tissue obtained at a single time point from both healthy and diseased tissues^{15,66}. While multiplexed imaging offers insights into cellular interactions across space, it lacks the ability to reveal the temporal evolution of these interactions. This limitation hinders our capacity to make predictions or conduct hypothetical experiments with authentic patient data, which is inherently heterogeneous. Establishing a framework for integrating data and multiscale modeling here serves as a foundational step. It provides a starting point to construct models that deep multiplexed imaging based biological data, thus extending our understanding of cellular dynamics of tissues. This approach will enable us to comprehend

how interactions unfold over time and will offer valuable insights into identifying effective therapies for specific patients."

"The results indicated that tumor phenotype considerably influences the ability of T cells to control tumor cell growth through inhibiting proliferation and increasing killing. The importance and magnitude of a tumor phenotype change was only clear after analysis of the multiscale model and was not intuitive from the multiplexed imaging data alone. Most recent work has focused on controlling T-cell phenotype for both the secretion of killing molecules and self-preservation^{31,32}. We found that T-cell phenotype also impacts its ability to change tumor phenotype, and a focus on converting tumor phenotype results in greater control than minimizing T cell exhaustion."

Additional minor points:

1. In this article, the authors emphasized the importance of spatial positioning for T cells and tumor cells. Apart from the initial position setting, other parameters, such as migration parameters, also play a vital role in this process. It is crucial to provide a rationale for these parameters under varying conditions. In the second part of the Results, the authors kept the ratio of tumor to T cells consistent with the *in vivo* mouse model but did not make any adjustments to other parameters.

Establishing parameters and making simplifying assumptions is a bedrock of modeling. Our goal was to understand how the T cell phenotype affects tumor growth in a spatial environment. Based on these goals, we derived our parameters either from lab-derived values that we were using in the *in vitro* and *in vivo* models of T cell therapy our group has been studying or from relevant literature. Particularly, important parameters such as the migration of the T cells are based on dynamic data taken from the literature, and change according to the biological context of the T cells. For example, there are already changes in T cell migration integrated into our model. For instance there are differing speeds for PD1- and PD1+ T cells and when T cells migrate within the tumor and encounter tumor antigen, slow down their migration speed to interact with MHC1+, antigen-presenting tumor cells. These parameters are based on direct *in vivo* real-time imaging data taken from T cell migrating in tumors from either intravital or 2-photon microscopy^{1,2}. This is just one example of additional parameters that are being changed during the simulation based on biological inputs. To clarify this for the reader we added the following explanation to the Results section where we introduce the model:

*"To create this model required encoding prior knowledge and lab-derived parameter values to create the rules governing individual cancer cells and T cells (see Supplemental Table 2, Supplemental Fig. 2, and 3). These parameters include data sourced from both deep molecular and time-resolved dynamic interactions of T cells and tumors. For example, in our model, T-cell migration reflects observed physiological changes, with distinct velocities based on biological input within the model, differing for PD-1- and PD-1+ T cells and whether T cells are engaging with tumor cells. Specifically, T-cell motility is decreased upon encountering MHC1+ antigen-presenting tumor cells. These modeled behaviors are informed by empirical *in vivo* imaging, utilizing techniques such as intravital and 2-photon microscopy to track T-cell dynamics within tumors^{45,49}. Such empirical grounding ensures our model parameters accurately simulates the biological activity of T cells in the tumor microenvironment."*

We did make simplifying assumptions such as not accounting for ECM data or chemokine-mediated migration. We did this particularly because we did not collect this information (ECM distribution, chemokine distribution) spatially from multiplexed imaging, but could be collected in future experiments.

Also based on these goals, we only changed factors that were surrounding T cell phenotype in our simulations. To make this more clear we have now provided a "Rationale" column with our parameter table in the Supplementary Information section that is in addition to the name and value in the model, whether it was sourced from data or primary literature, and the location within the source code. For example, only including a screenshot of part of the table (because of length):

Model name	Value	Source	Location	Rationale/Explanation
bounds	Vary	Multiplexed Imaging Data	main.py	Size of tumor
Depth	15 μm			Based on an average single cell diameter
n_tumors	Vary			Number of tumors found
n_tcells	Vary			Number of T cells found
n_dendritic	Vary			Number of T cells found
dendritic state active	Vary			Number of activated dendritic cells found
diameter	7.5 μm	Multiplexed Imaging Data	t_cell.py	Diameter of average T cell
initial PD1n.	Vary	Multiplexed		Based on percentage

1] Boissonnas, A., Fetler, L., Zeelenberg, I.S., Hugues, S., and Amigorena, S. (2007). In vivo imaging of cytotoxic T cell infiltration and elimination of a solid tumor. *J. Exp. Med.* 204, 345–356.

2] Thibaut, R., Bost, P., Milo, I., Cazaux, M., Lemaître, F., Garcia, Z., Amit, I., Breart, B., Cornuot, C., and Schwikowski, B. (2020). Bystander IFN- γ activity promotes widespread and sustained cytokine signaling altering the tumor microenvironment. *Nat. Cancer* 1, 302–314.

2. The authors incorporated a substantial amount of experimental data, such as the spatial information from CODEX, to initialize the model and enhance its realism. However, the model still lacks dynamic experimental data to validate its accuracy, particularly regarding the selection of parameters.

We have acquired both dynamic in vitro and in vivo experimental to validate its accuracy. Particularly, **Figure 2C** is an in vitro killing assay with T cells incubated with tumor cells and monitored dynamically (minute-scale) on how fast T cells kill tumors. We have emphasized this in the Results section of the manuscript with adding the following text:

"Quantification of killing by cytotoxicity indicated an important role of tumor phenotype on T-cell killing even at early time points (Fig. 2G). Moreover, when tumor cells were pre-treated with IFN- γ , we observed a remarkable increase in cytotoxicity, reaching nearly 60% by 12 hours compared to untreated tumor cells incubated with an equivalent number and type of T cells. This finding mirrors the observed increase in our in vitro experiment's cytotoxicity (Fig. 2C). Consequently, our simulation results not only replicated the kinetics and magnitude of our in vitro killing data but also reinforced the overall conclusion. *This alignment validates both our model setup and molecular parameters with dynamic data.*"

Similarly, we performed an in vivo experiment where we collected tumors from either one or three days post treatment from mice treated with different T cell treatments as shown in **Figure 4**. To stress the value of this validating data, we have added additional details to the Results section of the text:

"We also compared the cell-type percentages in the CODEX data with in silico percentages at day 3 to understand whether relative phenotype conversion rates were similar. We found good correlations of ending percentages for PDL1+ tumor cells (Figure 4F, R=0.99), PDL1-tumor cells (Figure 4G, R=0.97), and PD1+ T cells (Figure 4H, R=0.99). Similarly, we also tested the relationship of the cell populations at day 1 from our simulations compared to in

vivo data from our CODEX imaging and saw good correlation in cell type frequencies (Supplemental Fig. 6B). *This indicates our model is capturing appropriate cellular dynamics and relationships of the tumor cells in vivo.*"

While we do not have direct *in vivo* imaging of T cells within live tumors of animals (this would require an intravital or multiphoton set up), we have sourced many of our dynamic parameters from studies that have done exactly these types of studies evaluating migration rates of T cells. Since this was not clear, we have added the following text to our Results section of our manuscript.

*"To create this model required encoding prior knowledge and lab-derived parameter values to create the rules governing individual cancer cells and T cells (see Supplemental Table 2, Supplemental Fig. 2, and 3). These parameters include data sourced from both deep molecular and time-resolved dynamic interactions of T cells and tumors. For example, in our model, T-cell migration reflects observed physiological changes, with distinct velocities based on biological input within the model, differing for PD-1- and PD-1+ T cells and whether T cells are engaging with tumor cells. Specifically, T-cell motility is decreased upon encountering MHC1+ antigen-presenting tumor cells. These modeled behaviors are informed by empirical *in vivo* imaging, utilizing techniques such as intravital and 2-photon microscopy to track T-cell dynamics within tumors^{45,49}. Such empirical grounding of the model parameters ensures its simulation accurately captures the biological activity of T cells in the tumor microenvironment. We additionally encoded intracellular and intercellular interactions in Vivarium and tuned the parameters by comparing process performance with expected behavior standards."*

3. It is commendable to incorporate additional cell types into this model. However, there is a **lack of detailed description regarding the inclusion of DCs and their interactions with T cells**. It is crucial to consider specific molecular interactions such as MHC-I or MHC-II with TCRs, as well as the involvement of costimulatory or coinhibitory molecules with their respective receptors when exploring the interactions between DCs and T cells.

We agree that the description of the dendritic cells is important for the reader to understand the interactions that are added with its inclusion. Based on this, we have added additional text to the Results section to describe the key functions and interactions of the dendritic cells in the model and have updated Figure 7 to have an overview figure of the dendritic cell and T cell interaction in Figure 7A:

"We speculated therefore that T cells could come from outside the tumor enabling founder T cells to reside in supportive microenvironments like the lymph node (LN), while daughter cells migrate into the tumor. We therefore extended our model by adding another cell type of the dendritic cell to our model. Dendritic cells are critical antigen-presenting cells that take up antigen and stimulate antigen-specific T cells in the LN. T cells then divide and leave the LN for effector function in the target tissue. We built our model system after these principles, leveraging the same approach we took of literature and lab-derived parameters (Fig. 7A). In particular, we encode the ability for dendritic cells to uptake tumor antigen from dying tumor cells within the lymph node, become activated by apoptotic debris, and migrate to the lymph node. We also add into our model that some of our transferred antigen-specific T cells trafficked and reside within the tumor-draining lymph node, encounter dendritic cells, are activated by MHC1+ with tumor antigen expressed by the dendritic cells and proliferate before leaving the lymph node for the tumor microenvironment."

The addition of co-inhibitory and co-stimulatory molecules as the reviewer suggests will be additionally important to add to the model in future generations of the model. We did not include that level of detail with the dendritic cells in this model for a few reasons. The main reason is that our multiplexed imaging data

contains data surrounding the frequency and location of dendritic cells in the tumor and lymph node, but not whether they also contain co-inhibitory or co-stimulatory molecules. In addition, there is a pleiotropy of co-inhibitory and co-stimulatory molecules, lack of research about what occurs with different combinations on the surface of dendritic cells, lack of consensus about each function in interacting with T cells, and little understanding of the dynamic regulations of these within our tumor model. It would be prohibitive in both cost and practice to measure all such molecules simultaneously using CODEX multiplexed imaging, while also capturing the detailed information we desired about other cell types, T cell phenotypes, and tumor phenotypes that were the focus of this study and model. Because of the lack of data and consensus in the field on the different types of molecules, we made the simplifying assumption that dendritic cells would act to activate CD8+ T cells. As the technology, our data, and the model evolve this would be something we would want to expand.

4. In the final section of the Results, the authors introduced the concept of the lymph node (LN) as a supportive microenvironment for T cells outside the tumor. While the focus was primarily on the tumor regions in silico, the mouse model emphasized the tumor-draining LNs. It remains unclear how the characteristics of T cells in the tumor region differ between the mouse model with 25% PD1+ T cells and 75% PD1+ T cells, and whether these differences align with the in-silico results.

We have extensively characterized the tumor microenvironment also with the CODEX multiplexed imaging. In brief, the tumor microenvironment is very similar for both the 25% PD1+ T cells and 75% PD1+ T cells. The major differences between the 25% PD1+ T cells and 75% PD1+ T cells tumors have a reduced conversion of tumor phenotype to an inflamed phenotype from a proliferating phenotype, and smaller areas of inflamed (PDL1+ MHCII+) tumor cell and T cell areas next to dense immune infiltration. In comparison to our simulated data, this agrees well with overall numbers of cells. We present these comparisons of simulated versus CODEX data in Figure 4F-I and also in the organization of such simulated neighborhoods which we compared in Figure 4A-E. The neighborhoods of the CODEX data and additional descriptions of the tumor microenvironment can be found in our companion paper¹. We add this description to the Discussion section of the manuscript to make this more explicit:

"Comparing in silico results to CODEX multiplexed imaging reinforced the importance of T-cell phenotype and influence on tumor phenotype. Our simulations indicated agreement both in terms of percentages of different cell types and phenotypes and also in the organization of these cells with respect to each other. For example, inflamed tumor cells were found proximal to T cells within both simulation and CODEX imaging results."

"We found in this other work that T cells create distinct multicellular neighborhoods based on their phenotype and molecular expression profiles. For example, 2HC T cell treated tumors result in more productive T cell and tumor neighborhoods, whereas T cells not treated with 2HC secrete more anti-inflammatory cytokines and have T cell and tumor areas also enriched with regulatory neighborhoods."

While there is general agreement in overall results there are additional details such as the presence of CD4+ T regulatory cells we observe enriched in 75% PD1+ T cell treated tumors that we currently do not account for within our model. We describe that this is a future direction of the model to continue to add additional cell types to fully utilize the CODEX imaging data to paint a more complete picture of the tumor microenvironment:

"In future work, driven by molecular data acquired through multiplexed tissue imaging, the model should be expanded to include other cell types and anti-inflammatory molecules. Since the Vivarium framework for multiscale modeling is modular and compositional, it will be straightforward to add additional cell types with different representations of internal

mechanisms and environmental interactions, such as how we added the dendritic cell and lymph node process."

1. Hickey, J.W., Haist, M., Horowitz, N., Caraccio, C., Tan, Y., Rech, A.J., Baertsch, M.-A., Rovira-Clavé, X., Zhu, B., and Vazquez, G. (2023). T cell-mediated curation and restructuring of tumor tissue coordinates an effective immune response. Cell Rep. 42.

Reviewer #2: The reviewers' comments were adequately addressed in the manuscript. There was one statement, however, that was confusing: "Notably, we observed that for IFN γ -treated tumor there is a reduction in cells that were quiescent or in the G0 phase of the cell cycle (negative for IdU and pRb (S807 S811))" Should this statement be that there was an increase, not a reduction of quiescent cells following treatment?

Thank you for catching this error, you are correct. We have now fixed this in the text with the following:

*"Notably, we observed that for IFN-treated tumor there is an **increase** in cells that were quiescent or in the G0 phase of the cell cycle (negative for IdU and pRb (S807 S811)) (Fig. 2B)."*

Reviewer #3: The revision by Hickey et al. has greatly improved the presentation, content, and interpretability of the paper, and the results have become more interesting as a result. I commend the authors in their thorough efforts to address the reviews and in the process creating an excellent paper. I have a few remaining questions regarding the results, now that the methods and interpretations have become clearer.

In general, the model is highly complex, and **it can be hard to distinguish intuitively what salient features are emergent behavior due to spatiotemporal dynamics, and what features are the result of potential artifacts of model parameterization and initialization. It would be useful then, to examine a couple of aspects of the results where the distinction is not clear.**

First, Figures 3H, 5C, and 7C show the T-cell subtypes. In all three cases, representing outputs of the model with different situations, the 25% and 75% curves show the sharp transient of phenotypic switching at what appears to be the same time. It is unclear why this should be the case. There is some slight variation, but invariably around 45-48hrs, everything happens. In a stochastic simulation I'd expect more variation in when such transitions occur, unless there is some "timer effect" based on T-cell function that starts counting from the start of the simulation? If there is a "timer" effect, then the question is, would such a timer naturally be in place if the simulation were started "a few days earlier"? The point here is that these tumor and immune cells are in positions and states that have a history. How much of this transient behavior then is due to initialization?

Thank you for carefully reviewing our paper and for helping us make our model assumptions and biological context clearer. There is a sharp switch in phenotype for Figures 3H, 5C, and 7C. Here we will break down the multiple questions and resulting changes to the manuscript that we have taken.

To answer the question of whether there is a "timer effect":

We did have two timers within our model. We encoded the T cells on a timer of activation and refractory periods so we could specify length of T cell activation/engagement and refractory periods based on these processes described in the literature¹⁻³. Based on the reviewer's comment we also modified the code to replace the activation/refractory timer with a stochastic function called "probability_of_occurrence_within_interval" described in the supplemental text, that we have previously used for other processes that have a known expected behavior within a certain time. Based on these simulations, we can confirm the sharp decrease was due to setting a direct timer, though we still observe transitions in T cell phenotype transition and also associations with conversion of tumor phenotype consistent with our previous results. We now include these results within the text (italicized and quotes) as New Figure 7E and Supplemental Fig. 9E-F:

"While there is a sustained supply of T cells in the tumor microenvironment following incorporation of the lymph node process (Fig. 7B), there is a rapid phenotypic conversion for many T cells around 48 hours post-initialization (Supplemental Fig. 9A). Despite uniform treatment and exposure to an identical number of stimulations across all cells, the stark nature of this phenotypic shift appeared to be influenced by the deterministic mechanisms of T-cell activation and the fixed refractory periods in the model we initially established. Upon modifying these timers to stochastic events, we noted a more gradual shift in T-cell phenotype within the 25% PD-1+ T cell treated condition (Fig 7D). This observation suggests a potential delay in the transition to PD-1+ T cells within the tumor microenvironment. Nonetheless, a decline in PD-1- T cell numbers commenced around the 50-hour mark, due to established biological mechanisms where repeated T cell stimulations lead to exhaustion⁶⁰⁻⁶². Also, consistent with our previous results, the 25% PD-1+ T cells exhibited sustained efficacy in suppressing tumor growth (Supplemental Fig. 9E), due to their enhanced capacity for early conversion of the tumor phenotype (Supplemental Fig. 9F)."

New Figure 7E: Comparison of a probabilistic-based T cell activation and refractory timing versus using set timers within the simulation for the 25% PD1+ condition.

New Supplementary Fig. 9E-F: In running the same simulations but with probabilistic timers for T cell activation and refractory periods, we observe similar trends in tumor control by the T cells through earlier conversion of the tumor cell phenotype.

To answer the question surrounding the initialization states of the T cells:

The reason we initialize all cells with a similar activation state is based on three biological reasons. First, this is due to the therapy we are modeling: adoptive T cell therapy in cancer. This treatment includes taking out immune cells from a patient and expanding them to high numbers where most all cells are antigen-specific (or editing them so that they are), lymphodepleting the patient (depleting endogenous T cells), and then transferring all synchronously activated, antigen-specific T cells back to the patient intravenously at once. Consequently this is not modeling a "native" immune response to the tumor, with varying T cell initialization states that had once pre-existed in the tumor, rather modeling T cells that had been activated and added in concert.

Second, phenotype transition in our model is based on literature-derived mechanisms of repeated stimulation of the TCR will lead to exhaustion, decrease in cytokine production, and cell division shown through *in vitro* expansion of T cells¹. This is something that we also observe to occur at similar rates in our own *in vitro* data. Because T cells all start with a similar stimulation profile (initialization based on *ex vivo* stimulation), then the parts of the simulation that are stochastic are the potential interactions with tumor cells within the tumor microenvironment. This includes stochastic localization of the T cells, migration of the T cells, and interaction duration of the T cells with the tumor cells. Since each tumor cell expresses the cognate antigen gp100 and each tumor cell is antigen-specific, then the chances of T cells interacting with tumor cells is high, which will lead to similar changes in overall T cell phenotype change within the tumor with respect to time.

Third, even with similar activation conditions, we characterized the T cells post-activation. For different T cell culture conditions, we have profiled these cells post activation and based on the treatment (whether or not the cells were cultured with 2HC) in Figures 2B, C, Supplemental Figure 4, and in our companion manuscript⁴. This allows us to initialize the different treatment groups appropriately with the percentages of cells with different initial phenotypes.

Based on these three reasons we have added the following text to describe in greater detail the biology we are modeling and reasons for initialization states we set for the T cells:

"The improved tumor control was linked to increased numbers of PD-1- T cells in the tumor, though in both conditions T cells became PD-1+ over the course of the simulation (Fig. 3H, Supplemental Fig. 5B). *This is not surprising, since it is known that repeated stimulation of*

antigen-specific T cells leads to exhaustion which includes phenotype changes resulting in lower cytokine secretion and cell division^{60–62}. Similarly, since we mimic the design of our in vivo adoptive T-cell therapy experiments, all CD8+ T cells follow the same course ex vivo. Briefly, we use transgenic PMEL mice, wherein all CD8+ T cells are specifically reactive to the melanoma-associated antigen gp100, expressed by the B16-F10 tumor cell line. We harvest the immune cells from PMEL mice and activate these cells for 10 days in vitro with cognate antigen and anti-CD3 activation, upon which the cells undergo considerable proliferation. The stimulatory regime for these transgenic, antigen-specific T cells, results in a relatively homogeneous activation state across the cell population, allowing for the initialization of treatment groups (e.g., with or without 2HC) with accurate phenotypic proportions that we previously characterized by CyTOF (Fig. 3A-C). Similarly, we inject B16-F10 tumor cells to recipient wild-type mice and allow them to grow for 9 days, lymphodeplete on day 9 with sublethal irradiation (mirroring clinical practice for adoptive therapy), and transfer T cells into recipient mice at day 10. Since all T cells are antigen-specific and all tumor cells express antigen, there is a high probability of interaction between antigen-specific T cells and gp100-expressing tumor cells within the tumor microenvironment. Consequently, we would expect a phenotype shift from the T cells that we transferred into the tumor. However, what was not expected was that the phenotype of the T cells would influence the ability to convert tumor phenotype and that this would play such a critical role in controlling the tumor growth rate. Thus, greater tumor control from phenotype-switched T cells was due to greater ability to inhibit tumor proliferation rather than differences in inhibition from T cell direct killing."

Finally, the observation of all cells becoming exhausted by day 3 in our multiplexed CODEX imaging data is what led us to hypothesize that some of the T cells that were transferred in could reside within the lymph node and "escape" the synchronicity of tumor antigen stimulation and activation and provide an additional source of non-exhausted T cells to the tumor microenvironment. Indeed, this is what the new Supplemental Fig. 9A shows; that while there is a synchronous change in phenotype of T cells that arrived at the tumor together when transferred in, there is a continual supply of non-exhausted T cells in the long term.

This is an important point that we want the readers of the manuscript to understand, so we have made substantial additions (italicized) to the Results section to explain this behavior:

"We speculated therefore that T cells could come from outside the tumor enabling founder T cells to reside in supportive microenvironments like the lymph node (LN), while daughter cells migrate into the tumor. We therefore extended our model by adding another cell type of the dendritic cell to our model. Dendritic cells are critical antigen-presenting cells that take up antigen and stimulate antigen-specific T cells in the LN. T cells then divide and leave the LN for effector function in the target tissue. We built our model system after these principles leveraging the same approach we took of literature and lab-derived parameters, which we used in building the previous model system (Fig. 7A). In particular, we encode the ability for dendritic cells to uptake tumor antigen from dying tumor cells within the lymph node, become activated by these apoptotic debris, and migrate to the lymph node. We also add in our model that some of the transferred antigen-specific T cells move to and stay in the tumor-draining lymph node. There, they encounter dendritic cells and are activated by MHCI+ with tumor antigen expressed by these dendritic cells. After this activation, they proliferate before leaving the lymph node to the tumor microenvironment."

We compared simulations with 25% PD1+ T cell treatment conditions initialized with and without the extra LN process that incorporated dendritic cells. There were increased and sustained numbers of T cells over three days of simulation time in the conditions where dendritic cells were able to activate T cells in LN (Fig. 7B). This came from added numbers of PD1- T cells to the tumor from the LN (Fig. 7C, Supplemental Fig. 9A). This result more

closely matches our CODEX multiplexed imaging results of tumors (Fig. 4), where a subset of the T cells within the tumor are found to be PD1- even several days after adoptive transfer. *Particularly, this reduces the complete switch in phenotype from PD1- to PD1+ we observed in previous simulations such as Figure 3H, further suggesting the importance of both preserving T cell phenotype and also multiple waves of non-exhausted T cells in control of tumor growth.*"

Also text we added to the Discussion:

"This led us to add dendritic cells and a connection to lymph nodes within our model system, which suggested a sustained source for non-exhausted T cells in the tumor microenvironment. *This result indicates that while it is important to design T cells that traffic to the tumor, it may be just as important to also have a subset that will traffic to tumor-draining lymph nodes to be a constant supply of T cells over time or additive effects of continual additions of T cells over time.*"

While we are modeling the T cell therapy of tumors here, our model could be modified to test endogenous tumor responses as well. This would require modification of initial conditions of the T cells and establishing some of the T cells as not antigen-specific yet would still utilize all the underlying biological mechanisms we have coded into the model. We would expect less synchronous behavior of the T cells in this condition like the reviewer suggests, since there are cells with histories and consequently a lot more heterogeneity within the timing of T cell phenotype switching.

References

1. Zhao, M., Kiernan, C.H., Stairiker, C.J., Hope, J.L., Leon, L.G., van Meurs, M., Brouwers-Haspels, I., Boers, R., Boers, J., and Gribnau, J. (2020). Rapid in vitro generation of bona fide exhausted CD8+ T cells is accompanied by Tcf7 promoter methylation. *PLoS Pathog.* 16, e1008555.
2. Gallegos, A.M., Xiong, H., Leiner, I.M., Sušac, B., Glickman, M.S., Pamer, E.G., and van Heijst, J.W.J. (2016). Control of T cell antigen reactivity via programmed TCR downregulation. *Nat. Immunol.* 17, 379–386.
3. Salerno, F., Paolini, N.A., Stark, R., von Lindern, M., and Wolkers, M.C. (2017). Distinct PKC-mediated posttranscriptional events set cytokine production kinetics in CD8+ T cells. *Proc. Natl. Acad. Sci.* 114, 9677–9682.
4. Hickey, J.W., Haist, M., Horowitz, N., Caraccio, C., Tan, Y., Rech, A.J., Baertsch, M.-A., Rovira-Clavé, X., Zhu, B., and Vazquez, G. (2023). T cell-mediated curation and restructuring of tumor tissue coordinates an effective immune response. *Cell Rep.* 42.

Similarly, Figure 5E feels a little odd in terms of the initialization. In panel A, on the right, the 75% and 25% sections chosen appear reasonably similarly chosen, in that it captures a section of the tumor and also a little bit of the edge. However, in panel E, why are these initialization pieces set in a larger domain, and more to the point, in the lower left corner? In particular, I wonder how the fact that in 25%, the edge is exposed to the open domain above, whereas in 75%, the edge of the sample is trapped down in the corner. Does this

matter? It would be interesting to see that same simulation, but starting with the sample cells in the center of the open domain, allowing growth in all directions. Equally interesting would be starting these sample initial conditions in a narrow domain where the edge of the sample had open domain to grow into, but the three other sides of the sample were constrained by the edge of the domain (because in the larger original sample, they were inside the tumor and therefore constrained).

This is a great point and one that originally led us to hypothesize that why the location of T cells matters for preserving their phenotype, where we stated in the Results section:

"In contrast, the T cells in the in silico 75% PD1+ T cell condition initially attacked from the periphery but were soon surrounded by proliferating cancer cells seen at 30.8 h and remained so until 72 h (Fig. 5E, orange square). This suggests that the spatial location of T cells on the periphery of tumors may be critical for T-cell phenotype maintenance."

We do believe that these initializations have this effect on overall outcome. Consequently, per your suggestion we also performed the experiment where we start the simulations in the center of the frame for the 75% PD-1+ T cell condition. We also found that counterintuitively the condition where T cells can escape there are increased T cells over a longer period, but an increased tumor because of a delayed induction of tumor phenotype conversion. We have added new Supplemental Fig 8D-F and text to the results section:

"Consequently, delay of T-cell exhaustion came at the cost of enhanced early tumor growth rates (Fig. 6E), which even larger numbers of T cells were not able to overcome in the long run. Thus, trading less tumor engagement over time for an increased number of T cells in the future essentially gives the tumor a head-start in proliferation enabling exponential growth rates that extra T cells are not able to counteract through slightly enhanced killing later (Fig. 6F, Supplemental Fig. 8C). We confirmed this result by setting the 75% PD-1+ T cell condition from Figure 5 to start in the center, preventing T cells from becoming trapped (Supplemental Fig. 8D). Here we also observe an increase in the tumor growth rate for the center condition that the T cells are able to escape the tumor microenvironment and preserve phenotype but limit the earlier tumor conversion (Supplemental Fig. 8E-F). Since resting preserves phenotype, but does not enhance tumor outcome, a mechanism for enhanced tumor control in metabolically treated T cells is missing from our model."

New Supplemental Fig 8D-F: Experiments that have been initialized with CODEX data placed in different conditions of the environment. D) 75% PD-1+ condition is either initialized in the corner or the center of the spatial environment. E) Quantification of number of tumors shows that conditions where simulation is initialized in the center grow more quickly than those in the bottom due to increased T cell interactions which leads to F) a delayed exhaustion of T cells in the center initialization consistent with previous results.

Overall, this is truly an excellent integrated paper and I cannot thank the authors enough for fully embracing and expanding upon my previous review (and that of the other reviewers). If possible I'd love to see some exploration of the above points to ensure that the behaviors shown are not partially due to artifacts of the initialization.

We thank the reviewer for their enthusiasm for our work, careful reading, and excellent comments and suggestions that have improved the manuscript both in the data and presentation of this data.

Minor points and typos:
* "achieve anti-tumor killing direct recognition", missing "via" or similar

Thank you for catching this. We have fixed this to:

"...achieve anti-tumor killing via direct recognition..."

* The issue with T-cell hyphenation is still inconsistent: T-cell therapy, T-cell functionality, T-cell inhibition, T-cell persistence, etc.

We changed the remaining instances where T-cell is an adjective.

* Similarly, PD-1 and PD-L1 are often lacking the proper hyphens

We have replaced all instances of PD1 with PD-1 and PDL1 with PD-L1.

* Minor aesthetic detail, but in Fig 3D, the little inset window at the end of second row isn't positioned over the tumor in the way the large inset view would indicate.

Thank you for catching this and the careful reading of our manuscript. We have updated the figure inset accordingly.

Manuscript Number: CELL-SYSTEMS-D-22-00549R1 - "Integrating Multiplexed Imaging and Multiscale Modeling Identifies Tumor Phenotype Transformation as a Critical Component of Therapeutic T Cell Efficacy"

Response to Referees

Overall Response

We thank the referees for their time and valuable feedback and positive evaluation of our manuscript "Integrating Multiplexed Imaging and Multiscale Modeling Identifies Tumor Phenotype Transformation as a Critical Component of Therapeutic T Cell Efficacy" (Cell Systems-D-22-00549). We appreciate the effort to help us improve our manuscript and strengthen our findings. We have responded to the comments below in blue.

Reviewer comments:

Reviewer #1: The latest manuscript has effectively addressed the concerns raised by the reviewers. This article introduces a method to improve our understanding of the interactions between T cells and tumor cells in terms of spatiotemporal dynamics. Several intriguing ideas have been proposed. For example, it is suggested that the conversion of tumor phenotype may have a more significant impact than direct killing, and that external supplementation of T cells is crucial and necessary for maintaining the T cell phenotype in the tumor microenvironment. The results have also prompted further questions that require additional exploration. It was observed that PD-L1+ tumor cells inhibit T cells through the interaction between PD-1 and PD-L1, but at the expense of their own expansion ability. No differences were observed within the lymph nodes of different conditions, which may be due to the expansion preference of T cells with different phenotypes in lymph nodes or other functional structures in the tumor microenvironment, such as tertiary lymphoid structures. Only some minor questions and suggestions:

1) In the Figure 2D model, there were multiple modes of cell death, indicating that the number of cell deaths did not correspond to T cell killing. This requires correction for deaths due to other reasons, such as apoptosis.

Yes, we believe that this is a feature of our model and we have accounted for this in our analysis such as Supplemental Fig. 5A, 6C, 7B, 8C, 9D.

2) In the third paragraph of the last section, should the statement be "The sustained levels of T cells and percentages of PD-1- T cells within the tumor over long periods of time had a drastic impact on the total number of tumor cells past 36 hours of simulated time (Fig. 7C)"?

Thank you for catching that. We have added "percentages" based on your suggestion to this sentence to clarify.

3) This article solely focused on the function of IFN- γ on tumor cells. However, IFN- γ secreted by T cells can also affect T cells themselves. The authors should take this into consideration if they further develop the model later. IFN- γ has been demonstrated to play a critical role in the regulation of CD8 T cell expansion and contraction during immune responses.

Thank you for this suggestion, we will work to incorporate this in future models.

Reviewer #3: I'd like to thank the authors for doing such an extensive and complete response on this round of revision, well done!

Thank you for your suggestions throughout this process to improve our manuscript.